# Early methionine availability attenuates T cell exhaustion

Piyush Sharma [1] ✉, Ao Guo[1,2], Suresh Poudel[1], Emilio Boada-Romero [1],
Katherine C. Verbist[1], Gustavo Palacios [1], Kalyan Immadisetty[3,4],
Mark J. Chen [1,4], Dalia Haydar[5], Ashutosh Mishra[6], Junmin Peng [7],
M. Madan Babu [3], Giedre Krenciute [5], Evan S. Glazer [8] &
Douglas R. Green [1] ✉

T cell receptor (TCR) activation is regulated in many ways, including niche-specific nutrient availability. Here we investigated how methionine (Met) availability and TCR signaling interplay during the earliest events of T cell activation affect subsequent cell fate. Limiting Met during the initial 30 min of TCR engagement increased $Ca^{2+}$ influx, NFAT1 (encoded by *Nfatc2*) activation and promoter occupancy, leading to T cell exhaustion. We identified changes in the protein arginine methylome during initial TCR engagement and identified an arginine methylation of the $Ca^{2+}$-activated potassium transporter KCa3.1, which regulates $Ca^{2+}$-mediated NFAT1 signaling for optimal activation. Ablation of KCa3.1 arginine methylation increased NFAT1 nuclear localization, rendering T cells dysfunctional in mouse tumor and infection models. Furthermore, acute, early Met supplementation reduced nuclear NFAT1 in tumor-infiltrating T cells and augmented antitumor activity. These findings identify a metabolic event early after T cell activation that affects cell fate.

For an effective antitumor response, CD8[+] T cells are activated in draining lymph nodes (LN) (dLN) and in the tumor microenvironment (TME)[1,2]. The TME is a nutrient-poor niche in which the cytotoxic function of T cells is curtailed, at least in part by the process of T cell exhaustion[3]—a differentiation state characterized by a loss of functional and proliferative capacity[4]. Until now, exhaustion has been understood to be the result of chronic antigen exposure and associated changes in specific epigenetic signatures[5]. However, the acquisition of exhaustion signatures can occur early after T cell activation[6], suggesting that, in addition to chronic antigen exposure, this program can be influenced by factors acting during initial activation.

T cell activation leads to a rapid increase in metabolic and proliferative capacities necessary for T cell function and expansion, and changes in nutrient availability have been shown to modulate activation[7]. Methionine (Met)—an essential amino acid—is the sole donor for cellular methylome maintenance. Tumor cells act as a Met sink, depriving T cells of Met in the TME, and supplementation of Met enhances antitumor immunity[8]. It is unclear; however, how extracellular Met availability affects TCR-mediated T cell activation and fate.

Here we show that Met is consumed rapidly by T cells upon activation and is critical for maintenance of TCR ligation-induced proteomic methylation. CD8[+] T cell activation in reduced Met conditions drives

[1]Department of Immunology, St. Jude Children's Research Hospital, Memphis, TN, USA. [2]Department of Oncology, National Key Laboratory of Immune Response and Immunotherapy, The First Affiliated Hospital of USTC, Center for Advanced Interdisciplinary Science and Biomedicine of IHM, School of Basic Medical Sciences, Division of Life Sciences and Medicine, University of Science and Technology of China, Hefei, China. [3]Department of Structural Biology, Center of Excellence in Data Driven Discovery, St. Jude Children's Research Hospital, Memphis, TN, USA. [4]Department of Laboratory Medicine, Cleveland Clinic, Cleveland, OH, USA. [5]Department of Bone Marrow Transplantation and Cellular Therapy, St. Jude Children's Research Hospital, Memphis, TN, USA. [6]Center for Proteomics and Metabolomics, St. Jude Children's Research Hospital, Memphis, TN, USA. [7]Department of Structural Biology, St. Jude Children's Research Hospital, Memphis, TN, USA. [8]Department of Surgery, University of Tennessee Health Science Center, Memphis, TN, USA. ✉e-mail: piyush.sharma@stjude.org; douglas.green@stjude.org

T cells toward exhaustion. We found that arginine methylation of the calcium-activated potassium transporter KCa3.1 was reduced in Met-limited conditions, increasing Ca$^{+2}$-NFAT1 signaling and promoting exhaustion. Met supplementation in the peri-tumor environment decreased T cell exhaustion and promoted antitumor immunity. Dietary Met supplementation complemented anti-programmed cell death protein 1 (PD-1) treatment for tumor control and animal survival. Together, these data identify an early role for Met in T cell activation, affecting T cell fate decisions.

## Met limitation promotes T cell exhaustion

As nutrient metabolism ensures optimal T cell activation[9], we activated OVA-specific transgenic TCR (OT-I) CD8$^+$ T cells in complete medium and performed targeted mass spectrometry. A decrease in intracellular Met and other amino acids was observed as early as 10 min (Fig. 1a). Met is responsible for cellular methylome maintenance through the S-adenosyl Met (SAM) pathway, and alterations in this pathway adversely affect T cell function[10,11]. In complete medium, no changes were observed in SAM or S-adenosyl homocysteine (SAH) (Fig. 1b, upper panel). In Met-deficient medium, T cells depleted the intracellular Met pool, correlating with decreases in SAM and corresponding increases in SAH (Fig. 1b, bottom panel). Therefore, extracellular Met availability is required to maintain the SAM cycle during T cell activation.

Mice infected with chronic lymphocytic choriomeningitis virus (LCMV) have ~65% less Met in serum as early as day 2 postinfection[12] (Extended Data Fig. 1a). To determine whether Met limitation during TCR engagement affects T cell proliferation, we lowered Met to 0.03 mM in the culture medium. Cell-trace violet (CTV)-labeled OT-I CD8$^+$ T cells were activated with control (0.1 mM) or 0.03 mM Met for 30 min to 6 h, after which Met was restored to 0.1 mM, with no effects on proliferation (Fig. 1c). We then transferred these cells into mice bearing B16 or MC38 tumors expressing ovalbumin (B16-OVA, MC38-OVA) 24 h after activation (Fig. 1d), and observed defective tumor control in T cells activated in 0.03 mM Met compared to 0.1 mM Met across all timepoints, with the earliest being 30 min (Rag1$^{-/-}$ mice (Fig. 1e–g); WT mice (Extended Data Fig. 1b,c)). Similarly, we generated LCMV GP33-specific memory T cells and activated them with GP33 peptide in either 0.1 mM Met or 0.03 mM Met for 30 min, after which Met was restored to 0.1 mM for 24 h, and then cells were transferred into B16 expressing GP33 peptide (B16-GP33) tumor-bearing Rag1$^{-/-}$ mice (Extended Data Fig. 1d). We observed defective tumor control leading to poor survival in mice receiving T cells activated in 0.03 mM Met compared to 0.1 mM Met (Fig. 1h,i).

Assay for transposase-accessible chromatin with high-throughput sequencing (ATAC-seq) at 24 h postactivation (Extended Data Fig. 1e) revealed increased chromatin accessibility in T cells activated in 0.03 mM Met during the initial 30 min (Extended Data Fig. 2a,b). Over-representation analysis identified enrichment of exhaustion-linked genesets (Extended Data Fig. 2c), and HOMER analysis showed that transcription factor (TF) motifs associated with exhausted T cells[13,14] were highly accessible in 0.03 mM-activated T cells (Extended Data Fig. 2d).

We activated OT-I CD8$^+$ T cells in either control or 0.03 mM Met for the initial 30 min followed by 24 h culture in 0.1 mM Met and injected them into B16-OVA-bearing Rag1$^{-/-}$ mice. Tumor-infiltrating lymphocytes (TILs) were collected at day 9 after T cell injection. Again, animals receiving T cells activated initially in 0.03 mM Met showed defective tumor control (Extended Data Fig. 2e,f). The CD8$^+$ TILs displayed reduced CD62L$^+$, CD62L$^{high}$CD44$^{high}$ central memory cells (T$_{CM}$) with increased TOX expression and reduced interferon-gamma (IFNγ) production compared to TILs from T cells activated in Met-replete medium (Extended Data Fig. 2g).

T cell dysfunction is known to contribute to poor prognoses in cancer[15]. This dysfunctional state is in part driven by TOX[16] and

is accompanied by increased surface expression of PD-1 and Tim-3 and reduced effector cytokine production[4]. We activated CD45.1$^+$ or CD45.2$^+$ OT-I CD8$^+$ T cells in 0.1 (control) or 0.03 mM Met for 30 min, restored Met to 0.1 mM for 24 h, and mixed them at a ratio of 1:1 before transferring them into B16-OVA tumor-bearing Rag1$^{-/-}$ mice (Fig. 2a). Even though CD8$^+$ TIL initially activated in 0.03 mM Met outnumbered the 0.1 mM-activated CD8$^+$ TIL at day 12 (Fig. 2b,c), these cells exhibited increased PD-1 and Tim-3, reduced effector cytokines (Fig. 2d,e and Extended Data Fig. 2h), increased TOX expression (Fig. 2f,g) and reduced expression of the stemness-associated TF, TCF1 (Extended Data Fig. 2i). Altogether, these results suggest that reduced Met during early TCR signaling promotes acquisition of epigenomic changes associated with exhaustion and drives T cells toward a dysfunctional, exhausted state.

Exhausted T cells have been delineated as precursors of exhaustion (Tex$^{prog}$), which produce a replicative burst and effector function upon checkpoint blockade[17], and nonfunctional terminally exhausted cells (Tex$^{term}$)[18,19]. Although TOX is associated with maintenance of exhausted T cells, it is also expressed in T$_{EXprog}$[20], but the populations are differentiated by expression of TCF1 and Tim-3 (ref. 19). We therefore further dissected the TOX$^+$CD8$^+$ T cells from the experiment in Fig. 2f (Fig. 2h) and found decreased Tex$^{prog}$ (TCF1$^+$Tim-3$^-$) and increased Tex$^{term}$ (TCF1$^-$Tim-3$^+$) among T cells initially activated in 0.03 mM Met, suggesting a decreased pool of Tex$^{prog}$ cells (Fig. 2i). Furthermore, evaluation of total CD8$^+$ TILs from mice receiving T cells initially activated in 0.03 mM Met for 30 min revealed a lower frequency of Tex$^{prog}$ defined as TCF1$^+$ Tim-3$^-$ and Ly108$^+$Tim-3$^-$, and a corresponding increase in the frequency of Tex$^{term}$ cells defined as TCF1$^-$Tim-3$^+$, Ly108$^-$Tim-3$^+$ [19,21] (Extended Data Fig. 2j,k). Furthermore, the memory-associated markers CD127$^+$CD27$^+$ and CD62L$^+$ (Extended Data Fig. 2l–n) were reduced in T cells initially activated in 0.03 mM Met. Using another tumor model, MC38, CD8$^+$ TILs from mice receiving T cells initially activated in 0.03 mM Met for 30 min revealed increased Tex (PD-1$^+$Tim-3$^+$) and Tex$^{term}$ (Ly108$^-$Tim-3$^+$) as compared to T cells initially activated in 0.01 Met (Extended Data Fig. 2o).

We performed RNA sequencing (RNA-seq) of day 9 TIL from T cells originally activated for 30 min in 0.1 mM or 0.03 mM Met and then cultured in 0.1 mM Met for 24 h before transfer into B16-OVA tumor-bearing Rag1$^{-/-}$ mice. Among the observed transcriptomic changes, most represented increased expression of differentially expressed (DE) genes in TIL initially activated in 0.03 mM Met (Extended Data Fig. 3a,b). Over-representation analysis of the DE genes in T cells activated in 0.1 mM Met revealed enrichment of Gene Ontology Biological Process (GOBP) pathways associated with T cell effector functions along with increases in TFs associated with a stem-like T cell state and memory generation[22,23] (Extended Data Fig. 3c). Day 9 TIL initially activated in 0.03 mM Met showed enrichment of genes associated with terminal differentiation or dysfunction of T cells and TFs associated with hyperactivation and development and maintenance of T cell exhaustion[13,24,25] (Extended Data Fig. 3d). Geneset enrichment analysis (GSEA) also showed an enriched exhaustion-associated geneset[26,27] in CD8$^+$ TIL initially activated in 0.03 mM Met (Fig. 2j) and enrichment of hallmark genesets associated with T cell effector function in the control (0.1 mM Met) T cells (Extended Data Fig. 3e). These results suggest that Met availability during early TCR engagement shapes T cell differentiation, as reduced Met during initial activation promotes hyperactivation, driving T cells toward exhaustion.

## Extracellular Met metabolism regulates Ca$^{2+}$-NFAT1 axis

Among the earliest events in T cell activation is the influx of Ca$^{2+}$ (refs. 28,29); activation of calcineurin; and subsequent dephosphorylation and nuclear translocation of nuclear factor of activated T cells-1 (NFAT1)[30,31]. Because low Met levels for only the first 30 min of activation influenced the T cell phenotype, we investigated changes in Ca$^{2+}$ flux.

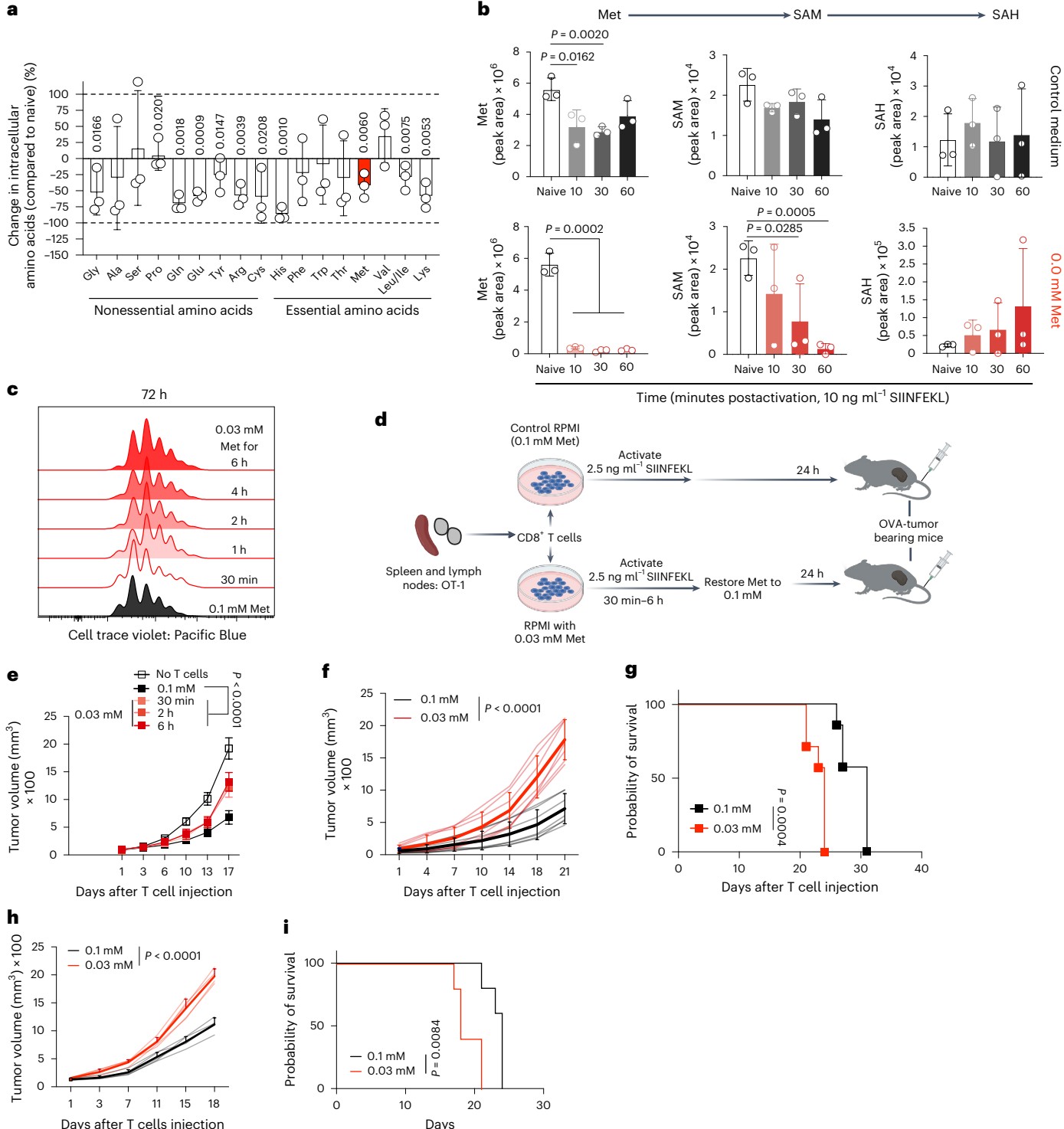

**Fig. 1 | TCR-mediated, rapid Met consumption governs T cell effector function. a,b**, Quantification of intracellular amino acids at 10 min (**a**) and SAM and SAH up to 60 min (**b**) in OT-I T cells activated with 10 ng ml⁻¹ SIINFEKL (*n* = 3 biological replicates). **c**, T cell proliferation by means of cell-trace violet staining of OT-I CD8⁺ T cells activated in either 0.1 mM Met or 0.03 mM Met for the times indicated before restoration to 0.1 mM Met in 0.03 mM Met conditions and then analyzed 72 h postactivation. Representative of three biological replicates per group. **d**, Schematic design for OT-I T cell activation initially in 0.1 or 0.03 mM Met, followed by restoration of Met to 0.1 mM for 24 h before injection into B16-OVA tumor-bearing mice. **e**, Tumor growth of B16-OVA in *Rag1⁻/⁻* treated with

OT-I CD8⁺ T cells activated as described in **d** for 30 min–6 h (*n* = 5 mice per group). **f,g**, Tumor growth (**f**) and survival (**g**) of B16-OVA tumors in *Rag1⁻/⁻* mice after transfer of 24-h-activated OT-I T cells with first 30 min of stimulation being in 0.1 or 0.03 mM Met with 2.5 ng ml⁻¹ SIINFEKL before restoration to 0.1 mM Met (*n* = 5 mice per group). **h,i**, Tumor growth (**h**) and survival (**i**) of B16-OVA tumors treated with activated GP33⁺-memory T cells as described in Extended Data Fig. 1d (*n* = 5 mice per group). Data are mean ± s.d. Paired two-tailed Student's *t*-test (**a**), unpaired one-tailed Student's *t*-test (**b**), two-way ANOVA (**e**, **f**, **h**) and Mantel–Cox log rank test (**g**, **i**). Illustrations in **d** created with BioRender.com.

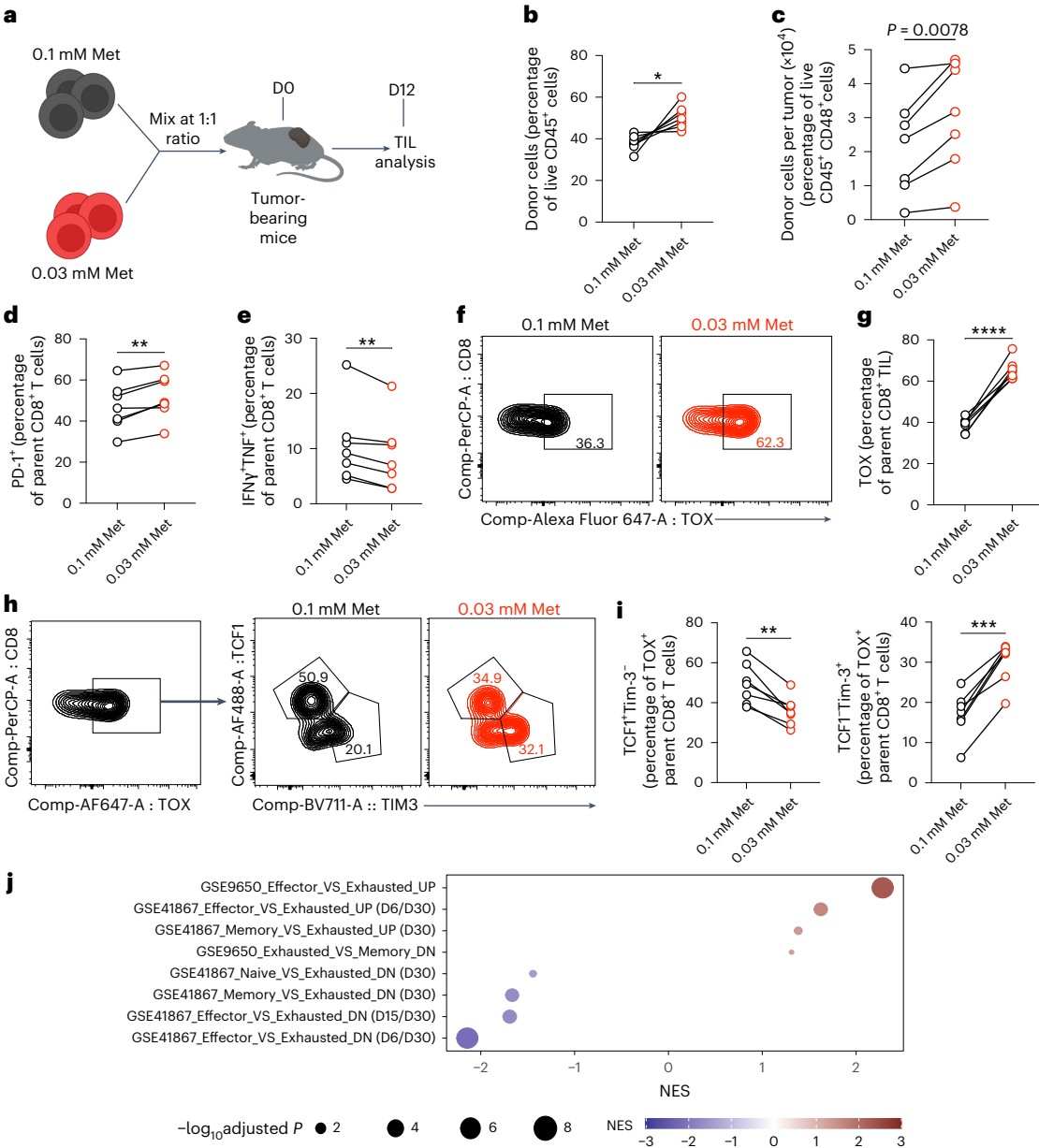

**Fig. 2 | Reduced Met availability during TCR signaling promotes T cell exhaustion. a**, Schematic of experimental design. OT-I CD8⁺ T cells with different congenic markers were initially activated in 0.1 mM or 0.03 mM Met for 30 min with replenishment of 0.1 mM Met for 24 h, transferred into a B16-OVA tumor-bearing *Rag1*⁻/⁻ mouse at a 1:1 ratio (day 0 (D0) and analyzed on day 12 (D12) after T cell transfer. **b,c**, Frequencies (**b**) and absolute number (**c**) of transferred T cells isolated from B16-OVA tumors at D12 posttransfer (*n* = 7 mice per group). **d,e**, Frequency of PD-1⁺ (**d**) and IFNγ⁺TNF⁺ (**e**) OT-I CD8⁺ TIL, initially activated in 0.1 mM and 0.03 mM Met and assessed at D12 postinjection (as in **a**) (*n* = 7

mice per group). **f,g**, Contour plot (**f**) and frequency of TOX⁺ cells (**g**) and of OT-I CD8⁺ TIL as in **d** (*n* = 7 mice per group). **h,i**, Representative contour plot (**h**) and quantification (**i**) of TCF1- and Tim-3-expressing cells from TOX⁺CD8⁺ TIL from OT-I CD8⁺ T cells isolated at D12 posttransfer as in **a** (*n* = 7 mice per group). **j**, Standard names of T cell exhaustion genesets in GSEA analysis of RNA-seq of OT-I CD8⁺ TIL on D9 after T cell injection as in Fig. 1f (*n* = 4). NES, normalized enrichment score. Data are mean ± s.d.; paired two-tailed Student's *t*-test. Illustrations in **a** created with BioRender.com.

We observed increased Ca²⁺ influx upon stimulation with anti-CD3/CD28 antibody in 0.03 mM compared to 0.1 mM Met (Fig. 3a–c and Extended Data Fig. 4a–d). The activation-induced Ca²⁺ flux was reduced by the calcium release-activated channel (CRAC) inhibitor YM-58483 (Extended Data Fig. 4e–g).

NFAT1 cooperates with the AP1 complex to ensure optimal T cell activation, but high levels of NFAT1 activity are associated with exhaustion[30,32]. Confocal imaging of NFAT1 activation at the single-cell level revealed an increase in activated NFAT1 in T cells activated for 30 min with anti-CD3/28 Dynabeads in 0.03 mM Met compared to 0.1 mM Met (Fig. 3d and Extended Data Fig. 5a), quantified as either

a ratio of nuclear NFAT1 to total cell NFAT1 (Fig. 3e and Extended Data Fig. 5b (0.0 mM Met)) or as nuclear NFAT1 fluorescence intensity (Extended Data Fig. 5c), which were abolished upon treatment with either cyclosporin A (CsA; calcineurin inhibitor) (Fig. 3d,e) or YM-58483 (CRAC inhibitor) (Extended Data Fig. 5d). Examination of NFAT2 at 30 min of activation revealed no difference in its nuclear localization in T cells activated in either 0.1 mM or 0.03 mM Met (Extended Data Fig. 5e). Next, we activated CD8⁺ T cells in complete medium for 24 h and rested them for 48 h before secondary activation in 0.03 mM or 0.1 mM Met and again observed increased nuclear NFAT1 levels in 0.03 mM Met, 30 min after activation (Extended Data

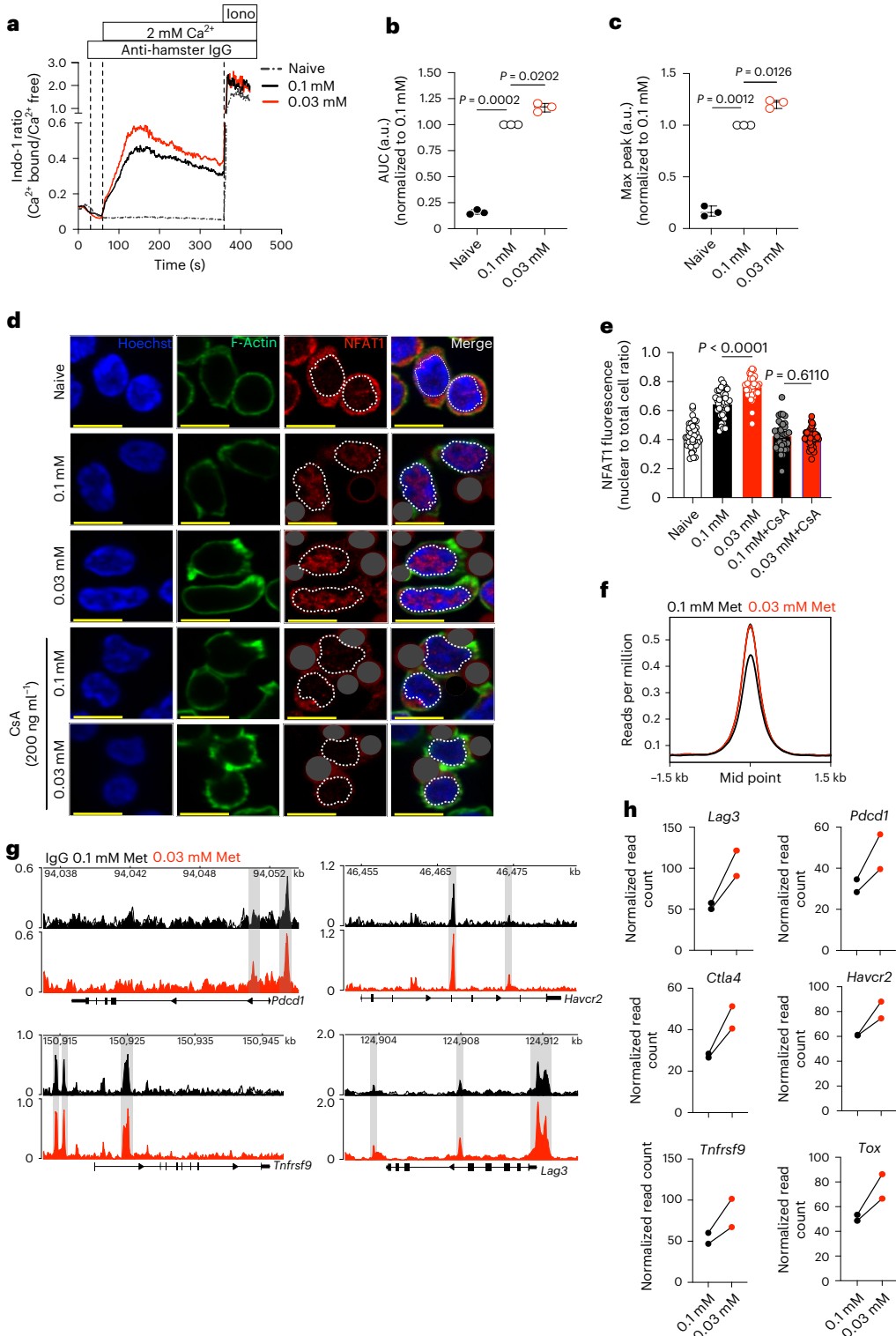

**Fig. 3 | Extracellular Met availability regulates Ca²⁺-mediated NFAT1 activity.**
**a**, Representative plot of Indo-1 analysis of Ca²⁺ flux of either CD8⁺ T cells naive or activated with anti-CD3 and anti-CD28 by anti-hamster IgG crosslinking in either 0.1 mM or 0.03 mM Met-containing Ca²⁺-free Ringer solution with addition of 2 mM Ca²⁺ to measure Ca²⁺ influx (*n* = 3 biological replicates per group). iono, ionomycin. **b,c**, Area under the curve (AUC) (**b**) and maximum peak signal (**c**) of the calcium flux from **a**, normalized to values of activated cells in 0.1 mM Met (*n* = 3 biological replicates per group). **d**, Representative images of T cells, cultured with or without anti-CD3/28 Dynabeads (dark gray masked), in 0.1 mM and 0.03 mM Met, with or without CsA treatment for 30 min, stained for NFAT1 (red), Hoechst (blue) and Phalloidin (green) (dashed lines indicate nuclei). The identical figure without masking or nuclear demarcation is shown in Extended

Data Fig. 5a. **e**, Quantification of NFAT1 intensity as nuclear to total cell ratio (**c**). Each circle represents one cell, *n* = 40 cells per group. Scale bar, 10 μm.
**f**, Histogram of NFAT1-binding signals (read count per million reads normalized to background) from NFAT1 CUT&RUN on T cells initially activated in 0.1 or 0.03 mM Met for 30 min and assessed 24 h postactivation. **g,h**, NFAT1 CUT&RUN peaks at the known target genes (**g**) and quantification of known NFAT1-binding regions of *Lag3*, *Pdcd1*, *Havcr2*, *Ctla4* and *Tnfrsf9* and *Tox* (**h**, normalized read count (see differential binding analysis)) in T cells initially activated in 0.1 or 0.03 mM Met for 30 min and assessed 24 h postactivation (*n* = 2 biological replicates per group). Data are mean ± s.d. Boxplots shows minimum and maximum value with median as center. Paired two-tailed Student's *t*-test (**b** and **c**), unpaired two-tailed Student's *t*-test (**e**).

Fig. 5f), suggesting that the effect of limiting Met on NFAT1 activation is independent of previous activation events.

NFAT1 is known to induce the transcription of several exhaustion-associated genes such as *Lag3*, *Pdcd1*, *Havcr2*, *Ctla4* and *Tnfrsf9* (ref. 32). We performed NFAT1 CUT&RUN on T cells activated in 0.1 mM or 0.03 mM Met for 30 min, followed by 0.1 mM Met for 24 h and observed increased NFAT1 binding in the low Met group (Fig. 3f). We observed that Met limitation for the initial 30 min resulted in increased NFAT1 binding to *Pdcd1*, *Havcr2*, *Lag3* and *Tnfrsf9* as well as *Ctla4 and Tox* gene loci (Fig. 3g,h). This correlated with increased transcription of most of these genes at 24 h (Extended Data Fig. 5g). These results suggest that reduced Met availability during initial TCR signaling drives T cells toward hyperactivation and eventual exhaustion, whereas sustained extracellular Met levels during TCR engagement optimizes $Ca^{2+}$-mediated NFAT1 signaling and subsequent T cell fate.

## KCa3.1 methylation regulates NFAT1

Met metabolism is the sole pathway for epigenetic and proteomic methylome maintenance, which affects T cell function[8]. When converted to SAH, SAM acts as the methyl donor; thus, the ratio of SAM to SAH is known as the cellular methylation potential[33]—an index of the capacity of a cell to maintain its methylome. We found that the SAM/SAH ratio was reduced as early as 10 min post-TCR engagement in the absence of extracellular Met (Extended Data Fig. 6a). Met metabolism is essential for DNA methylation[34]; however, we found no significant differences in global 5-methylcytosine levels at either 30 min or 24 h in T cells activated as above (Extended Data Fig. 6b). We similarly assessed H3K4me3 and H3K27me3 by CUT&RUN at 24 h postactivation and, again, no global changes were observed (Extended Data Fig. 6c,d). Thus, Met restriction for the initial 30 min of T cell activation does not seem to directly affect global DNA or histone methylation.

Since Met restriction influenced early methylation events independently of direct effects on DNA and histone methylation, we examined other potential methylation events. Post-translational protein arginine methylation mediated by protein arginine methyltransferases regulates protein function[35]. We found that arginine methylation in T cells activated in 0.0 mM or 0.03 mM Met was reduced 30 min postactivation when compared to 0.1 mM Met (Fig. 4a,b and Extended Data Fig. 6e). To identify differentially methylated proteins upon TCR engagement in limited Met, CD8+ T cells activated for 30 min were processed for methylarginine immunoprecipitation and quantification using multiplexed tandem mass tag-mass spectrometry (TMT-MS). Among common proteins differentially methylated in 0.0 mM and 0.03 mM Met activation conditions compared to 0.1 mM Met (Fig. 4c), we found a top candidate, KCa3.1, to be dimethylated at R350 in 0.1 mM Met, which was reduced in 0.0 mM and 0.03 mM Met (Fig. 4d and Supplementary Table 1). Furthermore, KCa3.1 R350 is conserved across species, including humans, suggesting that R350 is critical for KCa3.1 regulation (Extended Data Fig. 6f).

KCa3.1 is a calcium-activated potassium transporter encoded by *Kcnn4* and is activated by calmodulin (CaM) binding upon TCR-mediated $Ca^{2+}$ influx[36]. During T cell activation, the plasma membrane is hyperpolarized to ensure sufficient $Ca^{2+}$ flux for signaling[37], which is maintained by $K^+$ efflux through potassium transporters such as KCa3.1 (refs. 38,39). We hypothesized that R350 methylation regulates KCa3.1 activity, and loss of this methyl group may result in hyperactivation, leading to increased $Ca^{2+}$ influx and NFAT1 activation. To test this, we performed molecular dynamics (MD) simulations with the solved crystal structure of human KCa3.1 (hKCa3.1) bound to CaM at R352 (corresponding to R350 in mouse KCa3.1) (Extended Data Fig. 6g). The CaM-binding pocket encompasses D78-T79-D80-S81-E82-E83-E84, which is predominantly negatively charged/polar, where E84 forms a salt bridge with R352 of hKCa3.1. We found that both symmetrical dimethylation (SDMA) and asymmetrical dimethylation (ADMA) of R352 introduced a hydrophobic methyl group into the negatively

charged/polar CaM-binding pocket, weakening the hKCa3.1 and CaM interaction required for KCa3.1 function (Fig. 4e). Furthermore, MD simulations showed that the salt bridge interaction of R352 with E84 of the CaM-binding pocket diminished over time, especially in SDMA where both guanidino nitrogens are methylated, compared to single nitrogen methylation in ADMA (Fig. 4f and Extended Data Fig. 6h), suggesting that disruption of arginine methylation promoted stabilization of CaM binding, leading to increased activation and $Ca^{2+}$ influx.

To validate our in silico results, we used the chemical inhibitor TRAM-34 to reduce KCa3.1 activity[40,41] during T cell activation and observed reduced $Ca^{2+}$ flux only in 0.03 mM Met and not in 0.1 mM Met (Fig. 4g–i and Extended Data Fig. 6i) at the concentration of TRAM-34 we employed. Furthermore, TRAM-34 treatment normalized nuclear NFAT1 levels in T cells activated in 0.03 mM Met in a dose-dependent manner and had no effect on T cells activated in 0.1 mM Met (Extended Data Fig. 6j). We then activated OT-I T cells in 0.1 mM or 0.03 mM Met with either dimethylsulfoxide (DMSO) or TRAM-34 for 30 min, washed and cultured cells for an additional 24 h in 0.1 mM Met before transferring into B16-OVA tumor-bearing *Rag1*$^{-/-}$ mice (Extended Data Fig. 6k). Chemical inhibition of KCa3.1 for 30 min significantly improved antitumor activity in T cells initially activated in 0.03 mM Met (Fig. 4j), suggesting that increased KCa3.1 activity was responsible for the eventual decreased T cell function under this activation condition.

## Loss of KCa3.1 methylation increases TCR-induced NFAT1 activation

To specifically interrogate the role of R350 methylation of KCa3.1 in T cell activation, we cloned wild-type KCa3.1 (Kca3.1$^{WT}$) and a mutant with an alanine substitution (KCa.3.1$^{R350A}$) into retroviral vectors and expressed them in primary mouse CD8+ T cells in which endogenous KCa3.1 was deleted using CRISPR–Cas9 (Extended Data Fig. 7a–c). Expression of the two was similar (Extended Data Fig. 7d). $Ca^{2+}$ flux increased upon activation in T cells expressing KCa3.1$^{R350A}$ compared to those expressing Kca3.1$^{WT}$ (Fig. 5a–c and Extended Data Fig. 7e), which correlated with increased nuclear NFAT1 at 30 min (Fig. 5d,e and Extended Data Fig. 7f). This heightened nuclear NFAT1 in cells expressing KCa3.1$^{R350A}$ was visible under resting conditions and increased upon activation in a time-dependent manner (Extended Data Fig. 7g).

To investigate the epigenomic state of WT or mutant KCa3.1 T cells, we analyzed global chromatin accessibility by means of ATAC-seq after 24 h of in vitro activation. We found that the activated KCa3.1$^{R350A}$ T cells showed increased chromatin accessibility compared to KCa3.1$^{WT}$ T cells (Fig. 5f and Extended Data Fig. 8a). Furthermore, GSEA analysis revealed enrichment of GOBP terms associated with negative regulators of T cell-mediated immunity and genesets associated with T cell exhaustion[26,27] in T cells expressing mutant KCa3.1$^{R350A}$ (Fig. 5g,h), suggesting that ablation of R350 methylation drives T cells toward exhaustion early after activation. Upon comparing these ATAC-seq results with ATAC-seq of T cells activated in 0.03 mM versus 0.1 mM Met (Extended Data Fig. 2b), we found several common differentially accessible regions (DAR) (Extended Data Fig. 8b) with enrichment of T cell exhaustion-associated genesets (Extended Data Fig. 8c). HOMER motif analysis of WT or mutant T cells and common DARs was similarly enriched (Extended Data Figs. 2d and 8d,e). Despite these results, KCa.3.1$^{R350A}$ did not fully phenocopy the chromatin effects of activation in 0.03 mM Met, suggesting that other differentially methylated proteins in addition to KCa.3.1 may also impact chromatin accessibility following activation in the latter condition.

To investigate whether the observed exhaustion-specific chromatin state led to defective functionality of mutant KCa3.1 T cells, we transferred T cells expressing KCa.3.1$^{WT}$ or KCa.3.1$^{R350A}$ into B16-OVA tumor-bearing Rag1$^{-/-}$ mice. We found that T cells expressing KCa.3.1$^{R350A}$ displayed impaired tumor control and survival (Fig. 5i–k), suggesting that the absence of KCa3.1 methylation sets T cells on an exhaustion trajectory, resulting in impaired effector function.

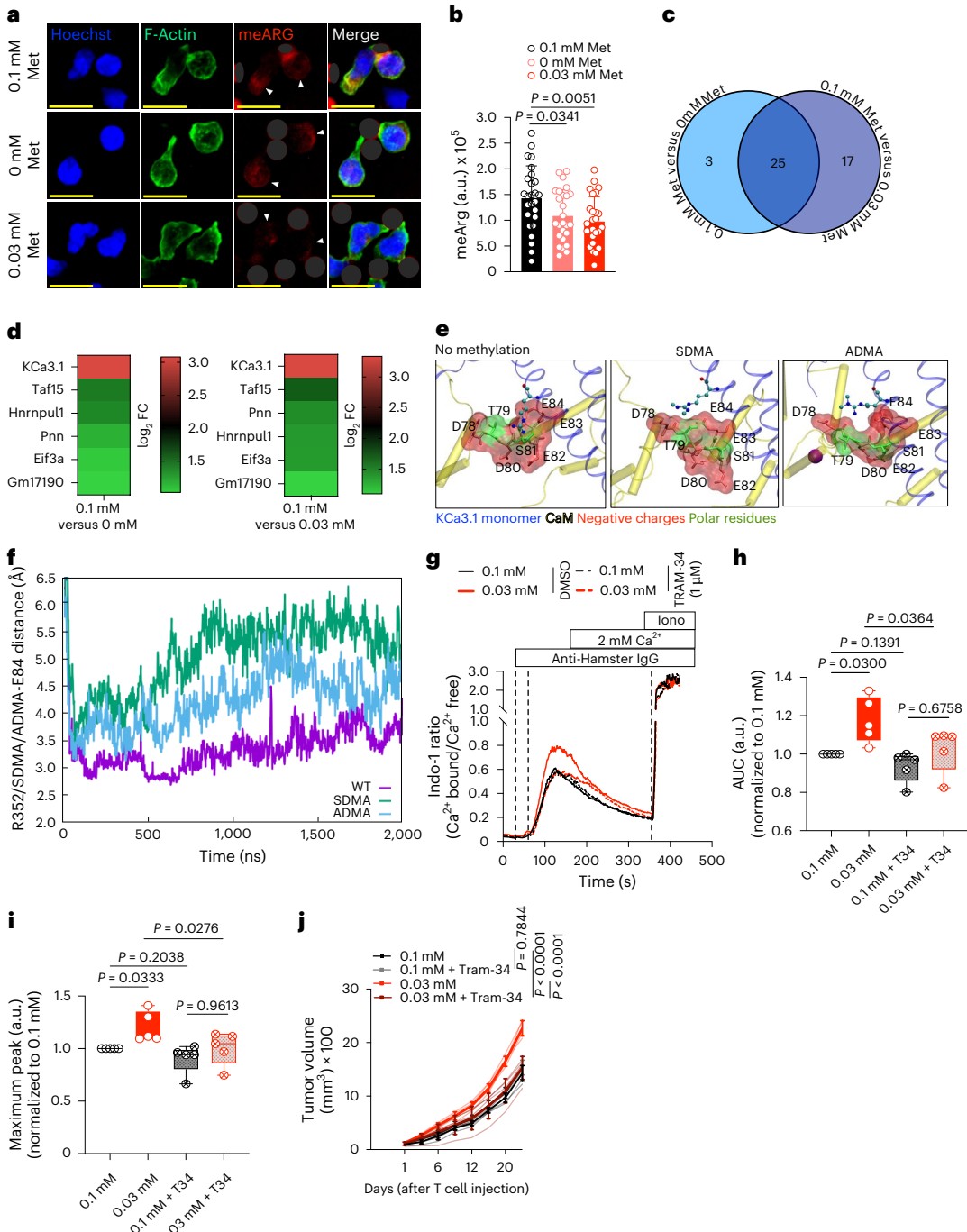

**Fig. 4 | TCR activation-mediated KCa3.1 methylation regulates T cell effector function. a**,**b**, Representative images (**a**) and quantification (**b**) of methylarginine (meArg) in CD8⁺ T cells activated with anti-CD3/28 Dynabeads (dark gray masked) in 0.1, 0 and 0.03 mM Met for 30 min, stained with pan-meArg (red, with arrows), Hoechst (blue) and Phalloidin (green) (each circle represents one cell; $n$ = 26 cells per group. Scale bar, 10 μm. **c**, Venn diagram of enriched meArg proteins at 30 min postactivation of T cells in 0.1 mM versus 0 mM or 0.1 mM versus 0.03 mM Met, by anti-CD3:IgG crosslinking for 30 min, as identified by TMT-MS. **d**, Heatmap of TMT-MS-identified proteins ($P$ < 0.05) with enriched meArg in T cells activated in 0.1 mM versus 0 mM or 0.03 mM Met as in **c** ($n$ = 2 biological replicates per group). **e**, Predicted interaction of KCa3.1 monomer with CaM as unmethylated (left), SDMA (center) and ADMA (right). The CaM-binding pocket is represented as surface rendering with interacting amino acids as sticks and R352 as ball and sticks. **f**, MD simulation analysis. Analysis of

variations of distance of salt bridge between CaM E84 and demethylated R352 WT (purple), SDMA R352 (green) and ADMA R352 (blue) over the course of simulation time (averaged across all four monomers, three MD trials). **g**, Representative plot of Indo-1 analysis of Ca²⁺ flux in CD8⁺ T cells activated with anti-CD3 and anti-CD28 by anti-hamster IgG crosslinking in Ca²⁺-free Ringer solution supplemented with either 0.1 mM or 0.03 mM Met and treated with either with DMSO or 1 μM TRAM-34 ($n$ = 5 biological replicates per group). **h**,**i**, AUC (**h**) and maximum peak signal (**i**) from **g** normalized to 0.1 mM Met ($n$ = 5 biological replicates per group). **j**, Tumor growth of B16-OVA tumors in $RagI^{-/-}$ mice after transfer of OT-I CD8⁺ T cells activated in 0.1 mM or 0.03 mM Met with either DMSO or 1 μM TRAM-34 for 30 min and cultured for 24 h ($n$ = 5 mice per group). Data are mean ± s.d. Boxplots shows minimum and maximum value with median as center. Unpaired two-tailed Student's $t$-test (**b**), paired two-tailed Student's $t$-test (**h**,**i**), two-way ANOVA (**j**).

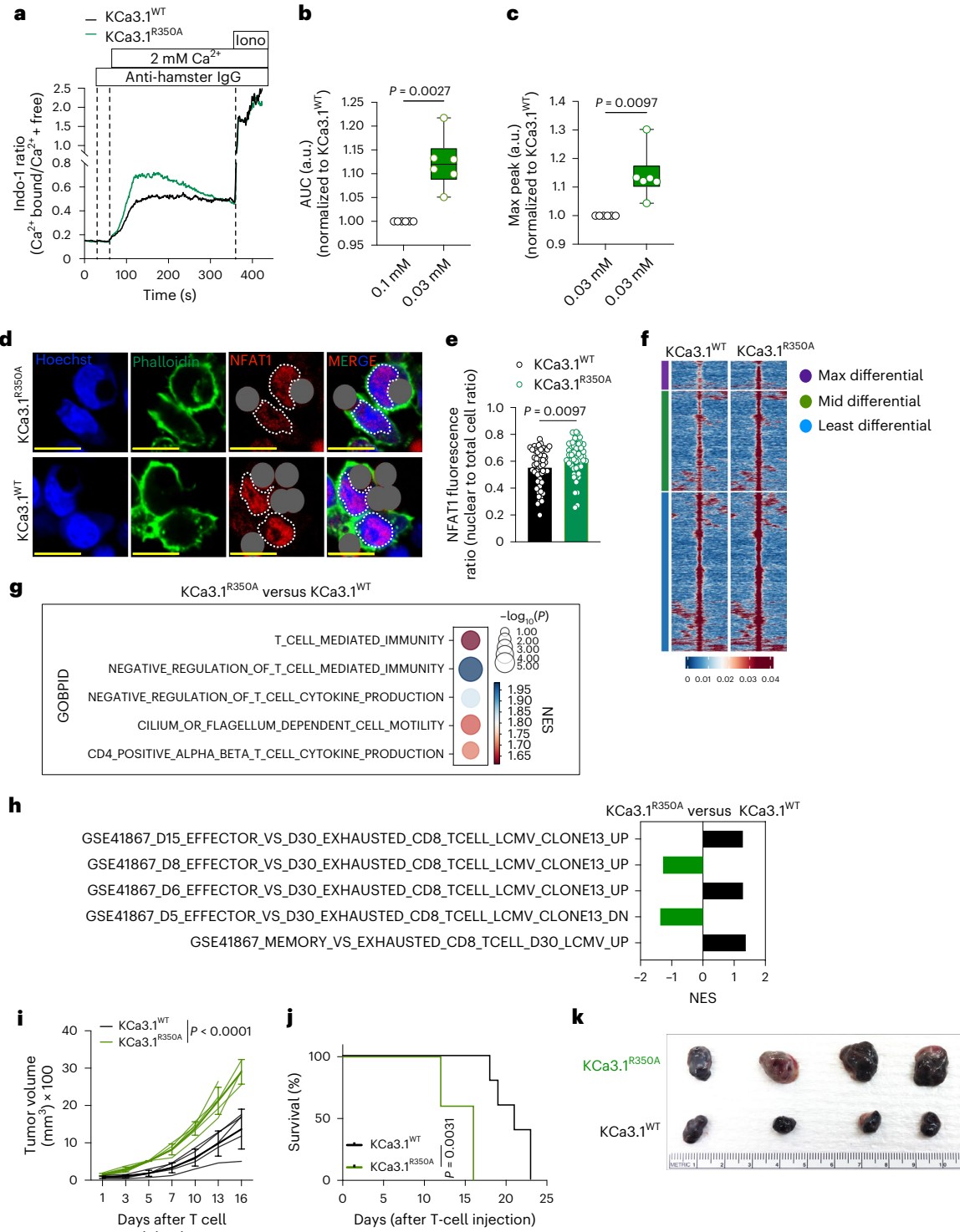

**Fig. 5 | Ablation of KCa3.1 R350 methylation increases Ca²⁺-mediated NFAT1 activity, promoting T cell dysfunction. a**, Representative plot of Indo-1 analysis of Ca²⁺ flux in KCa3.1$^{WT}$ and KCa3.1$^{R350A}$ T cells activated with anti-CD3 and anti-CD28 by anti-hamster IgG crosslinking in Ca²⁺-free Ringer solution ($n = 6$). **b,c**, AUC (**b**) and maximum peak signal (**c**) of calcium flux in **a**, normalized to the values of activated KCa3.1$^{WT}$ ($n = 6$ biological replicates per group). **d,e**, Representative image (**d**) and quantification of NFAT1 intensity as nuclear to total cell ratio (dashed lines indicate nuclei) (**e**) in KCa3.1$^{WT}$ and KCa3.1$^{R350A}$ OT-I T cells activated for 30 min with anti-CD3/28 Dynabeads (dark gray masked) (see Extended Data Fig. 7f for identical, unmasked figure) (each circle represents one cell, $n = 40$ cells). Scale bar, 10 μm. **f**, Chromatin accessibility heatmap of T cells expressing activated KCa3.1$^{WT}$ or KCa3.1$^{R350A}$ with each row representing peaks ($P < 0.05$ and log₂ FC > 1.5) displayed over the span of a 2-kb window with peak

as center (grouped from least to maximum differential region), analyzed 24 h postactivation with 2.5 ng ml⁻¹ SIINFEKL ($n = 3$). **g**, GOBP terms associated with negative regulation of T cells enriched in OT-I T cells expressing KCa3.1$^{R350A}$ versus KCa3.1$^{WT}$ activated as in **b** ($n = 3$). **h**, Standard names of T cell exhaustion genesets in GSEA analysis of differential accessible promoter reads from ATAC-seq of OT-I T cells expressing KCa3.1$^{WT}$ or KCa3.1$^{R350A}$, activated with 2.5 ng ml⁻¹ SIINFEKL for 24 h ($n = 3$). **i,j**, Tumor growth (**i**) and survival (**j**) of B16-OVA tumor-bearing RAG⁻/⁻ mice after transfer of KCa3.1$^{WT}$ and KCa3.1$^{R350A}$ OT-I T cells ($n = 5$). **k**, B16-OVA tumors harvested at D12 post transfer of KCa3.1$^{WT}$ and KCa3.1$^{R350A}$ OT-I T cells ($n = 4$). Data are mean ± s.d. Boxplots shows minimum and maximum values with median as center. Paired two-tailed Student's *t*-test (**b** and **c**), unpaired two-tailed Student's *t*-test (**e**), two-way ANOVA (**i**) and Mantel–Cox log rank test (**j**).

## Ablation of KCa3.1 methylation promotes T cell exhaustion

To determine whether ablation of R350 methylation drives T cell exhaustion in tumors, we transferred T cells expressing KCa.3.1[WT] or KCa.3.1[R350A] into B16-OVA tumor-bearing Rag1[−/−] mice. Tumors were isolated on day 12 after T cell injection and, consistently, much larger tumors were extracted from animals treated with T cells expressing KCa.3.1[R350A] (Extended Data Fig. 9a). KCa.3.1[R350A]-expressing TIL showed high surface expression of the exhaustion markers PD-1, Lag3 and Tim-3 (Extended Data Fig. 9b) that correlated with increased TOX expression (Extended Data Fig. 9c). Among these cells, we also found fewer Tex[prog] (CD69[lo]Ly108[hi])[42] and IFNγ- and tumor necrosis factor (TNF)-producing T cells (Extended Data Fig. 9d,e). Next, we performed a similar experiment in a competitive setting, wherein CD45.1[+]CD45.2[+] KCa.3.1[WT] and CD45.1[+] KCa.3.1[R350A]-expressing OT-I CD8[+] T cells were mixed at a 1:1 ratio and transferred into B16-OVA tumor-bearing Rag1[−/−] mice (Fig. 6a). TIL analysis at D12 showed increased PD-1[+]Tim-3[+] (Fig. 6b), increased TOX and decreased TCF1 expression (Fig. 6c,d) in KCa.3.1[R350A]-expressing T cells, which correlated with reduced expression of the effector cytokines IFNγ and TNF (Fig. 6e). As TCF1 and Ly108 are expressed in Tex[prog] (refs. 13,19), further analysis revealed a decreased frequency of TCF1[+]Tim-3[−] and Ly108[+]Tim-3[−] Tex[prog], and increased TCF1[−]Tim-3[+] and Ly108[−]Tim-3[+] Tex[term] populations (Fig. 6f,g and Extended Data Fig. 9f,g). Reduced TCF1 expression impairs CD62L[+] memory T cell generation[43], and we also observed reduced CD62L[+] cells in KCa.3.1[R350A]-expressing T cells, as compared to T cells expressing WT KCa3.1 (Extended Data Fig. 9h).

T cell exhaustion is a prominent feature of chronic viral infection[44]. We utilized the LCMV model in WT (CD45.2, Thy1.2) mice into which CD45.1[+]CD45.2[+] KCa.3.1[WT] and CD45.2[+] KCa.3.1[R350A]-expressing Thy1.1 (CD90.1[+]) P14[+] T cells were injected at a 1:1 ratio into mice infected with chronic Clone-13 strain. Thy1.1[+] T cells from spleens were analyzed on day 9 postinfection (Fig. 6h and Extended Data Fig. 9i) and we observed a significantly reduced population frequency of KCa.3.1[R350A]-expressing T cells (Extended Data Fig. 9j). KCa.3.1[R350A]-expressing T cells presented with increased TOX and decreased TCF1 expression and increased PD-1[+]Tim-3[+] expression, as compared to KCa3.1[WT] expressing cells (Fig. 6i–k and Extended Data Fig. 9k). Further, we found that KCa.3.1[R350A]-expressing T cells harbored reduced frequencies of TCF1[+]Tim-3[−] and Ly108[+]Tim-3[−] Tex[prog] cells and increased frequencies of TCF1[−]Tim-3[+] and Ly108[−]Tim-3[+] Tex[term] cells at day 9 postinfection (Fig. 6l,m and Extended Data Fig. 9l). Tex[prog] transition to the intermediate T exhausted effector-like (Tex[eff]) state is characterized by Cx3CR1 expression[19]. We found that the KCa.3.1[R350A]-expressing T cells showed increased frequencies of Tim-3[+]Cx3CR1[+]PD-1[+] Tex[eff] (Extended Data Fig. 9m). Thus, ablation of R350 methylation drives T cells toward an exhaustion trajectory at the early stages of infection. Together, these experiments support the idea that R350 methylation restricts KCa3.1 function, ensuring optimal T cell activation and effector function in both tumors and viral infections.

## Acute Met supplementation promotes antitumor T cell function

CD8[+] T cells are activated in secondary lymphoid tissues and within tumors[1,45]. Although T cells are known to be activated primarily in dLNs, a recent study showed that activation of TILs in the tumor microenvironment is necessary to acquire an effector-like phenotype[2]. Our results in Fig. 1h and Extended Data Fig. 5f support the idea that Met is required for optimal TCR activation, and this requirement is independent of the previous activation status of the T cell. Thus, we hypothesized that, upon tumor infiltration, CD8[+] T cells experience antigen presentation in a Met-deficient microenvironment, leading to altered effector function. To evaluate Met levels in TILs, we performed mass spectrometric analysis of CD8[+] TIL and dLN CD8[+] T cells from B16 subcutaneous tumor-bearing mice at day 9. We observed a ~50% reduction

in intracellular Met in CD8[+] TIL compared to that in dLN CD8[+] T cells (Extended Data Fig. 10a and Supplementary Table 2). We also observed a similar decrease in intracellular Met in CD8[+] TIL isolated from primary human colorectal carcinomas compared with corresponding CD8[+] T cells from PBMC (Extended Data Fig. 10b). To determine whether CD8[+] TIL with reduced intracellular Met showed increased nuclear NFAT1, we sorted activated CD44[+] CD8[+] T cells from B16 tumors and respective dLN at day 9 postimplantation and imaged for NFAT1. We observed increased levels of nuclear NFAT1 in CD44[+]CD8[+] TIL compared to counterparts from dLN (Fig. 7a), suggesting that reduced Met availability in the TME may have led to increased NFAT1 activation in those T cells.

Continuous Met supplementation in tumors has been shown to promote antitumor T cell activity, associated with global epigenetic maintenance[8]. Our findings, however, highlight an early role for Met in T cell fate, and we hypothesized that Met supplementation up to that of serum levels at the early stages of tumor development would increase T cell antitumor activity. We therefore injected 50 μl of 61 μM Met peritumorally once per day for 5 days when tumor volumes were 70–100 mm[3] (Extended Data Fig. 10c) and found that treatment decreased nuclear NFAT1 levels in CD44[+] CD8[+] TIL as compared to TIL from Hank's balanced salt solution (HBSS)-injected tumors (Fig. 7b). Indeed, nuclear NFAT1 in TIL was similar to or lower than that in CD8[+] T cells from dLN (Extended data Fig. 10d). Furthermore, acute Met supplementation delayed tumor growth in several mouse tumor models, thereby improving overall survival (Fig. 7c–e and Extended Data Fig. 10e,f). Met supplementation resulted in decreased PD-1 and Tim-3 surface expression in CD8[+] TIL, along with increased CD62L[hi]CD44[hi] Tcm and reduced CD62L[lo]CD44[hi] Tem (Extended Data Fig. 10g,h) compared to TIL in HBSS-treated tumors. In contrast to immunocompetent WT mice, Met injected peritumorally in NSG mice yielded no difference in tumor growth (Extended Data Fig. 10i), suggesting a lymphocyte-mediated effect. Depletion with an anti-CD8 antibody (Extended Data Fig. 10j) in B16-OVA tumor-bearing mice abrogated the antitumor effect of Met supplementation (Fig. 7f), indicating Met supplementation-mediated tumor control is CD8[+] T cell-dependent. Using a contralateral tumor model with B16-OVA implanted in one shoulder and in the opposite abdominal flank and injected Met peritumorally to the latter tumor impeded contralateral tumor growth (Extended Data Fig. 10k,l), further suggesting a T cell response. These results show that acute Met supplementation promoted tumor control by CD8[+] T cells. We also found that an 0.5% increase in dietary Met supplementation in B16 tumor-bearing mice showed better tumor growth control (Extended Data Fig. 10m), concordant with a study showing that Met supplementation in humans increased the effector function of CD8[+] T cells[8]. Met supplementation, therefore, enhances the antitumor function of CD8[+] T cells.

## Met supplementation enhances chimeric antigen receptor and immune checkpoint blockade treatment

Chimeric antigen receptor (CAR)-T cell treatment has been clinically successful in the treatment of hematological malignancies, but less so in solid tumors[46]. In a solid tumor model in Rag1[−/−] mice, we supplemented Met in the presence of transferred mouse B7-H3 CAR-T cells into animals bearing a B7-H3[+] F420 tumor[47] (Extended Data Fig. 10n). Again, Met supplementation improved tumor control and animal survival in this model (Fig. 7g).

Immune checkpoint blockade (ICB) is used widely either alone or in combination with cancer treatments[48]. To query whether Met supplementation might cooperate with ICB, we administered either IgG isotype or anti-PD-1 antibody along with HBSS or Met supplementation in subcutaneous MC38 tumor-bearing mice (Extended Data Fig. 10o). Peritumoral Met supplementation with anti-PD-1 reduced tumor growth and increased survival compared to controls (Extended Data Fig. 10p). As local Met injection may not be feasible in clinical

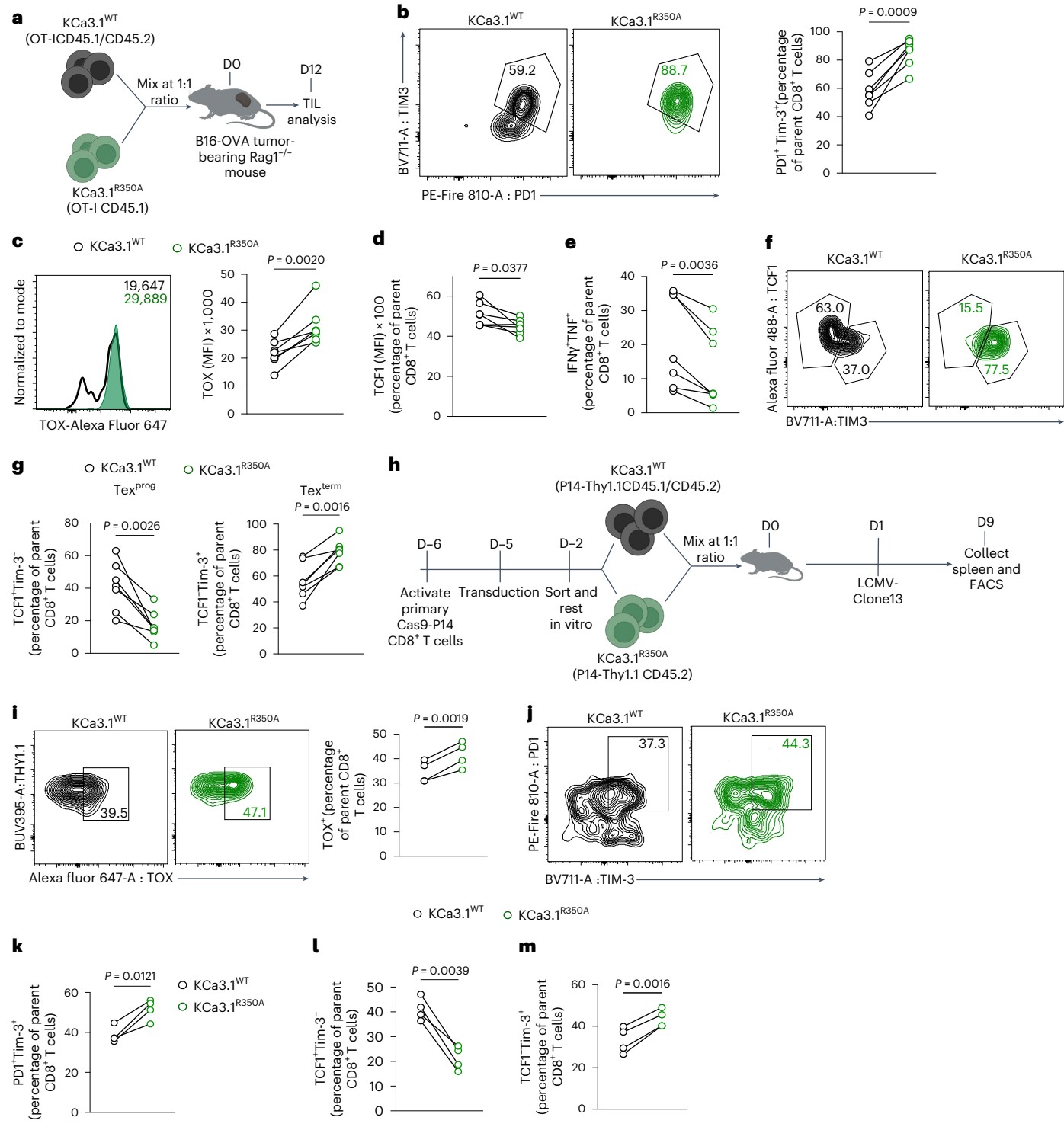

**Fig. 6 | Ablation of KCa3.1 R350 methylation promotes T cell exhaustion in tumors and infection models. a**, Schematic of experimental design to assess congenically distinct KCa3.1$^{WT}$ and KCa3.1$^{R350A}$ OT-I T cells mixed at a 1:1 ratio and transferred into B16-OVA tumor-bearing *Rag1*$^{-/-}$ mice, analyzed D12 after T cell transfer. **b–e** Representative plot and frequency of PD-1$^+$Tim-3$^+$ (**b**), histogram and quantification of TOX MFI (**c**), quantification of TCF1 MFI (**d**) and frequency of IFNγ$^+$TNF$^+$ (**e**) and among KCa3.1$^{WT}$ and KCa3.1$^{R350A}$ OT-I TIL at D12 after T cell transfer as in **a** (*n* = 7 mice per group). **f,g**, Representative plot (**f**) and frequency of TCF1$^+$Tim-3$^-$ Tex$^{prog}$ and TCF1$^-$Tim-3$^+$ Tex$^{term}$ (**g**) in KCa3.1$^{WT}$ and KCa3.1$^{R350A}$ OT-I TIL at D12 after T cell transfer as in **a** (*n* = 7 mice per group). **h**, Schematic of experimental design to assess congenically distinct KCa3.1$^{WT}$ and KCa3.1$^{R350A}$

Thy1.1 P14 T cells, mixed at a 1:1 ratio and transferred into WT mice infected with LCMV-Clone-13. Spleens were analyzed at D9 postinfection. **i**, Representative plot and frequency of TOX$^+$ in KCa3.1$^{WT}$ and KCa3.1$^{R350A}$ P14 CD8$^+$ T cells from spleens at D9 postinfection as in **g** (*n* = 4 mice per group). **j,k**, Representative plots (**j**) and frequency of PD-1$^+$Tim-3$^+$ (**k**) in KCa3.1$^{WT}$ and KCa3.1$^{R350A}$ P14 CD8$^+$ T cells from spleens at D9 postinfection as in **g** (*n* = 4 mice per group). **l,m**, Frequency of TCF1$^+$Tim-3$^-$ Tex$^{prog}$ (**l**) and TCF1$^-$Tim-3$^+$ Tex$^{term}$ (**m**) in KCa3.1$^{WT}$ and KCa3.1$^{R350A}$ P14 CD8$^+$ T cells from spleens at D9 postinfection as in **g** (*n* = 4 mice per group). Data are mean ± s.d. Paired two-tailed Student's *t*-test (**b–e**, **g**, **i–m**). Illustrations in **a** and **h** created with BioRender.com.

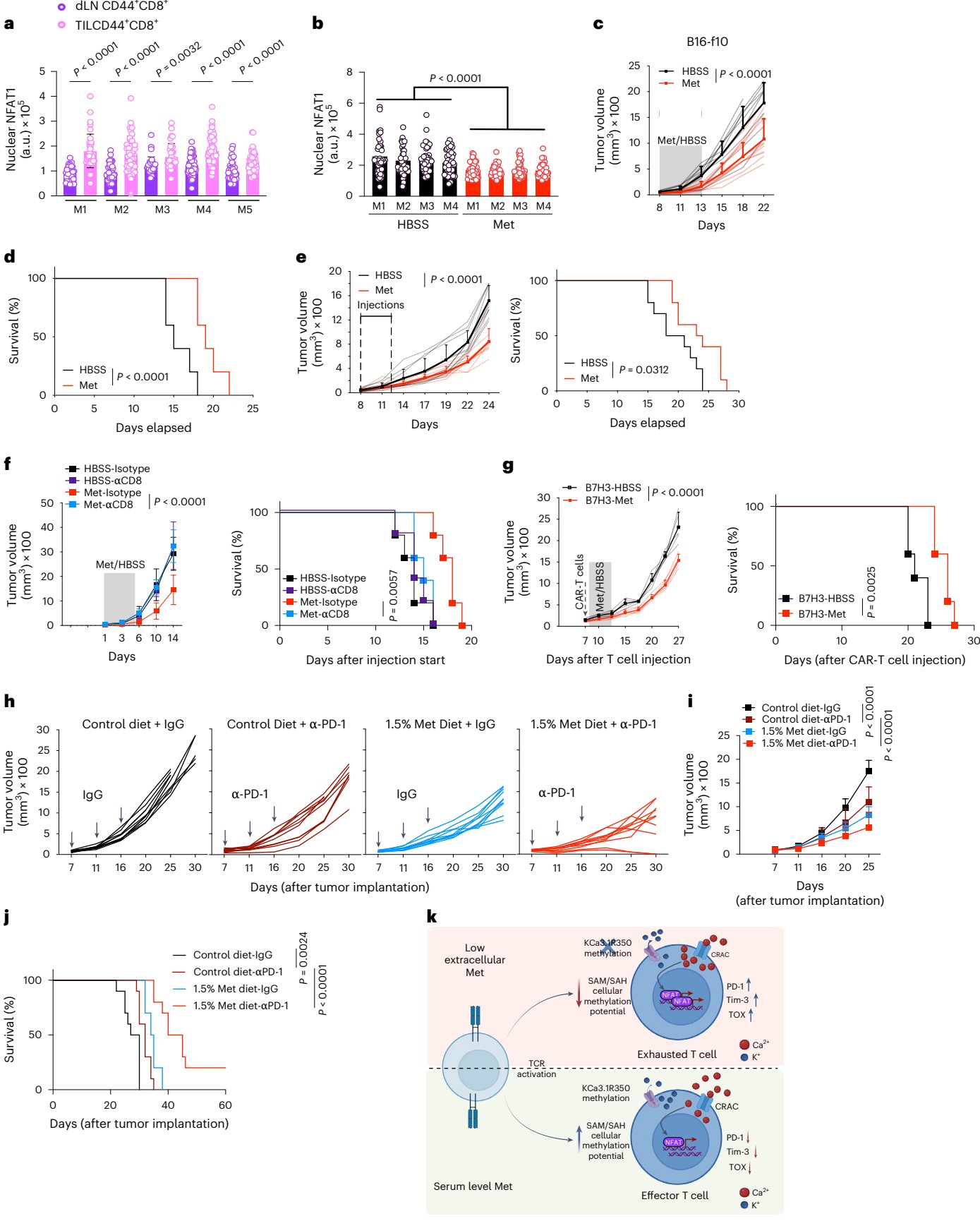

**Fig. 7 | Acute Met supplementation promotes CD8⁺ T cell-mediated tumor control and enhances ICB. a**, Nuclear NFAT1 quantification in CD44⁺CD8⁺ T cells isolated from B16 tumors and respective dLNs at D9 postimplantation (each circle represents one cell, $n$ = 40 cells, $n$ = 5 (Mouse (M)1–M5)). **b**, Quantification of nuclear NFAT1 in CD44⁺ CD8⁺ T cells isolated from B16 tumors after 5 days of peritumoral supplementation of 50 µl HBSS or 61 µM Met per day (each circle represents one cell; $n$ = 40 cells, $n$ = 4 (M1–M4)). **c–e**, Tumor growth (**c**) and survival (**d**) of B16 or MC38 (**e**) tumor-bearing WT mice, supplemented either with HBSS or 61 µM Met peritumorally for 5 days as in **b** ($n$ = 10 mice per group). **f**, B16 tumor growth (left) and survival (right) of WT mice, treated intraperitoneally with IgG isotype or anti-CD8 antibody at D1 and D3 and treated peritumorally with HBSS or Met from D7 to D12 as in **b** ($n$ = 5 mice per group, representative of two experiments). **g**, Tumor growth (left) and survival (right) of F420 tumor-bearing $Rag1^{-/-}$ mice injected with B7-H3 CAR-T cells at

D0 and peritumorally treated with HBSS or Met from D1 to D6 as in **b** ($n$ = 5 mice per group). **h–j**, Tumor growth (**h**,**i**) and survival (**j**) of MC38 tumor-bearing WT mice, treated either with anti-PD-1 or IgG isotype (arrows in **i**) and fed with either a control (1% Met) or Met-rich (1.5% Met) diet as shown in Extended Data Fig. 10h ($n$ = 10 mice per group). **k**, Diagram illustrating the impact of Met metabolism on TCR-dependent methylation of KCa3.1, leading to regulation of Ca²⁺ flux and subsequent activation of NFAT1. Low extracellular Met levels lead to decreased methylation potential, reducing KCa3.1 R350 methylation. This results in increased Ca²⁺ flux and downstream NFAT1 activation and consequent T cell hyperactivation and exhaustion. Data are mean ± s.d. Unpaired two-tailed Student's $t$-test (**a**), two-way ANOVA (**c**, **e–g**, **i**), and Mantel–Cox log rank test (**d–g**,**j**). Statistical analysis of **b** is described in Methods. Illustrations in **k** created with BioRender.com.

settings, we tested ICB with dietary Met supplementation (Extended Data Fig. 10q). Diet containing 1.5% Met (0.5% increase in dietary Met) enhanced the efficacy of anti-PD-1, leading to control of tumor growth and survival of the mice, compared to anti-PD-1 with control diet (Fig. 7h–j and Supplementary Table 3).

## Discussion

T cell exhaustion is understood to be a consequence of persistent antigen stimulation[4,44]; however, situations exist where antigen persistence does not result in exhaustion, such as in autoimmune T cells in Type I diabetes[23]. It is therefore possible that, in addition to chronic antigen exposure, nutrient limitation contributes to exhaustion[49]. A recent study suggested that T cells can follow an exhaustion trajectory within hours of antigen recognition[6]. Here, we propose that the state of T cell exhaustion is multifactorial, involving the assimilation of different input signals, including nutrients, and is engaged as early as the initial recognition of cognate antigen.

Studies have shown that strong TCR signals causes hyperactivation and activation-induced cell death, whereas weak TCR signals result in dampened responses[50]. NFAT family members are key regulators of T cell activation and interact with key partners to induce activation-associated genes including effector- and exhaustion-associated genes[29]. Continuous or partner-less NFAT activation is associated with hyperactivation, anergy and exhaustion[30,32]; therefore, NFAT activation must be controlled tightly for optimal T cell function. We found that, during rapid tumor proliferation in the early growth phase, CD8⁺ TIL display increased NFAT1 activation, which is regulated metabolically. Our data provide a new mechanism of Met metabolism in the regulation of NFAT1 activation through posttranslational modification of KCa3.1, without which CD8⁺ T cells progress toward exhaustion (Fig. 7k). We found that arginine methylation in KCa3.1 limits Ca²⁺-dependent NFAT1 signaling, ensuring optimal activation in tumors and infections. Whereas we used $Rag1^{-/-}$ immunodeficient mouse models for many of our studies to highlight the effects of Met restriction and the KCa3.1 mutant in CD8⁺ T cells, studies performed in immunocompetent mice yielded similar results to those in Rag1$^{-/-}$, suggesting the observed phenotype to be T cell intrinsic. Met supplementation to the TME preserves antitumor immunity[8] and limits exhaustion (this study), a treatment that can be harnessed to improve patient prognosis in combination with ICB therapies. Our results emphasize the importance of nonhistone protein methylation in the functional maintenance of T cells in diseases and describe a link between TCR signaling, metabolic regulation and TF activation. These results suggest that T cell exhaustion is a dynamic state controlled by several pathways over the course of T cell activation and differentiation. Our study supports the idea that metabolism-dependent, rapid proteomic modifications that occur upon TCR engagement influence subsequent T cell fate and provides mechanistic insight into how niche-specific nutrient deprivation can render T cells dysfunctional.

## Online content

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

# Methods

## Reagents

RPMI 1640 (Gibco, cat. no. 11875093) and Met-free RPMI 1640 (Gibco, cat. no. A1451701) supplemented with 10% dialyzed FCS (Atlanta Biologicals), sodium pyruvate (Gibco, cat. no. 11360070), NEAA (Gibco, cat. no. 11140050), glutamine (ThermoFisher Scientific, cat. no. 25030081) and penicillin-streptomycin cocktail (ThermoFisher Scientific, cat. no. 15070063) and stored at 4 °C. Met (cat. M5308, Sigma–Aldrich) was reconstituted in PBS (0.1 M) and stored at 4 °C. SIINFEKL peptide (cat. no. 60193, AnaSpec) peptide was reconstituted in PBS at 0.5 mg ml$^{-1}$ and stored at −20 °C. TRAM-34 (MedChem Express, cat. no. HY-13519) was reconstituted at 1 M with DMSO. CsA (Sigma–Aldrich, cat. 30024) was reconstituted to a concentration of 50 mg ml$^{-1}$ in DMSO. YM-58483 (Tocris, cat. no. 3939) was reconstituted at 20 mM stock in DMSO. All reconstituted reagents were stored at −20 °C and aliquoted to avoid repeated thawing.

## Mice

All mouse studies were conducted in accordance with protocols approved by the St. Jude Children's Research Hospital Committee on Care and Use of Animals, and in compliance with all relevant ethical guidelines. All mice were kept under a 12 h/12 h light/dark cycle in a specific-pathogen-free facility at the institute's Animal Resource Center. Age-matched and sex-matched mice were assigned randomly to the experimental and control groups. Cas9-OT/P14$^+$-I (Cas9 Knock-in with OT-I) or LCMV-GP33-specific TCR(P14)), C57BL/6J(WT), OT-I and *Rag1$^{-/-}$* mice were bred inhouse. The investigators were not blinded to the experiments and outcome assessments.

## Cell lines

B16F10, B16F10-Ova (B16-Ova) melanoma cells were provided by H. Chi. MC38 and MC38-Ova colon carcinoma cell lines were provided by D. Vignali. The F420 sarcoma cell line was provided by J. T. Yustein. Lewis lung carcinoma cells were purchased from the ATCC. Plat-e cells used for retroviral packaging were purchased from Cell Biolabs (cat. no. RV-101). All cell lines used were tested and confirmed to be mycoplasma-negative but not authenticated independently.

## Mouse and human CD8$^+$ T cell isolation and stimulation

Mouse CD8$^+$ T cells were isolated from the spleen and LN by physical disruption through a 70-µm strainer before treatment with erythrocyte-lysis buffer. Cells were subjected to CD8$^+$ T cell enrichment using the EasySep mouse CD8$^+$ T cell isolation kit (StemCell, cat. no. 19853), according to the manufacturer's protocol. Isolated T cells were then either activated with SIINFEKL peptide or mouse CD3/28 Dynabeads (ThermoFisher Scientific, cat. no. 11456D) in RPMI medium, as described above. Human CD8$^+$ T cells were isolated from apheresis rings from the St Jude Blood Donor Center. Blood obtained from an apheresis ring was mixed at a 1:1 ratio with PBS containing 2% FBS and layered carefully onto Lymphoprep (StemCell, cat. no. 07801) solution. The samples were then centrifuged at 800$g$ for 30 min at room temperature without braking. The peripheral blood mononuclear cell (PBMC) layer was aspirated and washed twice in PBS with 2% FBS before enrichment of CD8$^+$ T cells using the Human CD8$^+$ T Cell Isolation Kit (StemCell, cat. no. 17953), according to the manufacturer's protocol. CD8$^+$ T cells were activated using human CD3/28 Dynabeads (ThermoFisher Scientific, cat. no. 11131D) in RPMI medium. Apheresis rings were collected with written consent from the donors for research upon review and approval by the Institutional Review Board at St Jude Children's Research Hospital.

## Ca$^{2+}$ flux measurement and analysis

Isolated T cells were labeled with 3 µM Indo-1 (ThermoFisher Scientific, cat. no. I1203), Fluo-8 AM (AAT Bioquest, cat. no. 21080) or 5 µM ICR-1AM (Ion Biosciences) with 0.04% pluronic F-127 (ThermoFisher Scientific, cat. no. P6867) for either 1 h (Indo-1) or 2 h (Fluo-8, ICR-1) in HHBS at 37 °C and washed twice with Ca$^{2+}$-free Krebs-Ringer's solution (ThermoFisher Scientific, cat. no. J67839-AP). The cells were then resuspended in Ca$^{2+}$-free Krebs-Ringer's solution and incubated with anti-CD3ε (4 µg ml$^{-1}$; 2C11, cat. no. BE0001, Bio X Cell) and anti-CD28 (4 µg ml$^{-1}$; 37.51, cat. no. BE0015, Bio X Cell) on ice for 10 min. Then, 0.1 mM Met or 0.03 mM Met was added to the samples and the baseline was recorded before the addition of hamster IgG for crosslinking. Ca$^{2+}$ influx was measured after the addition of 2 mM Ca$^{2+}$; 5 µM ionomycin was added as a positive control for Ca$^{2+}$ flux. All measurements were recorded on Cytek 5-laser Aurora (SpectroFlo).

## Quantitative PCR

Isolated T cells were activated with 2.5 ng ml$^{-1}$ SIINFEKL in 0.1 mM Met or 0.03 mM Met for 30 min, after which Met in 0.03 mM was restored to 0.1 mM. T cells were then further cultured for 24 h. Cells were washed twice with PBS and RNA was isolated using Direct-zol RNA miniprep Kit (Zymo Research, cat. no. R2050). RNA was quantified and processed for cDNA synthesis using Reverse Transcriptase (ThermoFisher Scientific, cat. no. 28025-013). cDNA was then used to perform quantitative PCR with the specific primers (Supplementary Table 2) using SYBR green PCR master mix (ThermoFisher Scientific, cat. no. 4309155). PCR reactions were run and quantified using QuantStudio 7 flex.

## LCMV infection

LCMV Armstrong and Clone-13 was obtained from N. Shaabani, Scripps Research Institute. The virus was propagated in BHK21 cells as described[51] and stored at −80 ° C. For experiments, $1 \times 10^6$ plaque-forming units per animal of both Armstrong and Clone-13 was used. The virus was injected intravenously, and the infected animals were maintained under a 12 h/12 h light/dark cycle in a specific-pathogen-free, Biosafety Level 2 facility in the institute's Animal Resource Center.

## Cloning and virus production

*Kcnn4* gRNAs were designed using the Broad Institute platform (https://portals.broadinstitute.org/gpp/public/analysis-tools/sgrna-design) and cloned into the ametrine-expressing retroviral vector LMPd-gRNA-mPGK-Ametrine (sgRNA1:CACCGGGCAGGCTGTCAATGCCACG, AAACCGTGGCATTGACAGCCTGCCC; sgRNA2:CACCGTGTGGGGCAAGATTGTCTGC, AAACGCAGACAATCTTGCCCCACAC). KCa3.1$^{WT}$ and KCa3.1$^{R350A}$ cDNA were synthesized by Genscript; the latter contained an alanine substitution at R350. Both KCa3.1$^{WT}$ and KCa3.1$^{R350A}$ had substitutions at the PAM recognition sites of the gRNAs (Supplementary Table 3). Each cDNA was subcloned into the pMIGII. Plat-e cells were cotransfected with the cloned vectors and pCL-eco (Addgene, cat. no. 12371) in Opti-MEM (ThermoFisher Scientific, cat. no. 31985070) with Trans-IT-293 transfection reagent (Mirus Bio, cat. no. MIR 2705) and cultured at 37 °C for 12–15 h before changing the medium and culturing for an additional 24 h at 37 °C, followed by 24 h at 32 °C, 5% CO$_2$. The viral supernatant was harvested and spun at 1,800 rpm at room temperature to remove cell debris. Viral supernatant was either used immediately or stored at −80 °C until further use.

## Viral transduction

Naive Cas9-OT-I CD8$^+$ cells were isolated from the spleen and peripheral LNs of Cas9-OT-I mice using a Magnisort naive CD8$^+$ T cell isolation kit according to the manufacturer's instructions (ThermoFisher Scientific, cat. no. 8804-6825-74). Purified CD8$^+$ T cells were activated in vitro for 20 h with plate-bound anti-CD3ε (5 µg ml$^{-1}$; 2C11; Bio X Cell cat. no. BE0001) and anti-CD28 (5 µg ml$^{-1}$; 37.51, cat. no. BE0015, Bio X Cell) antibodies. Viral transduction was performed by spinfection at 900$g$ at 25 °C for 3 h with 10 µg ml$^{-1}$ polybrene (Sigma–Aldrich) followed by resting for 3 h at 37 °C in 5% CO$_2$. Cells were washed and cultured in medium supplemented with mouse rIL-2 (20 U ml$^{-1}$; Peprotech,

cat. no. 212-12) for 4 days. GFP-Ametrine double-positive cells were sorted using the Reflection cell sorter (iCyt) or the MoFlo XDP cell sorter and rested for 24 h in medium containing rIL-2 (10 U ml⁻¹; Peprotech, cat. no. 212-12) before activation or adoptive transfer into the animals.

## Flow cytometry

Antibodies from BioLegend included Pacific Blue anti-mouse Ly108 (330-AJ, 134608), BV570 anti-mouse CD62L (MEL-14, 104433), BV711 anti-mouse Tim-3 (RMT3-23, 119727), BV785 anti-mouse CD127 (A7R34, 135037), PE-Cy5 anti-mouse Granzyme-B (QA16A02, 372226), PE-Fire 700, anti-mouse CD4 (GK1.5, 100484), APC/Cy7 anti-mouse TNF (MP6-XT22, 506344), PerCP-Cy 5.5 anti-mouse CD62L (MEL-14, 104432), BV605 anti-mouse CD127 (A7R34, 135025), Pacific blue anti-mouse CD69 (H1.2F3, 104524), Pacific blue anti-mouse CD45.1 (110722), APC anti-mouse PD-1 (RL388, 109111), BV711 anti-mouse PD-1 (29F.1A12, 135231), PE anti-mouse IFNγ (XMG1.2, 505808), FITC anti-mouse TNF (MP6-XT22, 506304), Pacific Blue anti-human CD45 (HI30, 982306) and APC/Cy7 anti-human CD8 (RPA-T8, 344713). BUV615 anti-mouse CD69 (H1.2F3, 751593), BV480 anti-mouse CD45.1 (A20, 746666), Alexa Fluor 488 anti-mouse TCF1 (S33-966, 567018), Alexa Fluor 647 TOX (NAN448B, 568356), BUV 496 anti-mouse Ly108 (13G3, 750046), APC/Cy7 anti-mouse CD44 (IM7, 560568) and BUV805 anti-mouse CD8α (53-6.7, 612898) were acquired from BD Biosciences. BUV395 anti-mouse CD44 (IM7, 363-0441-82), PerCP-Cy 5.5 anti-mouse IL-2 (JES6-5H4, 45-7021-82), PerCP-eF710 anti-mouse CD27 (O323, 46-0279-42) and PE-Cy7 anti-mouse Tim-3 (RMT3-23, 12-5870-82, eBioscience) were obtained from ThermoFisher Scientific. APC anti-human/mouse Tox (REA473, 130-118-335) was obtained from Miltenyi Biotech. All antibodies were used in 1:200 dilution. For surface markers, the cells were stained for 30 min on ice in PBS containing 2% FCS and 0.1% sodium azide (Sigma–Aldrich, cat. no. S2002). To examine intracellular cytokine production, the isolated cells were stimulated with a cell activation cocktail (TOCRIS, cat. no. 5476) in the presence of monensin (BD Biosciences, cat. no. 554724) for 4 h, and stained for intracellular cytokines and TFs using a fixation/permeabilization kit (Ebiosciences, cat. no. 00-5523-00). Fixable viability dye (ThermoFisher Scientific) was used to exclude dead cells. Samples were acquired on a Cytek Aurora (SpectroFlo) or BD LSRII (FACSDiva) flow cytometer and analyzed using FlowJo software.

## In vivo tumor analysis

For all tumor experiments, $0.5 \times 10^6$ tumor cells were injected subcutaneously into the right flank of naive mice (aged 6–10 weeks). Met was diluted in HBSS to a concentration of 61 μM per 50 μl. Mice were injected peritumorally for 5 days with either 50 μl Met or HBSS. Tumors were measured with calipers every 2 days and tumor volumes were calculated using the formula: (width² × length)/2. For adoptive transfer experiments, T cells ($1 \times 10^6$) were transferred intravenously 7–9 days after tumor injection. For the contralateral tumor model, $0.5 \times 10^6$ B16 tumor cells were injected subcutaneously into the right flank and left shoulder of the same mouse. Met or HBSS was injected peritumorally to the right flank tumor (abdominal). For the checkpoint blockade experiment, $0.5 \times 10^6$ MC38 tumor cells were injected subcutaneously into the right flank of WT mice. The mice were given either anti-PD-1 antibody (RMP1-14, Bio X Cell) or rat IgG2a (2A3, Bio X Cell), intraperitoneally twice at a dose of 150 mg kg⁻¹ in 100 μl PBS at days 9 and 12. Met or HBSS was given peritumorally for 5 days from day 9 to day 13. For the CAR-T tumor model, $0.5 \times 10^6$ F420 tumor cells (derived from singly floxed *p53+/F-Col2.3* transgenic mice[52]) were injected subcutaneously into the right flank of *Rag1⁻/⁻* mice. Thereafter, $5 \times 10^6$ B7-H3 CAR-T cells (see below) were transferred intravenously 8–10 days postimplantation. Control (1% Met; Research Diets, cat. no. A24051402i) and Met-rich (1.5% Met; Research Diets, cat. no. A24051402i) (Supplementary Table 4) irradiated diets were obtained from Research Diets and maintained according to manufacturer's instructions. Sample size was chosen

based on the preliminary data. Mice were assigned randomly to different groups after tumor inoculation and tumor size was measured every 2–4 days. Investigators were not blinded to the tumor analysis. Animals were euthanized when tumor measurements exceeded ~2,000 mm³ as approved by the Institutional Animal Care and Use Committee.

## Generation of mouse CAR-T cells

Generation of mouse B7-H3-CAR constructs and CAR-T generation have been described previously[53]. Briefly, HEK293T cells were transfected transiently with the CAR-encoding plasmid, the Peg–Pam plasmid encoding MoMLV GAG–POL, and a plasmid encoding the VSVG envelope. Viral supernatants were collected at 48 h and filtered with a 0.45 m filter. The virus was then used to transduce the GPE86 producer cell line. B7-H3-CAR-expressing GPE86 cells were stained with Alexa Fluor 647 anti-human IgG, F(ab')2 fragment antibody (Jackson ImmunoResearch, cat. no. 109-606-006) and sorted using the BD FACSAria III system.

Naïve CD8⁺ T cells from 6- to 8-week old CD45.1⁺ mice were activated with plate-bound anti-CD3ε (1 μg ml⁻¹, 145-2C11; BioXcell, cat. no. BE0001), anti-CD28 (2 μg ml⁻¹, 37.51; Bio X cell, cat. BE0015) and 50 U ml⁻¹ rhIL-2 (Peprotech, cat. no. 200-02). On day 2 postactivation, activated CD8⁺ T cells were transduced with retrovirus expressing B7-H3-CAR on retronectin-coated (Takara, cat. no. T100B) nontissue-culture-treated plate in complete RPMI medium supplemented with 50 U ml⁻¹ rhIL-2. At 48 h after transduction, CAR-T cells were collected and expanded in the presence of 50 U ml⁻¹ rhIL-2 for another 3 days before adoptive transfer into tumor-bearing mice.

## TIL isolation

B16 tumors were harvested at the timepoints indicated and digested using a digestion cocktail (0.1% Collagenase IV and 0.01% DNase I) for 5 min at 37 °C. Digested tumors were passed through a 70 μm strainer and resuspended in PBS plus 2% dialyzed FCS and processed for analysis and/or sorting by flow cytometry, as described above.

Patient samples were collected following surgery and were processed immediately. Tumor samples were dissected into small pieces and digested at 37 °C with digestion cocktail for 40 min. Digested tumor samples were physically disrupted and passed through 70 μm filter. Samples were washed with PBS-2% dialyzed FCS twice and stained for CD45, CD3ε and CD8 and sorted for CD8⁺ T cells by flow cytometry. Purified T cells were washed once with 1% saline and flash-frozen in liquid nitrogen. Samples were collected upon informed consent and procedures approved by the IRB of St. Jude Children's Research Hospital.

## Immunofluorescence staining and imaging

CD8⁺ T cells were immobilized in eight-well poly L-lysine-coated IBIDI chambers and stimulated by the addition of CD3/28 Dynabeads (ThermoFisher Scientific, cat. no.11452D) for 30 min, followed by 10 min of fixation with 4% paraformaldehyde (Electron Microscopy Science, cat. no.15710) at 37 °C. The cells were washed once with TBS (50 mM Tris, 100 mM NaCl, pH 8.0) and permeabilized with permeabilization buffer (50 mM Tris, 100 mM NaCl, pH 8.0, 0.3% (v/v) Triton X-100) for 20 min at room temperature. Cells were washed again with TBS before blocking with TBS containing 2% BSA (Jackson ImmunoResearch, cat. no. 001-000-182) for 60 min at room temperature. The cells were stained overnight at 4 °C with the following primary antibodies: anti-NFAT1 (1:200; Cell Signaling Technology, cat. no. 4389), anti-NFAT2 (1:200; Cell Signaling Technology, cat. no. D15F1) and antimono and dimethyl-arginine (1:500; Abcam, cat. no. ab412). The samples were washed twice with TBS and incubated for 1 h at room temperature with the following antibodies: anti-rabbit Alexa Fluor plus 595 (1:1,000; ThermoFisher Scientific, cat. no. A-11012), anti-mouse CD8-APC (1:500; BioLegend; cat. no. 100712), Alexa Fluor 488 Phalloidin (1:1,000; ThermoFisher Scientific, cat. no. A12379) and Hoechst 33258 (1:1,000; ThermoFisher Scientific, cat. no. H3569). Samples were then imaged by Marianas spinning-disc confocal microscopy (Intelligent Imaging Innovations),

using the Prime95B sCMOS camera, and 405, 488, 561 and 640 nm laser lines. Sum fluorescence intensities were analyzed using Slide Book v.6 (Intelligent Imaging Innovations).

## Immunoblotting

T cells were washed once with ice-cold PBS and immediately lysed with 4× Laemmli buffer (Bio-Rad, cat. no. 1610747). Cell lysates were separated by SDS–PAGE and blotted with anti-Kcnn4 antibody (ThermoFisher Scientific, cat. no. PA5-33875) and horseradish-peroxidase-conjugated anti-Actin (Santa Cruz Biotechnology, cat. sc4777-HRP). Images were developed using Clarity Western ECL substrate (Bio-Rad, cat. no. 1705061) and acquired on Bio-Rad Chemi-Doc.

## Amino acid measurement

Mouse subcutaneous tumors and human tumor samples were isolated and processed as described above. Sorted T cells were washed once with ice-cold saline, flash-frozen with liquid nitrogen, and stored at −80 °C before analysis. Cell pellets were extracted using 750 μl of methanol/acetonitrile/water (5/3/2, v/v/v) and the supernatant was dried by lyophilization. Aliquots of 20–50 μl from plasma and tumor interstitial fluid were extracted with at least a 15-fold excess volume of the methanol/acetonitrile/water solution, and the supernatant was then collected and dried by lyophilization. Dried extracts containing the hydrophilic metabolites were dissolved in 30 μl of water/acetonitrile (8/2, v/v), and 10 μl was used to derivatize amino acids as described[54] with some modifications. The samples were placed in glass autosampler vials, and 35 μl of sodium borate buffer (100 mM, pH 9.0) was added and mixed by pipetting. Then, 10 μl of the 6-aminoquinolyl-N-hydroxysuccinimidyl carbamate (AQC, 10 mM in acetonitrile)-derivatizing reagent (Cayman Chemical) was added, the vial was sealed, mixed by vortexing and incubated at 55 °C for 15 min. The vial was cooled to room temperature and 1 μl was then analyzed by liquid chromatography with tandem mass spectrometry (LC–MS/MS). An ACQUITY Premier UPLC System (Waters Corp.) was used for the LC separations, using a nonlinear gradient condition as follows: 0–0.4 min 3% B; 0.4–8 min 3 to 96% B (using Curve 8 of the inlet condition in MassLynx); 8–12 min 96% B; 12–12.5 min 96 to 3% B; 12.5–14 min 3% B. Mobile phase A was water supplemented with 0.15% acetic acid, and mobile phase B was acetonitrile with 0.15% acetic acid. The column used was an Accucore C30 (50 × 2.1 mm, 2.6 μm) (ThermoFisher Scientific) operated at 50 °C. The flow rate was 300 μl min$^{-1}$ and the injection volume used was 1 μl. All LC/MS solvents and reagents were the highest purity available (water, acetonitrile, acetic acid, boric acid, sodium hydroxide) and were purchased from ThermoFisher Scientific. A Xevo TQ-XS Triple Quadrupole Mass Spectrometry (TQ-XS) (Waters Corp.) equipped with a multimode ESI/APCI/ESCi ion source was employed as detector. The TQ-XS was operated in positive ion mode using a multiple reaction monitoring mass spectroscopy method (MRM). The MRM conditions were set to a minimum of 15 points per peak with an automatic dwell time. The operating conditions of the source were: capillary voltage 3.8 kV, cone voltage 40 V, desolvation temperature 550 °C, desolvation gas flow 1,000 l h$^{-1}$, cone gas flow 150 l h$^{-1}$, nebulizer 7.0 bar and source temperature 150 °C. Authentic amino acid standards were purchased from Sigma–Aldrich and used to establish MRM conditions and calibration curves. The monitored parent/daughter ions, fragmentation collision energy (CE) and retention time window for each amino acid are listed in Supplementary Table 5. The MRM data acquired were processed using the software application Skyline v.21.2 (MacCoss Lab Software). One patient sample was determined to be a statistical outlier (Grubbs' Test) and was excluded from the graph.

## In silico model building of KCa3.1(SK4)–CaM complex and simulation

To study the impact of SK4s arginine dimethylation on the binding of CaM, we simulated the SK4 channel:CaM complex in a membrane layer by means of all-atom MD, which is considered a computational microscope[55]. The cryo-electron microscopy (EM) structure of the human SK4 channel:CaM complex state (PDB 6CNN) was used as the starting structure[56]. This tetramer complex consisted of four SK4 monomers, four CaMs, three calcium ions bound to each CaM (total one, two) and four K$^+$ ions bound to the channel selectivity filter.

Initially, protein was prepared was using the molecular operating environment software. The missing residues in the cryo-EM structure were modeled based on the human AlphaFold structure (AF-O15554-F1). Protonate3D module in molecular operating environment was used to assign appropriate protonation states at pH 7.4 and then energy minimized. Further, simulation system was prepared using CHARMM-GUI web server[57–59]. Protein was placed in the POPE membrane bilayer and protein orientation was obtained using the PPM web server[60]. The protein–membrane bilayer complex was immersed in the TIP3P water box[61] of size ~150 × 150 × 230 Å$^3$ with an edge distance of 40 Å on all sides. KCl salt was added to balance the charges and maintain a physiological salt concentration of 0.15 M. Overall, each wild-type simulation system contains four SK4 monomers, four CaMs, 12 Ca$^{2+}$ ions, ~355 K$^+$ ions, ~367 Cl$^-$ ions, ~618 POPE lipids and ~128,144 TIP3P waters. The total number of atoms in this system is ~498,944. Two more systems were generated using a protocol similar to that explained above, except that, in one system, R352 of the SK4 monomer was dimethylated symmetrically and in the second system, R352 was dimethylated asymmetrically. Dimethylation was introduced using the CHARMM-GUI web server. Note that arginines at position 352 in all four SK4 monomers were dimethylated in both systems.

Further, all three systems were simulated using AMBER20/22 MD software[62]. The MD simulation was performed in three steps. In step 1, energy minimization was carried out to remove potential steric clashes in two stages. In stage 1, each system was energy minimized for 5,000 steps, while positional restraints were applied to the protein and membrane: the first 2,500 steps with the steepest-descent algorithm and the remaining 2,500 steps using the conjugate gradient algorithm[63]. In stage 2, the second round of minimization was carried out with all restraints removed in 50,000 steps. The steepest-descent algorithm[64] was used for the first 25,000 steps, whereas the conjugate gradient algorithm was used for the remaining 25,000 steps. In step 2, equilibration was carried in six stages: (1) equilibrated for 125 ps with a timestep of 1 fs while applying positional restraints on protein ($K$ = 10 kcal mol Å$^{-2}$) and membrane ($K$ = 2.5 kcal mol Å$^{-2}$), (2) equilibrated for 125 ps with a timestep of 1 fs while applying positional restraints on the protein ($K$ = 5.0 kcal mol Å$^{-2}$) and membrane ($K$ = 2.5 kcal mol Å$^{-2}$), (3) equilibrated for 125 ps with a timestep of 1 fs while applying positional restraints on the protein ($K$ = 2.5 kcal mol Å$^{-2}$) and membrane ($K$ = 1.0 kcal mol Å$^{-2}$), (4) equilibrated for 500 ps with a timestep of 2 fs while applying positional restraints on the protein ($K$ = 1.0 kcal mol Å$^{-2}$) and membrane ($K$ = 0.5 kcal mol Å$^{-2}$), (5) equilibrated for 500 ps with a timestep of 2 fs while applying positional restraints on the protein ($K$ = 0.5 kcal mol Å$^{-2}$) and membrane ($K$ = 0.1 kcal mol Å$^{-2}$) and (6) equilibrated for 500 ps with a timestep of 2 fs while applying weak positional restraints on the protein ($K$ = 0.5 kcal mol Å$^{-2}$) and restraints on the rest of the system were removed completely. The first two equilibration simulations were performed under NVT conditions, whereas the last four equilibration simulations were performed under NPT conditions. In step 3, production simulations were carried out under periodic boundary conditions using the NPT ensemble, during which the temperature (310 K) and pressure (1 atm) were kept constant. Three replicates of each system were simulated, and each replicate was simulated for ≈1 μs. Langevin integrator and Monte Carlo barostat were used[65,66] to conduct NPT simulations, whereas the SHAKE algorithm was used to restrain bonds with hydrogens[67]. The particle-mesh Ewald method was used to treat long-range electrostatic interactions[68] and the nonbonded cutoff was 12 Å (force switching = 10 Å). CHARM36

all-atom additive forcefield[69,70] was used to treat the entire system. Overall, nine MD simulations were conducted across three systems, with each system comprising three replicates that each ran for approximately 1 μs. Consequently, the total MD simulation data accumulated for analysis amounts to approximately 9 μs, derived from the sum of simulation times for all systems and replicates (3 systems × 3 replicates × ~1 μs per replicate = ~9 μs). Two frames per nanosecond were used for the data analysis. Visual molecular dynamics[71] was used for visualization and graphic generation, whereas Tcl was used for data analysis. Gnuplot was used to generate the timeseries plots.

## ATAC-seq

OT-I CD8$^+$ T cells were activated with 2.5 ng ml$^{-1}$ SIINFEKL in either 0.1 mM Met or 0.03 mM Met with restoration of Met to 0.1 mM in 0.03 mM condition 30 min postactivation and incubated for 24 h. KCa3.1$^{WT}$ and KCa3.1$^{R350A}$ cells were activated by 2.5 ng ml$^{-1}$ SIINFEKL for 24 h. Cells were processed for ATAC-seq after 24 h of activation. Briefly, 50,000 cells were washed with 1 ml cold PBS by centrifugation at 500$g$ for 5 min at 4 °C. Cells were then resuspended in 50 μl cold cell lysis buffer (10 mM Tris pH 7.4, 10 mM NaCl, 3 mM MgCl$_2$ and 0.1% NP-40). Isolated nuclei were pelleted at 1,000$g$ for 10 min at 4 °C, resuspended in 50 μl transposition reaction mixture (25 μl TD buffer, 2.5 μl TDE1, 22.5 μl ddH$_2$O) (Illumina, cat. no. 20034197) and incubated at 37 °C for 30 min. DNA was purified using a MiniElute PCR purification kit (Qiagen, cat. no. 28004). Samples were then processed for library barcoding and amplification with Q5 High-Fidelity 2× Master Mix (NEB, cat. no. M0492S). Prepared libraries were sent for sequencing after quantification using Qubit and size distribution as determined using an Agilent 4200 TapeStation.

## Analysis of ATAC-seq data

Paired-end sequencing reads were trimmed using Trim Galore (v.0.5.0) (https://github.com/FelixKrueger/TrimGalore) with default parameters. Reads were aligned to the reference mouse mm10 assembly using Bowtie 2 (v.2.3.5.1)[72] with settings –end-to-end –very-sensitive -X 2000. The resulting alignments, recorded in a BAM file, were sorted, indexed and marked for duplicates with Picard Mark-Duplicates function (v.2.19.0). Afterward, the BAM file was filtered with SAMtools (v.1.9)[73], BamTools (v.2.5.1)[74] and scripts of nf-core/chipseq[75] to discard reads, mates that were unmapped, PCR/optical duplicates, not primary alignments, mapped to several locations or mapped to ENCODE-blacklisted regions[76]; only reads mapped in proper pairs were retained (-F 1804 -f 2 -q 30). The alignments were shifted by deepTools alignmentSieve[77] with –ATACshift and default parameters. Nucleosome-free reads (fragment length <109 bp) were separated from the BAM files with SAMtools (v.1.9). MACS (v.2.1.2)[78] was used to call peaks from the BAM files with narrowPeak setting, –extsize 200 and recommended mappable genome size (default value for other parameters). Chromatin accessibility signal was normalized by scaling to 1 million mapped reads using BEDTools (v.2.27.1)[79] and bedGraphToBigWig (v.377)[80] and visualized as heatmaps using deepTools plotHeatmap (v.3.2.1)[77].

Differentially accessibility analysis was performed using DiffBind[81] (v.3.4.7) (summit = 50, normalized by the trimmed mean of M values[82] approach, library size estimated by reads mapped to 'background' regions, the genomic regions (binned into 15,000 bp) overlapped with peaks). Peaks were annotated to the nearest genes with annotatePeak function in R package ChIPseeker (v.1.30.3) at the gene level using the default parameters[83]. Genes were ranked by fold change (FC) (log$_2$ transformed) of associated peaks located within ±1.5 kb of transcription start sites (if a gene was associated with several peaks, the mean was used). GSEA[84] was then performed against MSigDB collections with R package clusterProfiler (v.4.2.2)[85]. The common differentially accessible regions (DAR) between ATAC-Seq of KCa3.1WT and KCa3.1R350A and T cells activated in 0.1 mM or 0.03 mM Met were obtained by the overlapping of the DARs (at least 1 bp). Enrichment analysis was performed using the over-representation approach against MSigDB using the R package clusterProfiler.

## CUT&RUN

CUT&RUN experiments were performed as described previously[86] with slight modifications. Purified OT-I CD8$^+$ T cells were activated with 2.5 ng ml$^{-1}$ SIINFEKL in 0.1 mM and 0.03 mM Met for 30 min. Met was restored to 0.1 mM after 30 min and cultured for 24 h. Cells were washed with cold PBS and dead cells were removed using Easysep dead cell removal kit (StemCell, cat. no. 17899). Cells ($2 \times 10^5$) were washed twice with wash buffer (20 mM HEPES (Sigma–Aldrich, cat. no. H3375), 150 mM NaCl (Invitrogen, cat. no. AM9760G), 0.5 mM spermidine (Sigma–Aldrich, cat. no. S0266) and protease inhibitor cocktail (Sigma–Aldrich, cat. no. 5056489001), resuspended and bound to concanavalin-A coated magnetic beads (Bang Laboratories, cat. no. BP531). The resuspension buffer was removed using a magnetic stand and beads were resuspended in 100 μl antibody buffer (20 mM HEPES, 150 mM NaCl, 0.5 mM spermidine, 0.01% digitonin (Millipore, cat. no. 300410), 2 mM EDTA (Invitrogen, cat. no. AM9260G) and protease inhibitor cocktail). Primary antibodies (1:100) were added to the samples and incubated overnight at 4 °C. Next, samples were washed twice with cold Dig-wash buffer (20 mM HEPES, 150 mM NaCl, 0.5 mM spermidine, 0.01% digitonin and protease inhibitor) and pAG-MNase (Addgene, cat. no. 123461) was added and rotated at 4 °C for 1 h. Samples were washed twice and resuspended in 50 μl Dig-wash buffer and CaCl$_2$ (2 μl of 100 mM; Sigma–Aldrich, cat. no. 2115) was added followed by brief vortexing and incubation on ice for 30 min. Next, 50 μl of 2× STOP buffer (340 mM NaCl, 20 mM EDTA, 4 mM EGTA (cat. E3889, Sigma–Aldrich) and 100 μg ml$^{-1}$ RNase A (ThermoFisher Scientific, cat. no. EN0531) and 50 μg ml$^{-1}$ GlycoBlue (Invitrogen, cat. no. AM9515) was added and mixed by gentle vortexing and incubated for 30 min at 37 °C to release CUT&RUN fragments. Fragmented DNA was purified using the NEB Monarch PCR&DNA Cleanup Kit (New England Biolabs, cat. no. T1030S). DNA libraries were prepared using the NEBNext Ultra II DNA Library Prep Kit (New England Biolabs, cat. no. E7645S) and purified with AMPure SPRI beads (Beckman-Coulter, cat. no. B23318). Prepared libraries were quantified using Qubit and size distribution was determined with a Agilent 4200 TapeStation analysis before paired-end sequencing.

## CUT&RUN data processing

Paired-end sequencing reads were trimmed using Trim Galore (v.0.5.0) (https://github.com/FelixKrueger/TrimGalore) with default parameters. The reads were then aligned to the reference mouse mm10 assembly using Bowtie 2 (v.2.3.5.1)[72] with settings –end-to-end –very-sensitive –no-mixed –no-discordant -q –phred33 -I 10 -X 700. The resulting alignments, recorded in the BAM file, were sorted, indexed and marked for duplicates using the Picard MarkDuplicates function (v.2.19.0) (Picard toolkit (https://broadinstitute.github.io/picard/) Broad Institute GitHub Repository, 2019). Next, the BAM file was filtered with SAMtools (v.1.9)[73], BamTools (v.2.5.1)[74] and scripts of nf-core/chipseq[75] to discard reads, including mates that were unmapped, PCR/optical duplicates, nonprimary alignments, reads that mapped to several locations or to ENCODE-blacklisted regions[76], or with more than four mismatches (-F 0×004 -F 0×008 -F 0×0100 -F 0×0400 -f 0×001 -q1). MACS (v.2.1.2)[78] was used to call peaks from the BAM file with IgG control and recommended mappable genome size (the default values were used for the other parameters). NarrowPeak mode was used for NFAT1, H3K27me3 and H3K4me3. Binding signal was normalized by scaling to 1 million mapped reads using BEDTools (v.2.27.1)[79] and bedGraphToBigWig (v.377)[80] and visualized as heatmaps using deepTools plotHeatmap (v.3.2.1)[77].

In the CUT&RUN experiment with spike-in *Escherichia coli*, two modifications were made: (1) a hybrid reference of mouse mm10 and

*E. coli* ASM584v2 and (2) signals were normalized by scaling to per million reads mapped to *E. coli*.

Differential binding analysis was performed using DiffBind (v.3.4.7) (summit = 75 (for NFAT1) and 100 (for H3K27me3 and K3K4me3 modifications), normalized by the trimmed mean of M values approach library size estimated by reads mapped to 'background' regions. Peaks were annotated to the nearest genes with the annotatePeak function in R package ChIPseeker (v.1.30.3) at the gene level using default parameters. Genes were ranked by FC ($\log_2$ transformed; if a gene was associated with several peaks, the mean was used). GSEA was then performed against MSigDB collections with R package clusterProfiler (v.4.2.2).

## RNA sequencing
The isolated CD8$^+$ TIL were washed once with cold PBS and pelleted at 1,500 rpm at 4 °C. Total RNA was isolated using the Direct-zol RNA Microprep Kit (Zymo Research, cat. no. R2061) according to the manufacturer's instructions and quantified using an Agilent 4200 TapeStation. Libraries were prepared using the KAPA RNA HyperPrep Kit with RiboErase (HMR) (Roche, cat. no. 08098131702) and purified by AMPure SPRI beads (Beckman-Coulter, cat. no. B23318). Libraries were quantified and size distribution was determined using a Agilent 4200 TapeStation before paired-end sequencing was performed.

## RNA-seq data processing
Paired-end sequencing reads were mapped by the pipeline of St. Jude Center for Applied Bioinformatics. The reads were trimmed using Trim Galore (v.0.5.0) (https://github.com/FelixKrueger/TrimGalore) with default parameters. Reads were aligned to the reference mouse mm10 assembly plus ERCC spike-in sequences using STAR (v.2.7.5a)[87]. The resulting alignments, recorded in BAM file, were sorted, indexed and marked for duplicates with Picard MarkDuplicates function (v.2.19.0). Transcript quantification was calculated using RSEM[88]. Differential gene expression analysis was carried out with DESeq2 with Wald test (default parameters)[89]. To examine whether global changes in gene expression were present, the RUVg function in the RUVSeq package was used for normalization with ERCC spike-in followed by DESeq2 according to the RUVSeq manual[90]. GSEA was performed against the MSigDB database[91] with R package clusterProfiler (v.4.2.2)[85] with the genes ranked by the Wald statistic from DESeq2 analysis.

## Quantitative proteomics by TMT-MS
CD8$^+$ T cells were isolated from WT mice and activated with anti-CD3 antibody by crosslinking with IgG for 30 min. Activated and untreated cells were washed once with ice-cold PBS and flash-frozen in liquid nitrogen. Proteomic profiling of the whole proteome and methylome was carried out using a previously optimized protocol[92] with modifications. Briefly, the cells were lysed in 8 M urea lysis buffer using pulse sonication, and approximately 100 µg of protein per sample was digested with Lys-C (Wako, 1:100 w/w) at 21 °C for 2 h, followed by dilution to reduce urea to 2 M, and further digestion with trypsin (Promega, 1:50 w/w) at 21 °C overnight. The protein digests were acidified (trifluoroacetic acid to 1%), centrifuged at 21,000*g* for 10 min at 4 °C to remove any insoluble material, desalted with Sep-Pak C18 cartridge (Waters), and dried by Speedvac. Each sample was resuspended in 50 mM HEPES (pH 8.5), TMT labeled, mixed equally, desalted and fractionated by offline HPLC (Agilent 1220) using basic-pH reverse phase LC (Waters XBridge C18 column, 3.5 µm particle size, 4.6 mm × 25 cm, 180 min gradient, 80 fractions). Fractions were further concatenated into ten fractions for antibody-based sequential monomethylarginine and dimethylarginine peptide enrichment.

## Antibody-based sequential methylome enrichment
Each concatenated fraction (approximately 1 mg) was resuspended in 400 µl ice-cold IAP buffer (50 mM MOPS, pH 7.2, 10 mM sodium phosphate and 50 mM NaCl) and centrifuged at 21,000*g* for 10 min at 4 °C to remove any insoluble material. The peptides were then incubated with monomethylarginine antibody beads (mono-me-R; Cell Signaling Technology, cat. no. 12235S) at an antibody-to-peptide ratio of 1:20 (w/w, optimized through a pilot experiment) for 2 h at 4 °C with gentle end-over-end rotation. The antibody beads were then collected by a brief centrifugation, washed three times with 1 ml ice-cold IAP buffer and twice with 1 ml ice-cold PBS, while the supernatants were removed carefully and used for the sequential dimethylarginine peptide enrichment (di-me-R symmetric; Cell Signaling Technology, cat. no. 13563S). Peptides were eluted from the beads twice at room temperature with 50 µl of 0.15% TFA, dried, and analyzed by LC–MS/MS.

## LC–MS/MS analysis
Each sample was dried, reconstituted and analyzed by LC–MS/MS (a CoAnn 75 µm × 30 cm column, packed with 1.9 µm C18 resin from Dr. Maisch GmbH) interfaced with the Orbitrap Fusion MS (ThermoFisher Scientific). LC settings included an 80 min gradient of 15%–40% buffer B (70% acetonitrile, 2.5% DMSO, and 0.1% formic acid) with buffer A (2.5% DMSO, and 0.1% formic acid) at a flow rate of ~0.25 µl min$^{-1}$. MS settings included data-dependent (3-s cycle) mode with a survey scan in the Orbitrap (60,000 resolution, scan range 410–1,600 m/z, 1 × 106 AGC target and 50 ms maximal ion time), followed by sequential isolation of abundant ions in a 3-s duty cycle, with fragmentation by higher-energy collisional dissociation (38 normalized CE), and high-resolution detection of MS/MS ions in the Orbitrap (60,000 resolution, 1 × 105 AGC target, 105 ms maximal ion time, 1.0 m/z isolation window and 20 s dynamic exclusion).

## Database searches and TMT quantification
The MS/MS spectra were searched against the UniProt Mouse protein database (v.2020.04.22) using the COMET algorithm (v.2018.013) with the JUMP software suite[93,94]. Search parameters included MS1 mass tolerance of 20 ppm and MS/MS of 0.02 Da, fully tryptic, static mass shift for the TMT16 tags (+304.2071453) and carbamidomethyl modification of 57.02146 on cysteine, dynamic mass shift for Met oxidation (+15.99491), for monomethylarginine (+14.01565), and for dimethylarginine (+28.0313), maximal missed cleavage ($n$ = 3) and maximal dynamic modifications per peptide ($n$ = 5). All matched MS/MS spectra were filtered by mass accuracy and matching scores to reduce false discovery rate below 1%, based on the target-decoy strategy[95,96].

TMT quantification analysis was performed as previously reported[97] with the following modifications: (1) extracting reporter ion intensities from each peptide spectrum match (PSM); (2) correcting the intensities according to the isotopic distribution of each TMT reagent; (3) removing PSMs of very low intensities (for example, minimum value of 1,000 and median value of 5,000); (4) normalizing sample loading bias with the trimmed median intensity of all PSMs; (5) calculating the mean-centered intensities across samples (for example, relative intensities between each sample and the mean), (6) summarizing protein relative intensities by averaging related PSMs and (7) finally deriving protein absolute intensities by multiplying the relative intensities by the grand-mean of the three most highly abundant PSMs. Protein FC and *P* values of different comparisons were calculated based on the protein intensities, using a log transformation and moderated *t*-test with the limma R package[98,99].

## Quantification and statistical analysis
Data were plotted and analyzed using GraphPad Prism (GraphPad Software, v.9.2.0). Statistical significance was calculated using unpaired one- or two-tailed Student's *t*-tests. Mixed-model two-way ANOVA was performed to compare tumor growth curves. The log rank (Mantel–Cox) test was performed to compare the mouse survival curves. Data distribution as assumed to be normal but this was not formally tested. For Fig. 7b, a linear mixed effects model with $\log_{10}(y)$ as the response variable, group as a fixed-effect predictor variable and mouse as a

random-effect predictor variable was fit to the data. This model represents data with cells as random samples from each mouse. The results showed that the mean $\log_{10}(y)$ values of the two mouse groups differed by $-0.131$ units (95% confidence interval, $-0.163$; $-0.0999$; $P = 6.98 \times 10^{-15}$). Statistical significance was set at $P < 0.05$. Data are presented as mean ± s.e.m. No statistical methods were used to predetermine sample sizes.

### Reporting summary

Further information on research design is available in the Nature Portfolio Reporting Summary linked to this article.

## Data availability

Data acquired in this manuscript have been deposited in Gene Expression Omnibus repository under the following accession numbers: ATAC-seq (GSE299550; https://www.ncbi.nlm.nih.gov/geo/query/acc.cgi?acc=GSE299550), CUT&RUN (GSE299551; https://www.ncbi.nlm.nih.gov/geo/query/acc.cgi?acc=GSE299551), RNA-seq (GSE299554; https://www.ncbi.nlm.nih.gov/geo/query/acc.cgi?acc=GSE299554). TME-Proteomics data for arginine methylation are deposited in Proteomics Identification Database (PRIDE) with identifier number PXD064423 (https://www.ebi.ac.uk/pride/archive/projects/PXD064423). Source data are provided with this paper.

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

## Acknowledgements

We thank P. Fitzgerald, S. Sirasanagandla, M. Yang and L. Harris for technical and administrative assistance; L. Long and H. Chi (St. Jude Children's Research Hospital) for providing retroviral vectors and Cas9-OT-I/P14 mice; C. Guy for microscope imaging assistance; M. Morrison for organizing patient samples (University of Tennessee Health Science Center) and the staff at the St. Jude Immunology Flow Cytometry Core Facility for cell sorting. Statistical calculations were performed using GraphPad Prism. We thank S. Pounds for the additional assistance with the statistical analysis of the data in Fig. 7b. This work was supported by National Institutes of Health grants AI123322 and CA231620 to D.R.G., American Lebanese Syrian Associated Charities (ALSAC) to G.K. and National Cancer Institute grant K99CA256262 to D.H.

## Author contributions

P.S. conceived the project, designed and performed most experiments, interpreted results and wrote the manuscript. A.G. contributed CUT&RUN, ATAC-seq and RNA-seq experiments. K.C.V. and E.B.-R. helped develop the flow cytometry panel and performed experiments. G.P. processed the LC–MS/MS experiments. D.H. and G.K. assisted with CAR-T cell experiments. K.I. and M.M.B. performed MD simulations and analyses. S.P. and M.J.C. analyzed the CUT&RUN, ATAC-seq and RNA-seq data. A.M. and J.P. performed and analyzed TMT-MS experiments. E.S.G. provided patient tumor samples. D.R.G. supervised the experimental designs, interpreted the results and wrote the manuscript.

## Competing interests

E.S.G. consults for Anviron and G.K. has patent applications in the fields of cell or gene therapy for cancer and has received honoraria from Cell Signaling and Kineticos within the last 2 years. During the course of this research, D.R.G. consulted for Sonata Therapeutics, Ventus Therapeutics, Mycos and ASHA Therapeutics, and received honoraria, travel and/or research support from Horizon/Amgen, Lilly and Boeringher-Ingleheim. The other authors declare no competing interests.

## Additional information

**Extended data** is available for this paper at https://doi.org/10.1038/s41590-025-02223-6.

**Correspondence and requests for materials** should be addressed to Piyush Sharma or Douglas R. Green.

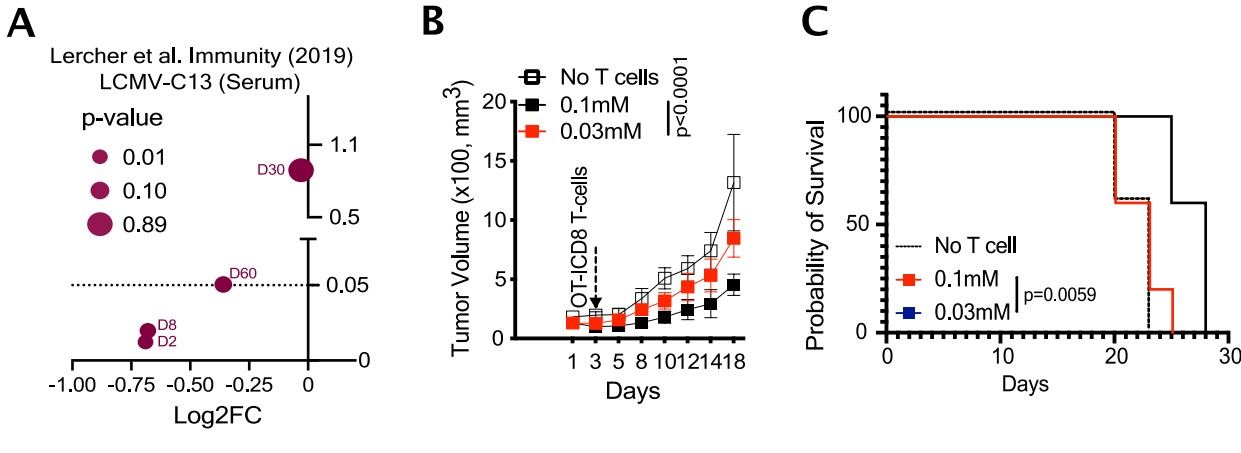

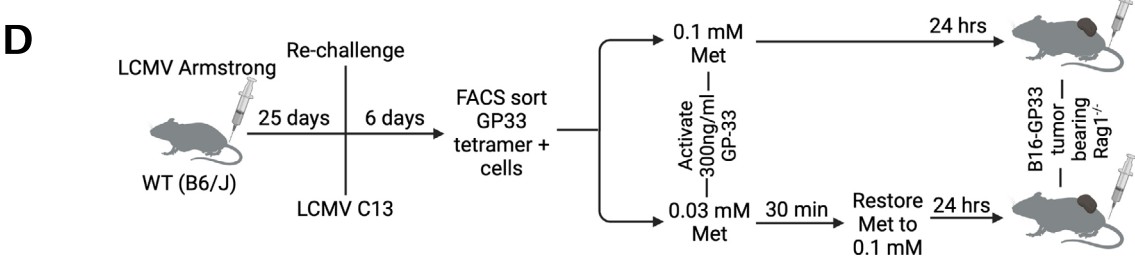

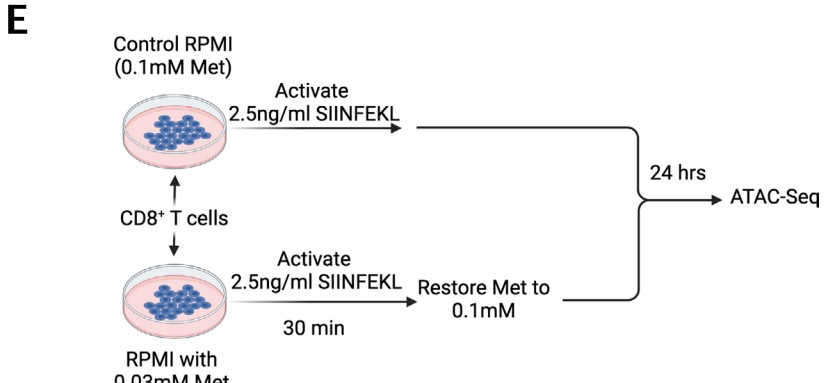

**Extended Data Fig. 1 | Methionine availability during TCR activation determines T cell fate. a**, Log2 fold change (FC) in serum Met levels post LCMV-C13 chronic infection (data from ref. 7). **b**, **c**, Tumour growth (**b**) and survival (**c**) of B16-OVA tumour-bearing WT mice treated with or without OT-I CD8+ T cells activated in 0.1 or 0.03 mM Met for 30 min before restoring Met to 0.1 mM for 24 h, as described in Fig. 1d (n = 5 mice/group). **d**, Schematic of experimental design (Fig. 1d) for generation of LCMV-GP33 specific T cells, which were then activated as shown before injection into B16-OVA tumour-bearing *Rag1−/−* mice. **e**, Schematic design for ATAC-Seq on OT-I T cells initially activated in 0.1 or 0.03 mM Met followed by restoration of Met to 0.1 mM for 24 h. Data are mean±s.d. 2-way ANOVA (**b**), and Mantel-Cox log rank test (**c**). Illustrations in **d**, and **e** were generated with Biorender.com.

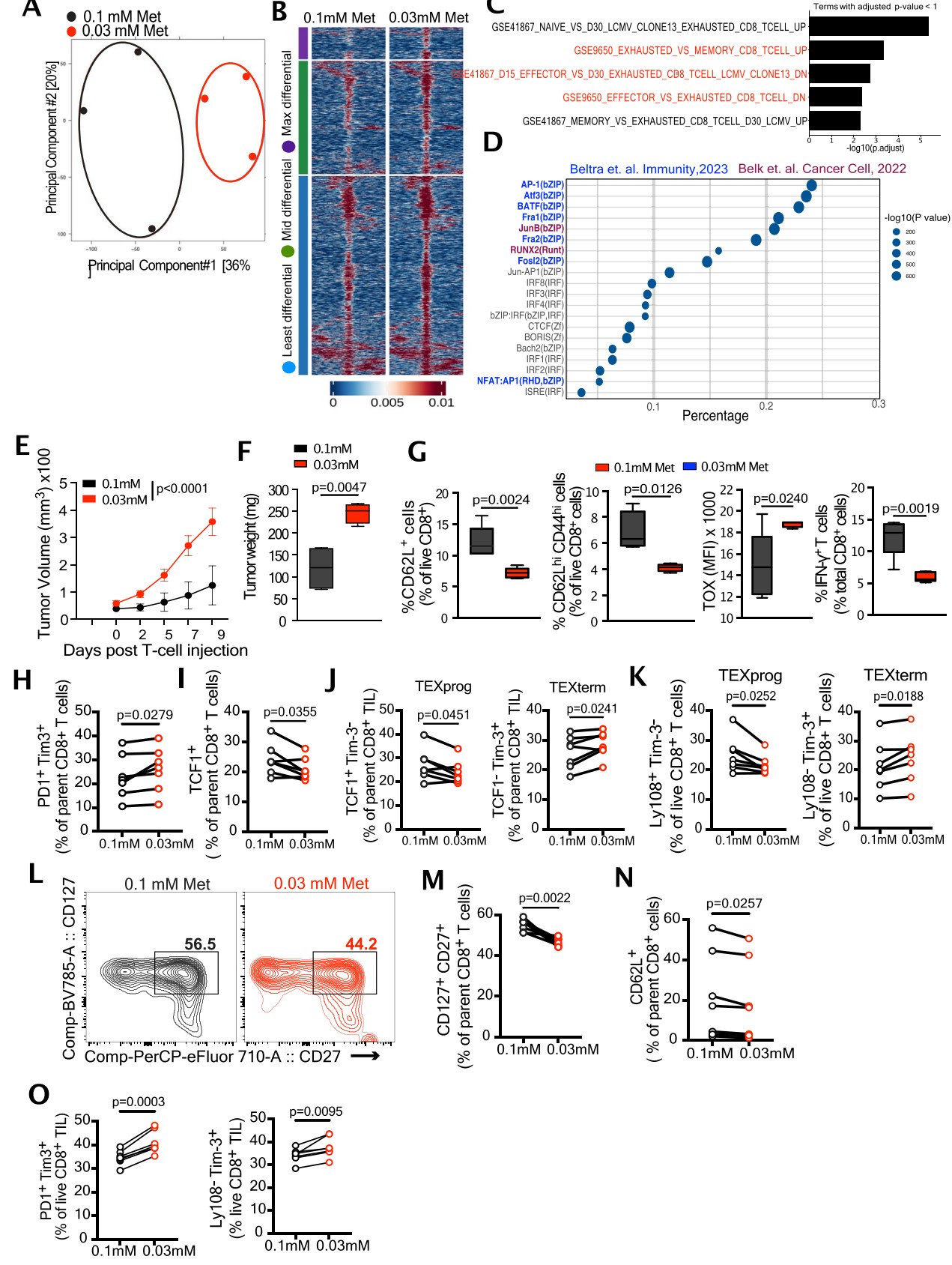

**Extended Data Fig. 2 | See next page for caption.**

**Extended Data Fig. 2 | Reduced methionine during TCR activation promotes T cell exhaustion. a**, PCA analysis of ATAC-Seq of OT-I T cells activated with 2.5 ng/ml SIINFEKL in 0.1 mM and 0.03 mM Met, then cultured in 0.1 mM Met for 24 h. **b**, Chromatin accessibility heatmap of T cells initially activated in 0.1 or 0.03 mM Met with each row representing peaks (p < 0.05 and log2 fold-change >1.5) displayed over the span of a 2 kb window with peak as center (grouped from least to max differential region), analyzed from ATAC-Seq from (**a**). **c**, Over-representation analysis of the differentially accessible regions (DAR) from (**b**) showing the top 5 gene sets associated with exhaustion. **d**, Homer analysis of (**a**) showing top 20 motifs and their percent coverage. Colour corresponds to the common exhaustion-associated motifs described in the indicated studies (blue and purple). **e**, **f**, Tumour growth (**e**) and weight (**f**) of B16-OVA tumour-bearing *Rag1*⁻/⁻ mice injected with OT-I T cells as in Fig. 1d. **g**, Frequency of CD62L⁺, Tcm (CD62L^hi CD44^hi), TOX expression (MFI), and IFNγ⁺ in OT-I CD8⁺ TIL isolated

from B16-OVA tumour-bearing *Rag1*⁻/⁻ mice at D9 post-injection with OT-I T cells activated as in Fig. 1d. **h**, Frequency of PD1⁺ Tim-3⁺ on OT-I CD8⁺ T cells at D12 post transfer of OT-I T cells as in Fig. 2a. **i**, Frequency of TCF1⁺ on OT-I CD8⁺ T cells at D12 post transfer of OT-I T cells as in Fig. 2a (n = 7). **j**, **k**, Frequency of Tex^prog and Tex^term identified via differential expression of TCF1 and Tim-3 (**j**) or Tim-3 and Ly108 (**k**) on OT-I CD8⁺ T cells as in Fig. 2a. **l**–**n**, Representative contour plot and quantification of CD127⁺CD27⁺ memory cells (**l**, **m**) and CD62L⁺ T memory cells in OT-I CD8⁺ T cells as in Fig. 2a. **o**, Frequencies of PD1⁺Tim-3⁺ exhausted T cells and Tim-3⁺Ly108⁻ TEXterm cells in OT-I CD8⁺ TIL isolated at D12 post of cells activated as in Fig. 1d from MC38-OVA tumour-bearing *Rag1*⁻/⁻ mice. (**a**–**d**: n = 3 mice/group; e-g: n = 5 mice/group; h-o: n = 7 mice/group). Data are mean±s.d. Boxplots shows min and max value with median as center. 2-way ANOVA (**e**), unpaired two-tailed Student's t-test (**f**, **g**) paired two-tailed Student's t-test (**h**–**p**).

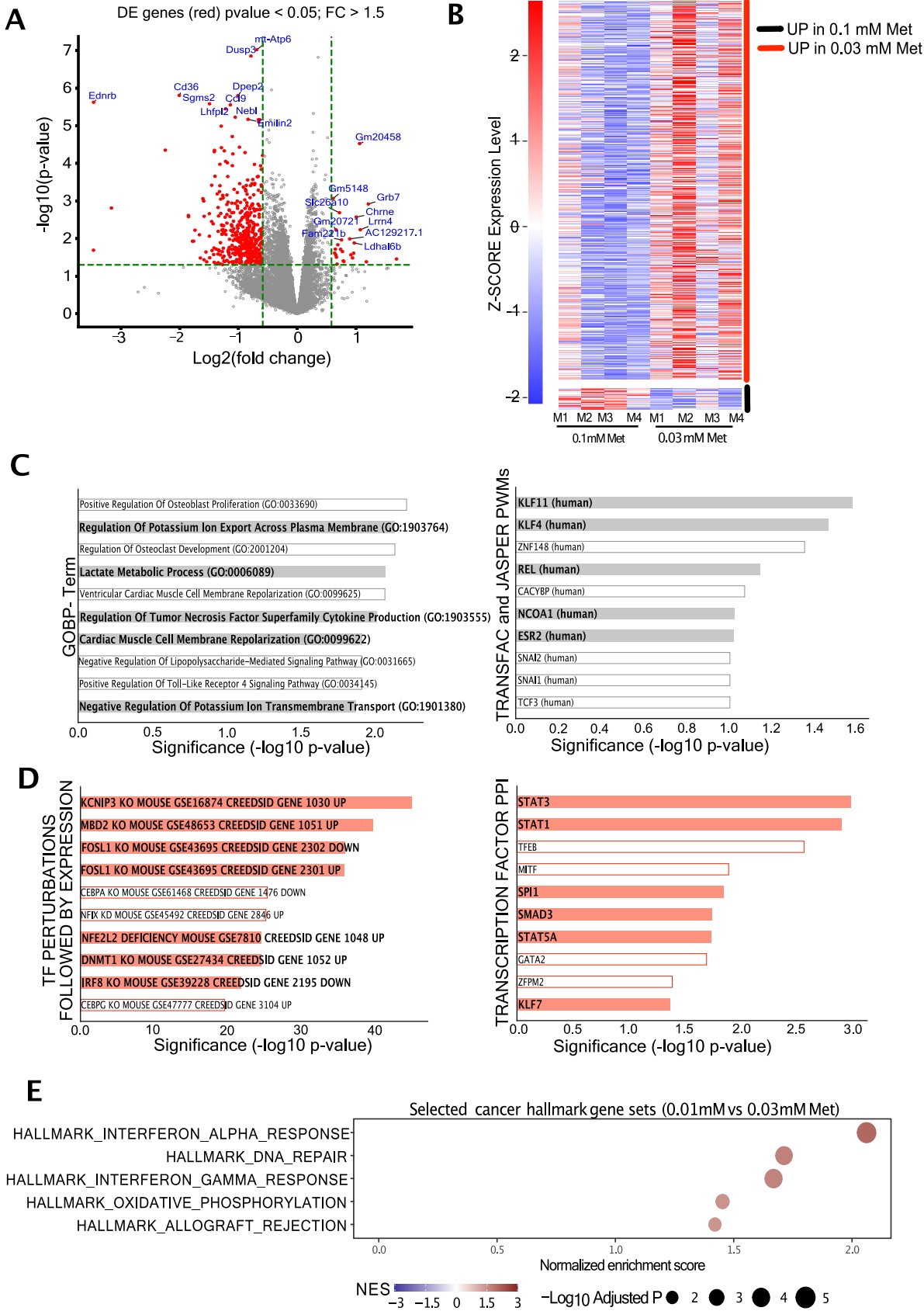

**Extended Data Fig. 3 | See next page for caption.**

**Extended Data Fig. 3 | RNA-Seq analysis shows enrichment of an exhaustion signature in low Met. a**, Volcano plot and top ten differentially expressed (DE) genes (red) (0.1 mM vs 0.03 mM, p < 0.05, FC > 1.5) from OT-I CD8⁺ TIL isolated from mice injected with OT-I T cells, initially activated in 0.1 mM Met or 0.03 mM Met (as in Fig. 1d), at day 9 post injection into B16-OVA tumour-bearing mice. **b**, Heatmap of DE genes, which shows increased expression in cells initially activated in either 0.03 mM Met or 0.1 mM Met from the RNAseq in (**a**). **c**, Over-representation analysis of DE genes with higher expression in TIL initially activated in 0.1 mM Met, showing enrichment of biological processes and transcription factors associated with T cell effector function and regulation. **d**, Over-representation analysis of DE genes with higher expression in TIL initially activated in 0.03 mM Met, showing enrichment of gene sets and transcription factors associated with promotion and maintenance of the exhausted T cell state. **e**, GSEA analysis of hallmark gene sets as analyzed from RNA-Seq of OT-I TIL initially activated in 0.1 or 0.03 mM Met, isolated from B16-OVA tumours at D9 post transfer. (**a**–**e**: n = 4 mice/group). RNA-Seq statistics were applied as described in Methods.

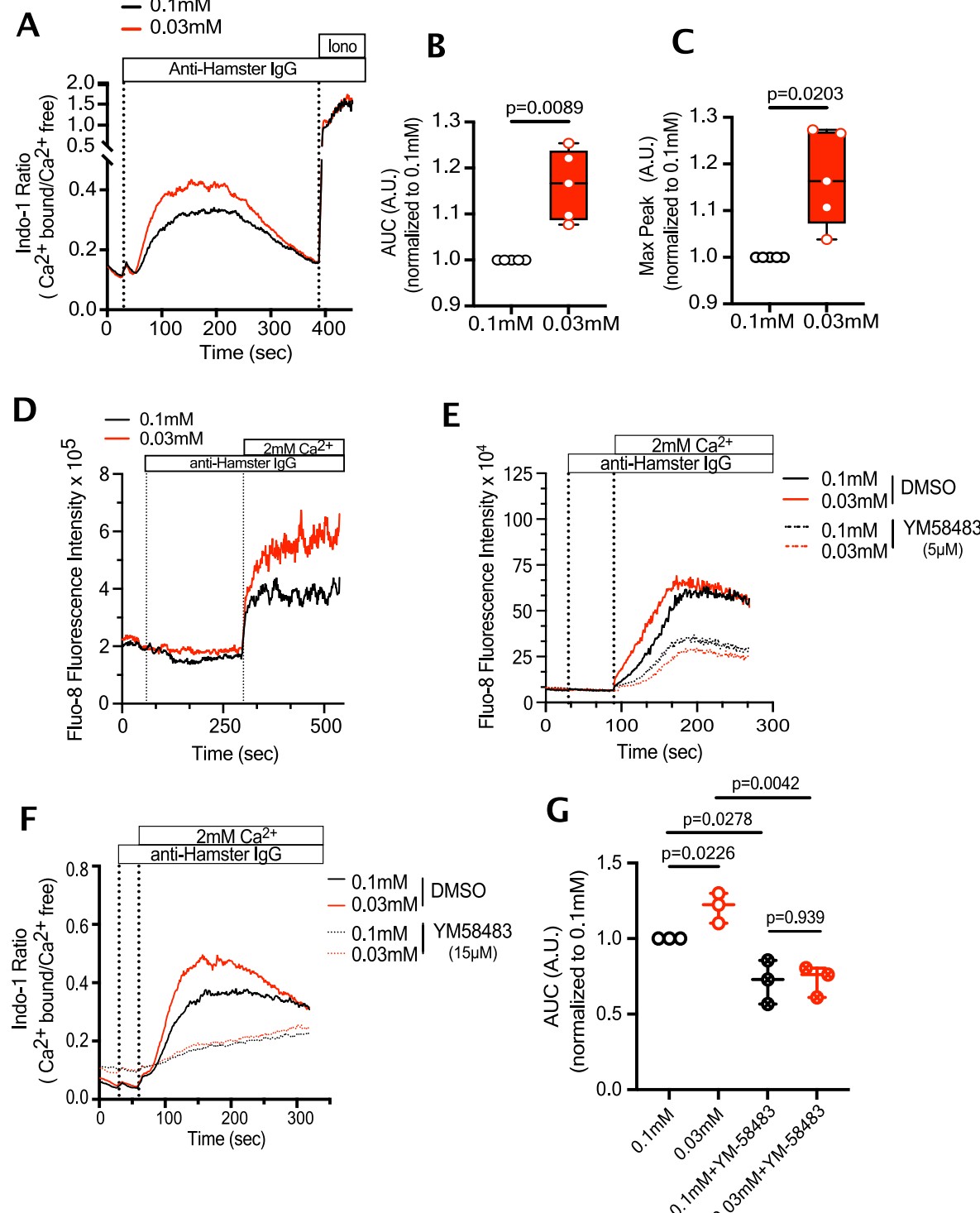

**Extended Data Fig. 4 | Met limitation promotes TCR-induced Ca²⁺ flux.**
**a**, Representative plot of Indo-1 analysis of Ca²⁺ flux of CD8⁺ T cells in ringer solution with 2 mM Ca²⁺, activated with anti-CD3 and anti-CD28 by anti-hamster IgG crosslinking in either 0.1 mM or 0.03 mM Met (n = 5). **b**, **c**, Normalized AUC (**b**) and max peak value (**c**) of Ca²⁺ flux analysis in (**a**) (n = 5). **d**, Representative plot of Fluo-8 AM analysis of Ca²⁺ flux in CD8⁺ T cells activated with anti-CD3 and anti-CD28 and anti-hamster IgG crosslinking in either 0.1 mM or 0.03 mM Met-containing Ca²⁺ Ringer solution with addition of 2 mM Ca²⁺ to measure Ca²⁺ influx (n = 2). **e**, Representative plot of Fluo-8 AM analysis of Ca²⁺ flux in

CD8⁺ T cells activated with anti-CD3 and anti-CD28 and anti-hamster IgG crosslinking in Ca²⁺-free Ringer solution containing either 0.1 mM or 0.03 mM Met and treated for one hour with DMSO or 5 μM YM58483 (n = 2). **f**, **g**, Representative plot (**f**) and normalized AUC (**g**) of Indo-1 analysis of Ca²⁺ flux in CD8⁺ T cells activated with anti-CD3 and anti-CD28 and anti-hamster IgG crosslinking in Ca²⁺-free Ringer solution containing either 0.1 mM or 0.03 mM Met and treated with either DMSO or 15 μM YM58483. (**a**–**c**: n = 5 mice/group; **d**, **e**: n = 2 mice/group; **f**, **g**: n = 3 mice/group). Data are mean±s.d. Boxplots shows min and max value with median as center. Paired two-tailed Student's t-test (**b**, **c**).

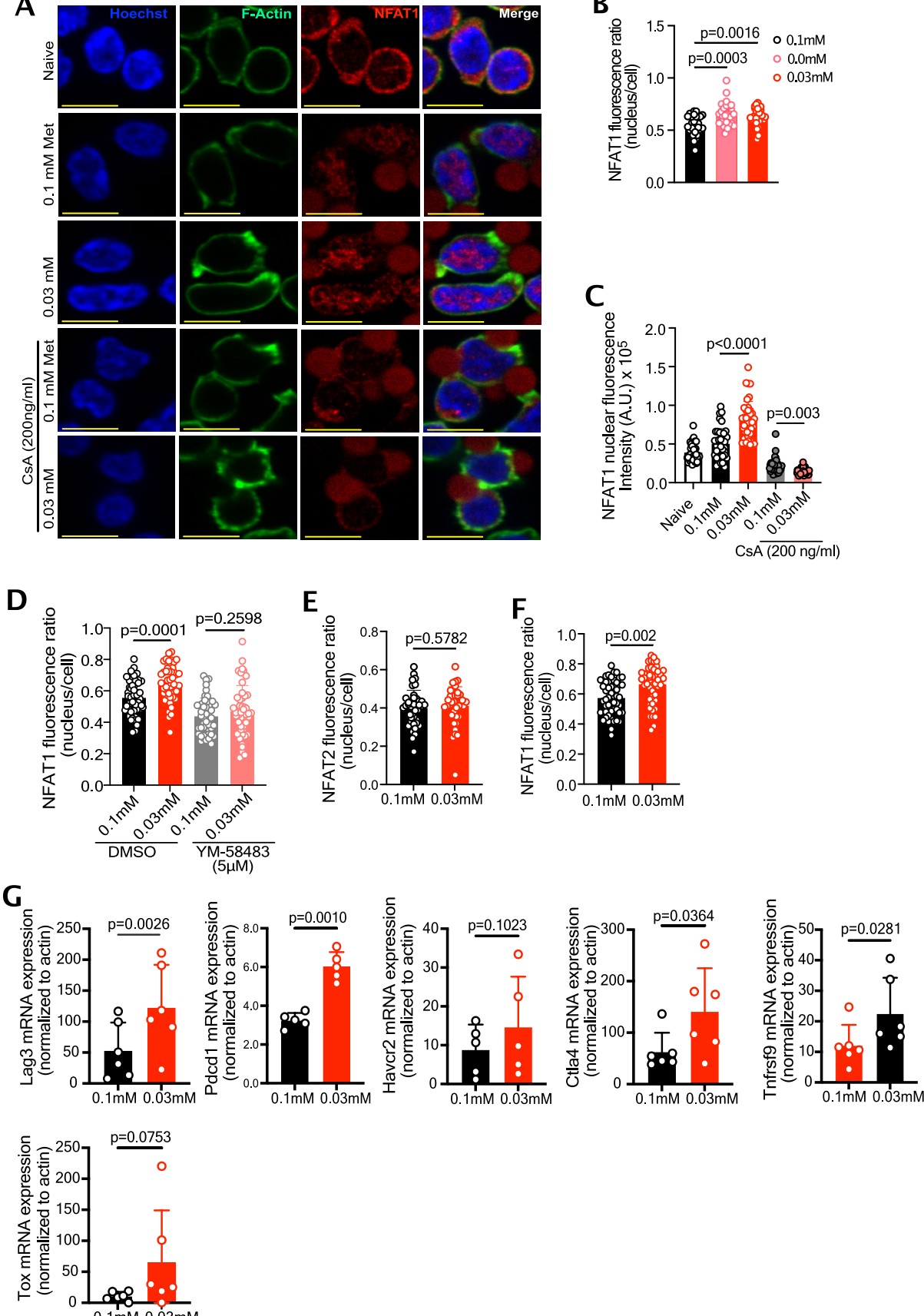

**Extended Data Fig. 5 | See next page for caption.**

**Extended Data Fig. 5 | Met limitation results in increased TCR-mediated NFAT1 activation. a**, Unedited image of Fig. 3d showing the unmasked, red anti-CD3/38 Dynabeads which are masked grey in the main Figure to highlight NFAT1 staining. **b**, NFAT1 intensity quantification (nuclear to total cell ratio) of CD8+ T cells activated in either 0.1 mM, 0.00 mM or 0.03 mM Met for 30 min with anti CD3/28 Dynabeads and stained for NFAT1 (each circle represents one cell, n = 28 cells/group). **c**, Quantification of nuclear NFAT1 from the experiment shown in Fig. 3d. **d**, NFAT1 intensity quantification (nuclear to total cell ratio) of CD8+ T cells activated for 30 min by CD3/28 Dynabeads in either 0.1 mM or 0.03 mM Met in the presence of DMSO or 5 µM YM58483 (each circle represents one cell, n = 47 cells/group). **e**, NFAT2 quantification (nuclear to total cell ratio) of CD8+ T cells, activated in 0.1 mM or 0.03 mM Met for 30 min with anti-CD3/28 Dynabeads (each circle represents one cell, n = 40 cells). **f**, NFAT1 quantification (nuclear to total cell ratio) of previously activated OT-I T cells, treated with anti-CD3/28 Dynabeads for 30 min. T cells were initially activated with 2.5 ng/ml SIINFEKL for 24 h in control medium and rested with 10 ng/ml mIL-7 for 48 h (each circle represents one cell, n = 50 cells/group). **g**, Normalized RNA expression quantified by quantitative-PCR performed 24 h post activation of OT-I CD8+ T cells activated with 2.5 ng/ml SIINFEKL in 0.1 mM or 0.03 mM Met for 30 min followed by 0.1 mM Met (n = 5–6 biological replicates/group). Data are mean±s.d. Unpaired two tailed Student's t-test (**b**–**f**), paired two-tailed Student's t-test (**g**).

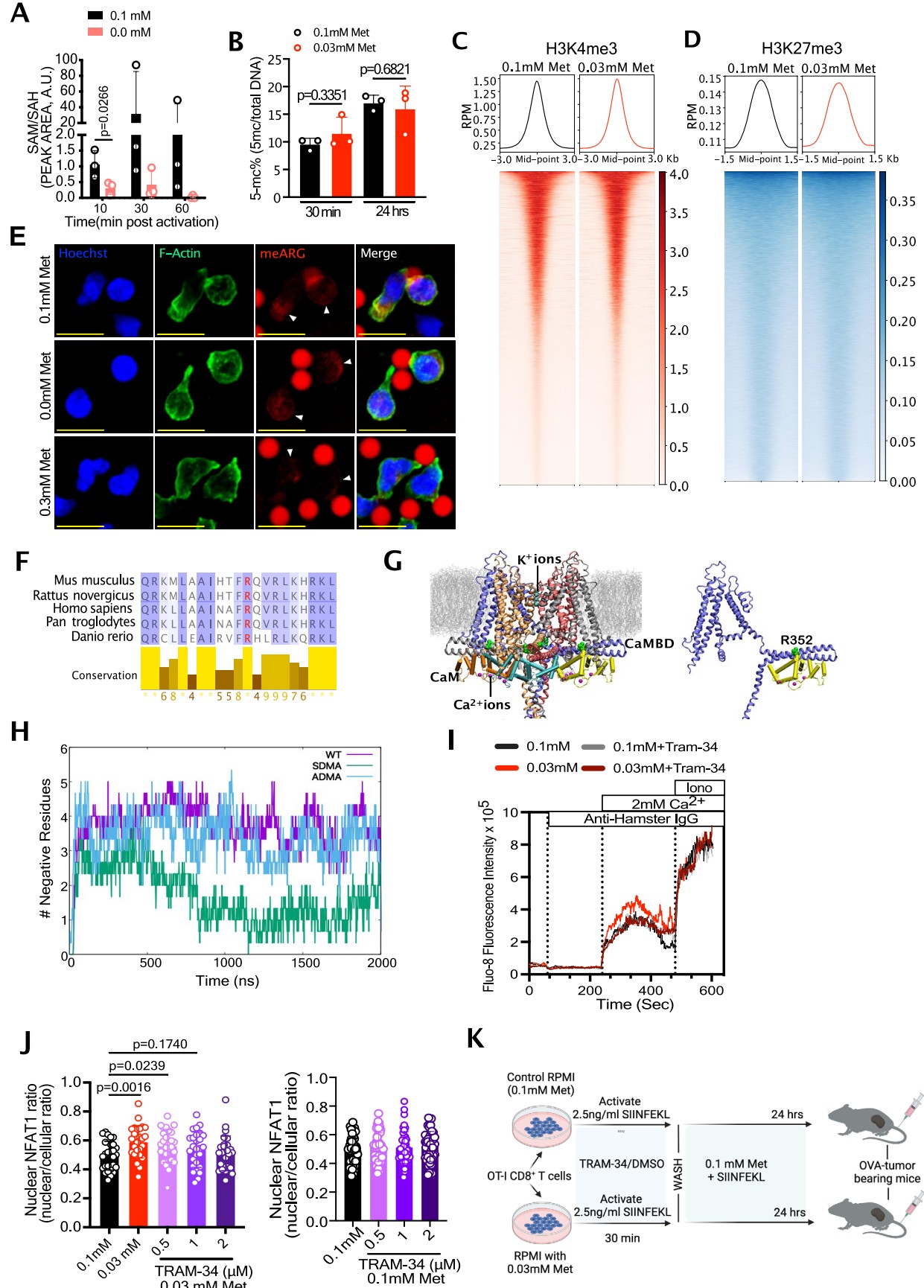

**Extended Data Fig. 6 | See next page for caption.**

**Extended Data Fig. 6 | Initial Met limitation alters KCa3.1 methylation, but not DNA or histone methylation. a**, Methylation potential of 10 ng/ml SIINFEKL-activated OT-I T cells in 0.1 mM or 0.0 mM Met at 10, 30 and 60 min. **b**, %5-methylcytosine (%5-mc) of OT-I CD8$^+$ T cells activated with 2.5 ng/ml SIINFEKL in 0.1 mM or 0.03 mM Met for 30 min, then 0.1 mM Met for 24 h. **c**, **d**, Histograms (top) and heat-map (bottom) of H3K4me3 (**c**) and H3K27me3 (**d**) CUT&RUN (read count per million, normalized to background) in OT-I CD8$^+$ T cells activated as in (**b**) at 24 h post activation. **e**, Unedited image of Fig. 4a with unmasked, red CD3/28 Dynabeads (grey in main Figure to highlight me-Arg staining). **f**, Conserved arginine (red) in KCa3.1, analyzed by protein-BLAST and visualized using Jalview. **g**, Tetrameric assembly of human KCa3.1-Calmodulin complex in membrane (left). KCa3.1 is represented as ribbons, calmodulin quartet as cartoon, interacting Ca$^{2+}$ ions as purple spheres, and K$^+$ ions as cyan spheres. KCa3.1 monomer (blue) shown in complex with calmodulin (yellow) and demethylated arginine-352 (green) (right). **h**, MD simulations analysis.

Illustration of total counts of negatively charged residues within a 4 Å radius of all three R352/SDMA/ADMA of huKCa3.1 over simulation time (averaged across all four monomers, three MD trials). **i**, Representative plot of Fluo-8 AM analysis of Ca$^{2+}$ flux in CD8$^+$ T cells activated by IgG crosslinking anti-CD3 and anti-CD28 in Ca$^{2+}$-free Ringer solution with either 0.1 mM or 0.03 mM Met and treated with either DMSO or 1 µM TRAM-34 (n = 2 mice/group). **j**, NFAT1 quantification (nuclear to total cell ratio) in OT-I CD8$^+$ T cells activated for 30 min with anti CD3/28 Dynabeads in 0.03 mM Met (left) or 0.1 mM Met (right) in 0.5, 1 or 2 µM TRAM-34 (each circle=one cell, n = 27–45 cells/group). **k**, Schematic design for OT-I T cell activation in 0.1 or 0.03 mM Met, treated with DMSO or TRAM-34 for 30 min, followed by washing and culturing in 0.1 mM Met plus SIINFEKL (2.5 ng/ml) for 24 h before injection into B16-OVA tumour-bearing *Rag1*$^{-/-}$ mice. (**a**–**d**: n = 3 mice/group) Data are mean±s.d. Unpaired one-tailed Student's t-test (**a**) or unpaired two-tailed Student's t-test (**b**, **j**). Illustration in **k** created with Biorender.com.

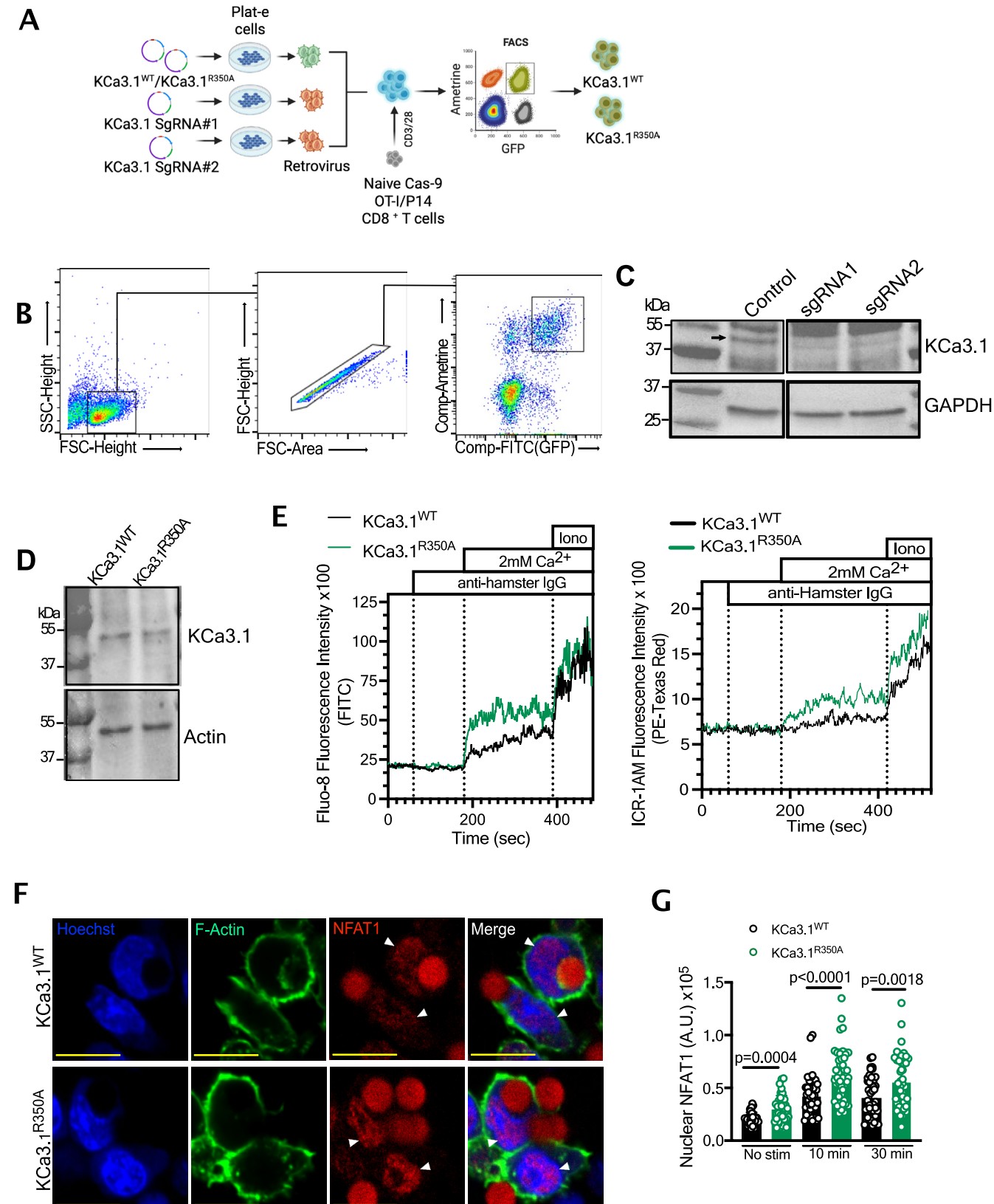

**Extended Data Fig. 7 | See next page for caption.**

**Extended Data Fig. 7 | KCa3.1 R350 methylation regulates Ca²⁺- mediated NFAT1 activation. a**, Schematic of experimental design of endogenous KCa3.1 knockdown with ectopic expression of KCa3.1$^{WT}$ and KCa3.1$^{R350A}$ in Cas9-OT-I or P14⁺ CD8⁺ T cells. **b**, Sorting strategy for *in vitro*-generated KCa3.1$^{WT}$ and KCa3.1$^{R350A}$ T cells; gate shows the cells that express WT or R350A KCa3.1 with endogenous KCa3.1 knockdown. **c**, Immunoblot showing knockdown of KCa3.1 with sgRNA1 and sgRNA2, compared to control. **d**, Immunoblot of KCa3.1 expression in KCa3.1$^{WT}$- and KCa3.1$^{R350A}$-expressing CD8⁺ T cells. **e**, Representative plots of Fluo-8 (left) and ICR-1 AM (right) analysis of Ca²⁺ flux in KCa3.1$^{WT}$ and

KCa3.1$^{R350A}$ T cells activated with anti-CD3 and anti-CD28 by anti-hamster IgG crosslinking in Ca²⁺-free Ringer solution. **f**, Unedited image of Fig. 5d showing the unmasked CD3/38 Dynabeads which are masked grey in the main figure to highlight NFAT1 staining. **g**, Nuclear NFAT1 quantification of KCa3.1$^{WT}$ and KCa3.1$^{R350A}$, either control (no Dynabeads) or activated with anti-CD3/28 Dynabeads for 10 min (each circle represents one cell, n = 37–45 cells/group). Data are mean±s.d. Unpaired two-tailed Student's t-test. Illustrations in **a** created with BioRender.com.

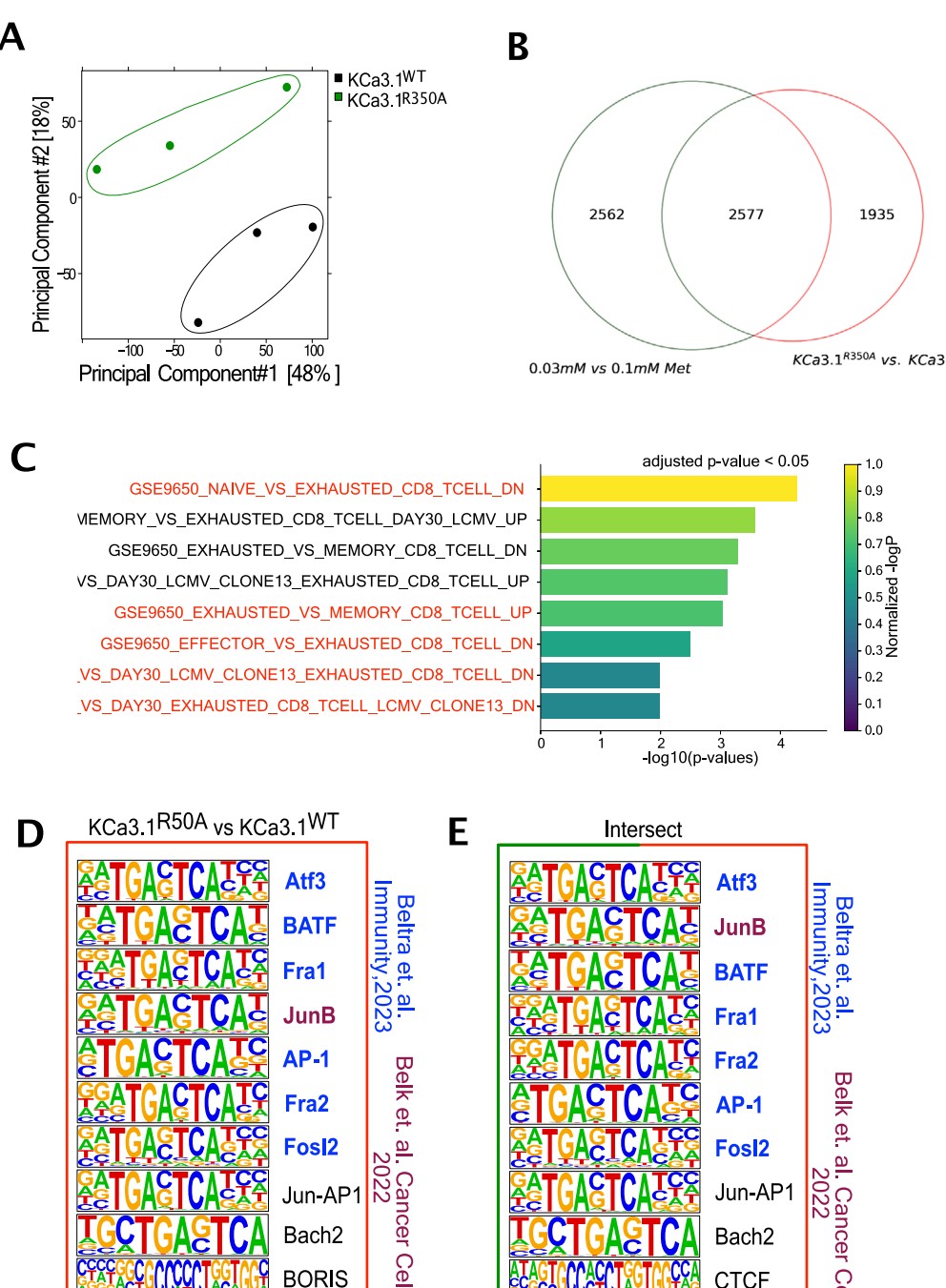

**Extended Data Fig. 8 | Ablation of KCa3.1 R350 methylation shows similar ATAC-Seq signature as of T cells activated in reduced Met. a**, PCA analysis of ATAC-Seq of KCa3.1$^{WT}$- and KCa3.1$^{R350A}$- activated with 2.5 ng/ml SIINFEKL for 24 h. n = 3. **b**, Common DAR (p < 0.05, fold change>1.5) between ATAC-Seq of OT-I cells expressing KCa3.1$^{WT}$ or KCa3.1$^{R350A}$, or OT-I cells initially activated in 0.1 mM or 0.03 mM Met as in (**a**) and Extended Data Fig. 2b. **c**, Over-representation analysis of the common DAR from (**b**) showing the top 8 gene sets associated with exhaustion and activation. **d**, **e**, Top 10 motifs analyzed by HOMER in ATAC-Seq DAR's from 24 h activated OT-I CD8$^{+}$ T cells expressing KCa3.1$^{R350A}$ vs KCa3.1$^{WT}$ (**d**), and from the common DAR's from (**b**, **e**). Colour corresponds to the common exhaustion-associated motifs also described in the indicated studies (blue and purple).

# Extended Data Fig. 9

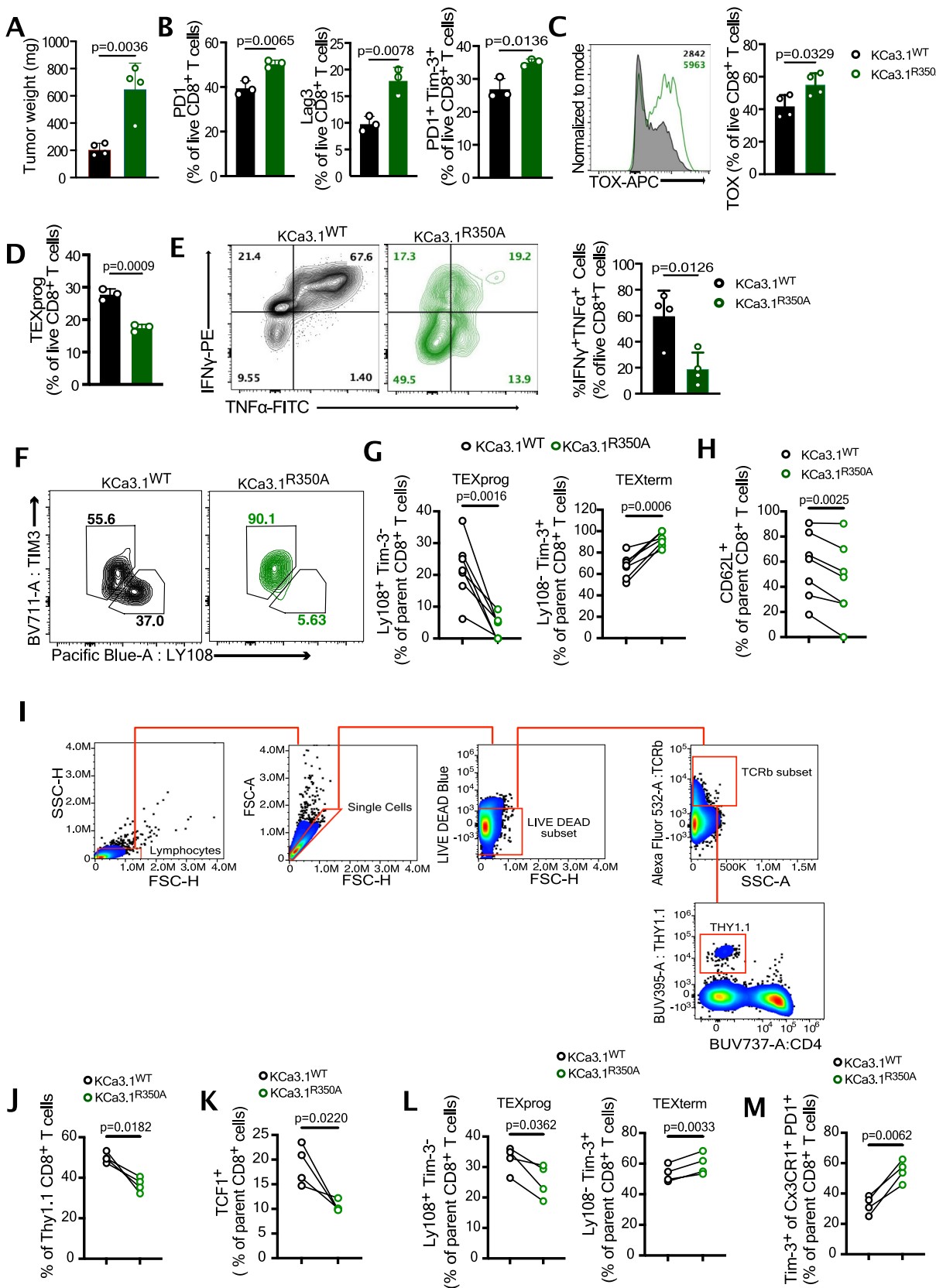

**Extended Data Fig. 9 | See next page for caption.**

**Extended Data Fig. 9 | Ablation of KCa3.1 R350 methylation promotes T cell exhaustion in an *in-vivo* tumor and infection model. a**, Tumour weight of B16-OVA tumours isolated at D12 post transfer of OT-I T cells expressing KCa3.1$^{WT}$ or KCa3.1$^{R350A}$. **b**, Surface expression of PD1$^+$, Lag3$^+$ and PD1$^+$Tim-3$^+$ in CD8$^+$ TIL isolated at D12 post transfer into B16-OVA tumour-bearing mice of OT-I T cells expressing KCa3.1$^{WT}$ or KCa3.1$^{R350A}$. **c, d,** Representative histogram of TOX expression and quantification (**c**) and of Tex$^{prog}$ (CD69$^{lo}$Ly108$^{hi}$) (**d**) in CD8$^+$ TIL expressing KCa3.1$^{WT}$ or KCa3.1$^{R350A}$ isolated at D12 as in (**b**). **e**, Representative contour plot (left) and quantification of IFNγ$^+$TNF$^+$ (right) in OT-I T cells expressing KCa3.1$^{WT}$ or KCa3.1$^{R350A}$ at D12 post transfer as in (**b**). **f–h**, Representative contour plot (**f**) and quantification (**g**) Tex$^{prog}$ (Ly108$^+$Tim-3$^-$) and Tex$^{term}$ (Ly108$^-$Tim-3$^+$) and CD62L$^+$ (**h**) on congenically marked OT-I T cells expressing KCa3.1$^{WT}$ or KCa3.1$^{R350A}$, mixed and transferred at a ratio of 1:1. TIL were isolated from B16-OVA tumours at D12 post T cell transfer. **i**, Gating strategy to identify transferred Thy1.1$^+$ CD8$^+$ T cells in spleen at day 9 post infection. **j**, Frequency of KCa3.1$^{WT}$- and KCa3.1$^{R350A}$-expressing P14 T cells in spleens, D9 post infection with LCMV-Clone-13. **k**, Frequency of TCF1$^+$ cells in KCa3.1$^{WT}$- and KCa3.1$^{R350A}$-expressing P14 T cells in spleens, analyzed as in (**i**). **l**, Frequency of Tim-3$^-$Ly108$^+$ Tex$^{prog}$ and Tim-3$^+$Ly108$^-$ Tex$^{term}$ in KCa3.1$^{WT}$- and KCa3.1$^{R350A}$-expressing P14 T cells in spleens, analyzed as in (**i**). **m**, Frequency of Tim-3$^+$Cx3CR1$^+$PD1$^+$ Tex$^{eff}$ KCa3.1$^{WT}$- and KCa3.1$^{R350A}$-expressing P14 T cells in spleens, analyzed as in (**i**). (**a**, **c**, **e**, **h–l**: n = 4 mice/group; **b**, **d**: n = 3 mice/group; **f**, **g**: n = 7 mice/group) Data are mean±s.d. Unpaired two-tailed Student's t-test (**a–e**) and paired two-tailed Student's t-test (**f**, **l**).

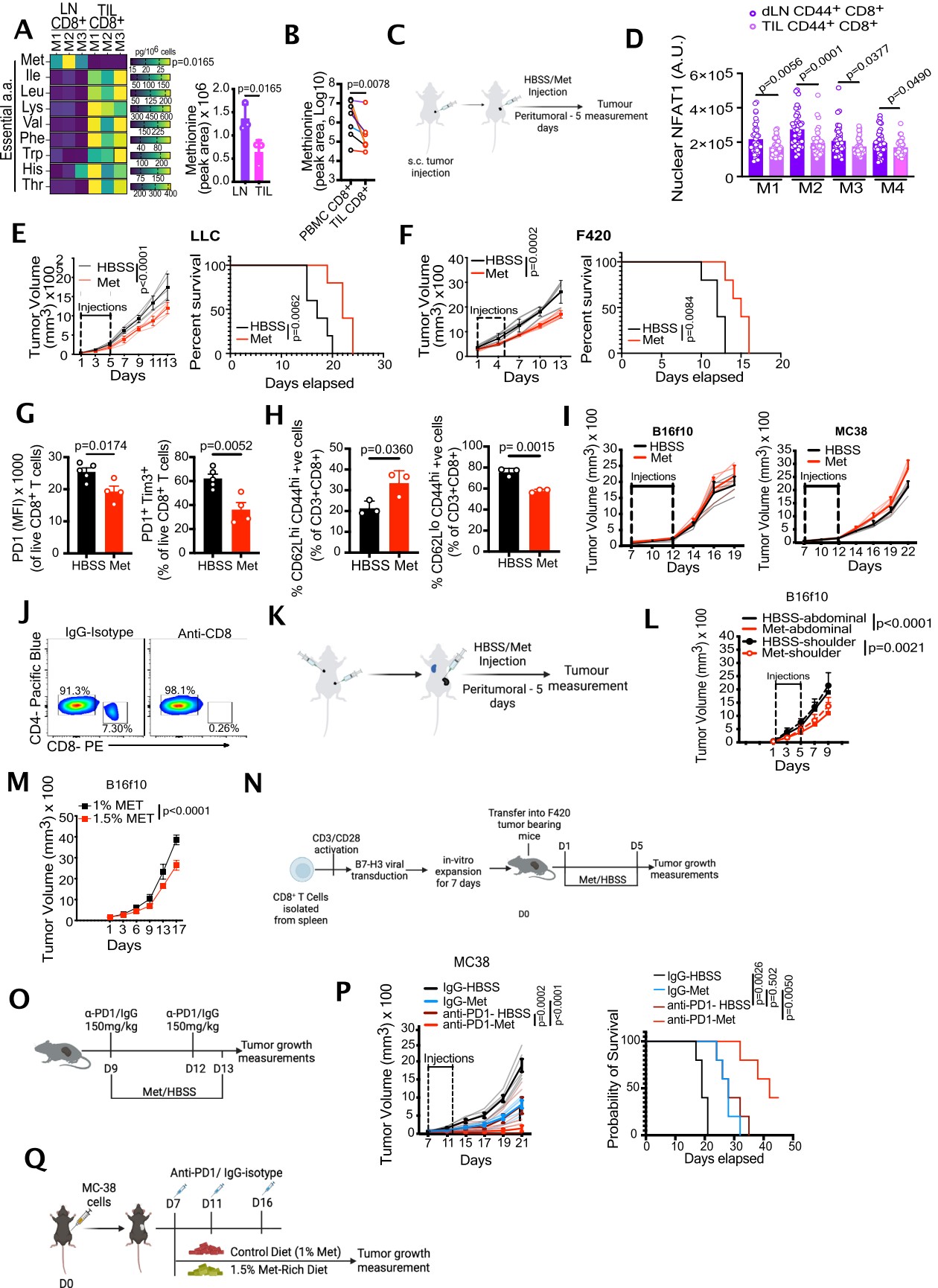

**Extended Data Fig. 10 | See next page for caption.**

**Extended Data Fig. 10 | Acute Met supplementation enhances T cell mediated tumour control and tumour immunotherapies. a**, Quantification of amino acids (left) and peak area of Met (right) in CD8⁺ T cells from dLN and tumours at day 12 post implantation. **b**, Quantification of intracellular Met in CD8⁺ T cells from PBMC and primary patient colorectal tumours. **c**, Experimental design of acute HBSS/Met peri-tumoural treatment. **d**, Nuclear NFAT1 quantification of CD44⁺ CD8⁺ T cells from B16 subcutaneous tumours and dLN 5 days post HBSS/Met injection as in (**c**) (M=mouse, each circle=one cell, n = 40 cells/group). **e**, **f**, Tumour growth (left) and survival (right) of WT mice with sub-cutaneous Lewis-lung carcinoma (LLC) tumours (**e**) (representative of two experiments), and F420 osteosarcoma tumours (**f**) treated as in (**c**). **g**, **h**, MFI of PD1, frequency of PD1⁺Tim-3⁺ (**g**), Tcm (CD62LʰⁱCd44ʰⁱ) and Tem (CD62LˡᵒCD44ʰⁱ) (**h**) on CD8⁺ TIL isolated from B16 tumours 2 days after 5 daily peri-tumoural injections with HBSS or Met. **i**, Growth of B16 and MC38 tumours in NSG mice treated as in (**c**). **j**, Representative plot showing % CD8 following anti-CD8 and IgG-treatment.

**k**, Experimental design of contralateral tumour experiment. **l**, Growth of B16 tumours implanted as in (**k**) following treatment of flank tumours as in (**c**) (representative of two experiments). **m**, B16 growth in WT mice fed with either 1% Met chow or 1.5% Met chow for 7 days post implantation. **n**, **o**, Experimental design to assess the effect of HBSS or Met supplementation on CAR-T cell therapy in murine solid tumours (**n**) and on ICB (anti-PD1) treatment. **p**, Tumour growth (left) and survival (right) of MC38 in WT mice, treated either with anti-PD1 or IgG and supplemented with HBSS or Met as in (**o**) (representative of two experiments)., **q**, Experimental design of effect of Met-rich diet on tumour growth upon anti-PD1 or IgG treatment. (**a**, **h**: n = 3 mice/group; b: n = 8 patients/group; **e-f**, **i**, **l-m**, **p**: n = 5 mice/group; g: 4–5 mice/group; k: n = 4 mice/group). Data are mean±s.d. Unpaired (**a**, **d**, **g**, **h**) and paired (**b**) two-tailed Student's t-test, two-way ANOVA (**e**, **f**, **p** (left), **i**, **l**, **m**) and Mantel-Cox log rank test (**e**, **f**, p(right)). Illustrations in **c**, **k**, **n**, **o** and **q** created with Biorender.com.

# Reporting Summary

## Statistics

For all statistical analyses, confirm that the following items are present in the figure legend, table legend, main text, or Methods section.

| n/a | Confirmed | |
|---|---|---|
| ☐ | ☒ | The exact sample size (*n*) for each experimental group/condition, given as a discrete number and unit of measurement |
| ☐ | ☒ | A statement on whether measurements were taken from distinct samples or whether the same sample was measured repeatedly |
| ☐ | ☒ | The statistical test(s) used AND whether they are one- or two-sided |
| | | *Only common tests should be described solely by name; describe more complex techniques in the Methods section.* |
| ☐ | ☒ | A description of all covariates tested |
| ☐ | ☒ | A description of any assumptions or corrections, such as tests of normality and adjustment for multiple comparisons |
| ☐ | ☒ | A full description of the statistical parameters including central tendency (e.g. means) or other basic estimates (e.g. regression coefficient) AND variation (e.g. standard deviation) or associated estimates of uncertainty (e.g. confidence intervals) |
| ☐ | ☒ | For null hypothesis testing, the test statistic (e.g. *F*, *t*, *r*) with confidence intervals, effect sizes, degrees of freedom and *P* value noted |
| | | *Give P values as exact values whenever suitable.* |
| ☒ | ☐ | For Bayesian analysis, information on the choice of priors and Markov chain Monte Carlo settings |
| ☒ | ☐ | For hierarchical and complex designs, identification of the appropriate level for tests and full reporting of outcomes |
| ☒ | ☐ | Estimates of effect sizes (e.g. Cohen's *d*, Pearson's *r*), indicating how they were calculated |

*Our web collection on statistics for biologists contains articles on many of the points above.*

## Software and code

Policy information about availability of computer code

| Data collection | BD FACSDiva software (LSRII and Fortessa) or SpectroFlo (Cytek Aurora)was used to collect flow cytometry data. Slidebook 6(3i) was used to collect confocal microscopy data. Bio-Rad ChemiDoc was used for immunoblot data acquisition. |
|---|---|
| Data analysis | Flowjo 10.8.2 for FACS; GraphPad 9.5.0 for statistics; slidebook 6 was used for microscope images; Affymetrix Expression Console v1.1; limma v.3.34.9 ; ggplot2 (v2.2.1); Trim Galore (version 0.5.0) ; Bowtie 2 (version 2.3.5.1) ; Picard MarkDuplicates function (version 2.19.0); SAMtools (version 1.9); BamTools (version 2.5.1); MACS (version 2.1.2); BEDTools (version 2.27.1); bedGraphToBigWig (version 377); deepTools plotHeatmap (version 3.2.1); DiffBind (version 2.16.0); ChIPseeker (version 1.26.2); clusterProfiler (version 3.18.1); STAR (version 2.7.5a) for CUT&RUN, ATAC-Seq and sequencing data. |

For manuscripts utilizing custom algorithms or software that are central to the research but not yet described in published literature, software must be made available to editors and reviewers. We strongly encourage code deposition in a community repository (e.g. GitHub). See the Nature Portfolio guidelines for submitting code & software for further information.

## Data

Policy information about **availability of data**

All manuscripts must include a **data availability statement**. This statement should provide the following information, where applicable:

- Accession codes, unique identifiers, or web links for publicly available datasets
- A description of any restrictions on data availability
- For clinical datasets or third party data, please ensure that the statement adheres to our **policy**

RNA-seq, ATAC-seq and CUT&RUN data that support the findings of this study have been deposited in the Gene Expression Omnibus (GEO; https://www.ncbi.nlm.nih.gov/geo/) under accession number GSE299554, GSE299550 and GSE299551 respectively. Proteomics data has been deposited in PRIDE database with identifier number PXD064423. Publicly available database used in this study are from the Molecular Signatures Database (http://www.broadinstitute.org/gsea/msigdb/)

## Human research participants

Policy information about **studies involving human research participants and Sex and Gender in Research.**

| | |
|---|---|
| Reporting on sex and gender | Blood were from healthy blood donor. Age and other information of blood donor are unavailable.<br>Tumor samples were collected after surgery. Patients were de-indentified. |
| Population characteristics | No data on population characteristics was collected for this manuscript. |
| Recruitment | Blood from healthy normal blood donor were collected by the Blood Donor Center at St. Jude Children's Research Hospital. Apheresis rings for research are byproduct of normal donations. Samples were used without any selection.<br>For tumor samples, patients over 18 years of age were consented for their participation. |
| Ethics oversight | All human studies were in compliance with the Declaration of Helsinki. Blood donors were recruited by the Blood Donor Center at St. Jude Children's Research Hospital. Cancer patients were recruited by University of Tennessee Health Science Center. Blood donors provided written consent for research use of their blood products not used in transfusions and Cancer Patients provided written consent that tumor biopsies will be used for research purposes only,  which has been reviewed and approved by the Institutional Review Board at St. Jude Children's Research Hospital and University of Tennessee Health Science Center. |

Note that full information on the approval of the study protocol must also be provided in the manuscript.

# Field-specific reporting

Please select the one below that is the best fit for your research. If you are not sure, read the appropriate sections before making your selection.

☒ Life sciences  ☐ Behavioural & social sciences  ☐ Ecological, evolutionary & environmental sciences

For a reference copy of the document with all sections, see nature.com/documents/nr-reporting-summary-flat.pdf

# Life sciences study design

All studies must disclose on these points even when the disclosure is negative.

| | |
|---|---|
| Sample size | No sample size calculation was performed to predetermine sample size. We determined the sample size based on similar experiments reports in previous publications (). Sample size was selected to maximize the chance of uncovering mean difference which is also statistically significant. |
| Data exclusions | One of the patient data was excluded according to the outlier test performed on Graphpad Prism. IN proteomics, one biological replicate M2 quantitative proteomics data showed higher variability (>SD), and therefore removed from the final differential expression analysis. |
| Replication | All the experimental finding were reproduced as validated by at least two independent experiments. For CUT&RUN experiment, at least two replicates were collected for each group. For amino acid LC-MS/MS measure, three independent biological replicates for group were analyzed. |
| Randomization | Age-and sex-matched mice were assigned randomly to experimental and control groups. For other experiments samples are randomly located into experiment groups. |
| Blinding | The investigators were not blinded to group allocation during data collection or analysis, as there was no subjective measurement in our experiments. This approach is considered standard for experiments of the type performed in this study. |

# Reporting for specific materials, systems and methods

We require information from authors about some types of materials, experimental systems and methods used in many studies. Here, indicate whether each material, system or method listed is relevant to your study. If you are not sure if a list item applies to your research, read the appropriate section before selecting a response.

## Materials & experimental systems

| n/a | Involved in the study |
|-----|------------------------|
| ☐ | ☒ Antibodies |
| ☐ | ☒ Eukaryotic cell lines |
| ☒ | ☐ Palaeontology and archaeology |
| ☐ | ☒ Animals and other organisms |
| ☒ | ☐ Clinical data |
| ☒ | ☐ Dual use research of concern |

## Methods

| n/a | Involved in the study |
|-----|------------------------|
| ☒ | ☐ ChIP-seq |
| ☐ | ☒ Flow cytometry |
| ☒ | ☐ MRI-based neuroimaging |

## Antibodies

**Antibodies used**

The following antibodies were used for flow cytometry:
Antibodies from BioLegend included Pacific Blue anti-mouse Ly108 (330-AJ, 134608), BV510 anti-mouse KLRG1 (2F1, 138421), BV570 anti-mouse CD62L (MEL-14, 104433), BV711 anti-mouse Tim-3 (RMT3-23, 119727), BV785 anti-mouse CD127 (A7R34, 135037), PE-Cy5 anti-mouse Granzyme-B (QA16A02, 372226), PE-Fire 700, anti-mouse CD4 (GK1.5, 100484), APC/Cy7 anti-mouse TNF-alpha (MP6-XT22, 506344), PE-Cy7 anti-mouse KLRG1 (2F1/KLRG1, 138416), PerCP-Cy 5.5 anti-mouse CD62L (MEL-14, 104432), BV605 anti-mouse CD127 (A7R34, 135025), Pacific blue anti-mouse CD69 (H1.2F3, 104524), Pacific blue anti-mouse CD45.1 (110722), APC anti-mouse PD1 (RL388, 109111), BV711 anti-mouse PD1 (29F.1A12, 135231), PE anti-mouse IFN-g (XMG1.2, 505808), FITC anti-mouse TNF-alpha(MP6-XT22, 506304), Pacific Blue anti-human CD45 (HI30, 982306), APC/Cy7 anti-human CD8 (RPA-T8, 344713). BUV563 anti-mouse LAG3 (C9B7W, 741350), BUV615 anti-mouse CD69 (H1.2F3, 751593), BUV805 anti-mouse CXCR3 (173, 748700), BV421 anti-mouse EOMES (X4-83, 567166), BV480 anti-mouse CD45.1 (A20, 746666), Alexa-Fluor 488 anti-mouse TCF1 (S33-966, 567018), Alexa-Fluor 647 TOX (NAN448B, 568356), BUV 496 anti-mouse Ly-108 (13G3, 750046), APC/Cy7 anti-mouse CD44 (IM7, 560568) and BUV805 anti-mouse CD8α (53-6.7, 612898) were acquired from BD Biosciences. BUV395 anti-mouse CD44 (IM7, 363-0441-82), PerCP-Cy 5.5 anti-mouse IL-2 (JES6-5H4, 45-7021-82), PerCP-eF710 anti-mouse CD27 (O323, 46-0279-42), and PE-Cy7 anti-mouse Tim3 (RMT3-23, 12-5870-82, eBioscience) were obtained from Thermo Fisher. APC anti-human/mouse Tox (REA473, 130-118-335) was obtained from Miltenyi Biotech.
All flow cytometry antibodies are used as 1:200.

The following antibodies were used for imaging:
anti-NFAT1 (1:250, cat. 4389, Cell Signaling Technology), anti-NFAT2 (1:250, D15F1, Cell Signaling Technology) and anti-mono and dimethyl arginine (1:500, cat. ab412, Abcam), anti-rabbit Alexa Fluor plus 595 (1:1,000, cat. A-11012, Thermo-Fisher), anti-mouse CD8-APC (1:500, cat. 100712, BioLegend), Alexa fluor 488 Phalloidin (1:1000, A12379, Thermo-Fisher) and Hoechst (1:1000, H3569, Thermo-Fisher).

The following antibodies were used for cell culture: anti-CD3 (145-2C11; Bio X Cell, BE0001) and anti-CD28 (37.51; Bio X Cell, BE0015-1).

**Validation**

The specificities of listed FACS antibodies have been validated by the manufacturer by flow cytometry.
Antibodies from BioLegend
Pacific Blue anti-mouse Ly108 (330-AJ, 134608); https://www.biolegend.com/en-us/products/pacific-blue-anti-mouse-ly108-antibody-6083?GroupID=BLG7404
BV510 anti-mouse KLRG1 (2F1, 138421); https://www.biolegend.com/en-us/products/brilliant-violet-510-anti-mouse-human-klrg1-mafa-antibody-9943
BV570 anti-mouse CD62L (MEL-14, 104433); https://www.biolegend.com/en-us/products/brilliant-violet-570-anti-mouse-cd62l-antibody-7369
BV711 anti-mouse Tim-3 (RMT3-23, 119727); https://www.biolegend.com/en-us/products/brilliant-violet-711-anti-mouse-cd366-tim-3-antibody-14918
BV785 anti-mouse CD127 (A7R34, 135037); https://www.biolegend.com/en-us/products/brilliant-violet-785-anti-mouse-cd127-il-7ralpha-antibody-10803
PE-Cy5 anti-mouse Granzyme-B (QA16A02, 372226); https://www.biolegend.com/en-us/products/pe-cyanine5-anti-human-mouse-granzyme-b-recombinant-antibody-21713
PE-Fire 700 anti-mouse CD4 (GK1.5, 100484); https://www.biolegend.com/en-us/products/pefire-700-anti-mouse-cd4-antibody-19781
APC/Cy7 anti-mouse TNF-alpha (MP6-XT22, 506344); https://www.biolegend.com/en-us/products/apc-cyanine7-anti-mouse-tnf-alpha-antibody-12117
PE-Cy7 anti-mouse KLRG1 (2F1/KLRG1, 138416); https://www.biolegend.com/en-us/products/pe-cyanine7-anti-mouse-human-klrg1-mafa-antibody-8312
PerCP-Cy 5.5 anti-mouse CD62L (MEL-14, 104432):https://www.biolegend.com/en-us/products/percp-cyanine5-5-anti-mouse-cd62l-antibody-4272?GroupID=BLG10534
BV605 anti-mouse CD127 (A7R34, 135025): https://www.biolegend.com/it-it/products/brilliant-violet-605-anti-mouse-cd127-il-7ralpha-antibody-8539

Pacific blue anti-mouse CD45.1(110722),;https://www.biolegend.com/en-us/products/pacific-blue-anti-mouse-cd45-1-antibody-3105
APC anti-mouse PD1 (RL388, 109111);https://www.biolegend.com/en-us/products/apc-anti-mouse-cd279-pd-1-antibody-6672?
GroupID=BLG4702
BV711 anti-mouse PD1 (29F.1A12, 135231);https://www.biolegend.com/en-us/products/brilliant-violet-711-anti-mouse-cd279-pd-1-
antibody-12303
PE anti-mouse IFN-g (XMG1.2, 505808);https://www.biolegend.com/en-us/products/pe-anti-mouse-ifn-gamma-antibody-997
FITC anti-mouse TNFa (MP6-XT22, 506304);https://www.biolegend.com/en-us/products/fitc-anti-mouse-tnf-alpha-antibody-976
Pacific Blue anti-human CD45 (HI30, 982306);https://www.biolegend.com/en-us/products/pacific-blue-anti-human-cd45-
antibody-13991
APC/Cy7 anti-human CD8 (RPA-T8, 344713);https://www.biolegend.com/en-us/products/apc-cyanine7-anti-human-cd8-
antibody-6391

Antibody from BD Biosciences
BUV563 anti-mouse LAG3 (C9B7W, 741350); https://www.bdbiosciences.com/en-us/products/reagents/flow-cytometry-reagents/
research-reagents/single-color-antibodies-ruo/buv563-rat-anti-mouse-cd223.741350
BUV615 anti-mouse CD69 (H1.2F3, 751593); https://www.bdbiosciences.com/en-us/products/reagents/flow-cytometry-reagents/
research-reagents/single-color-antibodies-ruo/buv615-hamster-anti-mouse-cd69.751593
BUV805 anti-mouse CXCR3 (173, 748700); https://www.bdbiosciences.com/en-us/products/reagents/flow-cytometry-reagents/
research-reagents/single-color-antibodies-ruo/buv805-hamster-anti-mouse-cd183-cxcr3.748700
BV421 anti-mouse EOMES (X4-83, 567166); https://www.bdbiosciences.com/en-us/products/reagents/flow-cytometry-reagents/
research-reagents/single-color-antibodies-ruo/bv421-mouse-anti-eomes.567166
BV480 anti-mouse CD45.1 (A20, 746666); https://www.bdbiosciences.com/en-us/products/reagents/flow-cytometry-reagents/
research-reagents/single-color-antibodies-ruo/bv480-mouse-anti-mouse-cd45-1.746666
Alexa-Fluor 488 anti-mouse TCF1 (S33-966, 567018); https://www.bdbiosciences.com/en-us/products/reagents/flow-cytometry-
reagents/research-reagents/single-color-antibodies-ruo/alexa-fluor-488-mouse-anti-tcf-7-tcf-1.567018
Alexa-Fluor 647 TOX (NAN448B, 568356);https://www.bdbiosciences.com/en-us/products/reagents/flow-cytometry-reagents/
research-reagents/single-color-antibodies-ruo/alexa-fluor-647-rat-anti-tox.568356
BUV 496 anti-mouse Ly-108 (13G3, 750046);https://www.bdbiosciences.com/en-us/products/reagents/flow-cytometry-reagents/
research-reagents/single-color-antibodies-ruo/buv496-mouse-anti-mouse-ly-108.750046
APC/Cy7 anti-mouse CD44 (IM7, 560568); https://www.bdbiosciences.com/en-us/products/reagents/flow-cytometry-reagents/
research-reagents/single-color-antibodies-ruo/apc-cy-7-rat-anti-mouse-cd44.560568
BUV805 anti-mouse CD8α (53-6.7, 612898); https://www.bdbiosciences.com/en-us/products/reagents/flow-cytometry-reagents/
research-reagents/single-color-antibodies-ruo/buv805-rat-anti-mouse-cd8a.612898

Antibody from Miltenyi Biotech
APC anti-human/mouse Tox (REA473, 130-118-335); https://www.miltenyibiotec.com/US-en/products/tox-antibody-anti-human-
mouse-reafinity-rea473.html#apc:30-tests-in-60-ul

Antibody from Thermo-Fisher
PE-Cy7 anti-mouse Tim3 (RMT3-23, 12-5870-82); https://www.thermofisher.com/antibody/product/CD366-TIM3-Antibody-clone-
RMT3-23-Monoclonal/12-5870-82

The specificities of listed imaging antibodies have been validated by the manufacturer by imaging.
anti-NFAT1 (cat. 4389, Cell Signaling Technology); https://www.cellsignal.com/products/primary-antibodies/nfat1-antibody/4389
anti-NFAT2 (cat. D15F1, Cell Signaling Technology); https://www.cellsignal.com/products/primary-antibodies/nfat2-d15f1-rabbit-
mab/8032?site-search-type=Products&N=4294956287&Ntt=d15f1&fromPage=plp&_requestid=2192860
anti-mono and dimethyl arginine (cat. ab412, Abcam);https://www.abcam.com/mono-and-dimethyl-arginine-antibody-7e6-
ab412.html
anti-rabbit Alexa Fluor plus 595 (cat. A-11012, Thermo-Fisher); https://www.thermofisher.com/antibody/product/Goat-anti-Rabbit-
IgG-H-L-Cross-Adsorbed-Secondary-Antibody-Polyclonal/A-11012
Alexa fluor 488 Phalloidin (cat. A12379, Thermo-Fisher); https://www.thermofisher.com/order/catalog/product/A12379
Hoechst 33258 (cat. H3569, Thermo-Fisher); https://www.thermofisher.com/order/catalog/product/H3569

The listed antibodies have been validated by the manufacturer:
anti-CD3e (145-2C11; Bio X Cell, BE0001):https://bxcell.com/product/m-cd3e/
anti-CD28 (37.51; Bio X Cell, BE0015-1): https://bxcell.com/product/m-cd28/

# Eukaryotic cell lines

Policy information about cell lines and Sex and Gender in Research

| | |
|---|---|
| Cell line source(s) | B16-Ova was kindly provided by Hongbo Chi and MC38-Ova cell line were kindly provided by Dr. Dario Vignali. F420 cell line was provided by Dr. Jason T Yustein. Plat-e cells were purchased from Cell Biolabs (cat. RV-101). |
| Authentication | The cell line used was not authenticated |
| Mycoplasma contamination | B16-Ova, MC38, MC38-Ova, LLC, F420 were checked for mycoplasma contamination and found to be negative. |
| Commonly misidentified lines (See ICLAC register) | No commonly misidentified cell line were used. |

## Animals and other research organisms

Policy information about studies involving animals; ARRIVE guidelines recommended for reporting animal research, and Sex and Gender in Research

| | |
|---|---|
| Laboratory animals | Mice were housed and bred at the St. Jude Children's Research Hospital Animal Resource Center in specific pathogen-free conditions. Mice were on 12-hour light/dark cycles that coincide with daylight in Memphis, TN, USA. The St. Jude Children's Research Hospital Animal Resource Center housing facility was maintained at 20–25 °C and 30–70 % humidity. All genetic models were on the C57BL/6 background. Both male and female mice were used for analysis and quantification. All mice were used at 6-10 weeks old. We crossed Rosa26-Cas9 knock-in mice with OT-I/P14 transgenic mice to express Cas9 in antigen-specific CD8+ T cells (called Cas9-OT-I mice or Cas9-P14). The Cas9 mice were fully backcrossed to the C57BL/6J background. Rag1–/– mice were purchased from the Jackson Laboratory. |
| Wild animals | This study did not involve wild animals |
| Reporting on sex | Sex of the mice was not considered during experimental planning. |
| Field-collected samples | The study did not involve samples collected from the field. |
| Ethics oversight | Mouse studies were conducted in accordance with protocols approved by the St. Jude Children's Research Hospital Committee on Care and Use of Animals and in compliance with all relevant ethical guidelines. |

Note that full information on the approval of the study protocol must also be provided in the manuscript.

## Flow Cytometry

### Plots

Confirm that:

☒ The axis labels state the marker and fluorochrome used (e.g. CD4-FITC).

☒ The axis scales are clearly visible. Include numbers along axes only for bottom left plot of group (a 'group' is an analysis of identical markers).

☒ All plots are contour plots with outliers or pseudocolor plots.

☒ A numerical value for number of cells or percentage (with statistics) is provided.

### Methodology

| | |
|---|---|
| Sample preparation | The spleens and peripheral lymph nodes (PLNs) were gently separated under nylon mesh using the flat end of a 3-mL syringes. Red blood cells were removed by ACK lysing buffer, followed by washing cells with isolation buffer. After spinning down, the cell pellets were resuspended and filtered with nylon mesh before staining. For the examination of tumour infiltrating lymphocytes, tumours were excised, minced and digested with 0.5 mg/ml Collagenase IV (Roche) + 200 UI/ml DNase I (Sigma) for 40 min at 37 °C, and then passed through 70-μm filters to remove undigested tumor tissues. |
| Instrument | Fortessa (BD Bioscience) or Aurora (Cytek). |
| Software | Flowjo 10.8.2 |
| Cell population abundance | For sorting CRISPR/Cas9 generated KCa3.1WT/R350A mutant, the population varied from 25-50%. |
| Gating strategy | Based on the pattern of FSC-A/SSC-A, cells in the lymphocyte gate were used for analysis of T cell subsets. Singlets were gated according to the pattern of FSC-H vs. FSC-A. Positive populations were determined by the specific antibodies, which were distinct from negative populations. |

☒ Tick this box to confirm that a figure exemplifying the gating strategy is provided in the Supplementary Information.

