## [Peer Review File · Nature Immunology]

Early methionine availability attenuates T cell exhaustion.

Corresponding Author: Dr Douglas Green

Version 0:

Decision Letter:

9th Jan 2025

Dear Dr Green,

As you are aware we sent your revised Article, "An early, novel arginine methylation of KCa3.1 attenuates subsequent T cell exhaustion." to 3 referees. We now have reports from the original Nature reviewers 1 and 3, but we had to replace Nature reviewer 2 with a new mediating reviewer 4 to check the calcium data as reviewer 3 declined to re-review for us.

Also thanks for sending your Author Response to me. I am pleased to say that we are satisfied by this further revision plan and would thereby be happy to see a revision before we make a final decision on publication.

We therefore invite you to revise your manuscript taking into account all reviewer and editor comments. Please highlight all changes in the manuscript text file in Microsoft Word format.

* If you have not done so already please begin to revise your manuscript so that it conforms to our Article format instructions at <http://www.nature.com/ni/authors/index.html>. Refer also to any guidelines provided in this letter.

* Please include a revised version of any required reporting checklist. It will be available to referees to aid in their evaluation of the manuscript goes back for peer review. They are available here:

Reporting summary:

Please note, Extended Data figures and tables are online-only (appearing in the online PDF and full-text HTML version of the paper), peer-reviewed display items that provide essential background to the Article but are not included in the printed version of the paper due to space constraints or being of interest only to a few specialists. A maximum of ten Extended Data display items (figures and tables) is typically permitted. When re-submitting your manuscript, please ensure that any supplementary figures and tables that are more critical to the manuscript's conclusions are converted to Extended data to increase these data's visibility.

Finally, please ensure that you retain unprocessed data and metadata files after publication, ideally archiving data in

perpetuity, as these may be requested during the peer review and production process or after publication if any issues arise.

Link Redacted

We hope to receive your revised manuscript within 6 weeks. If you cannot send it within this time, please let us know. We will be happy to consider your revision so long as nothing similar has been accepted for publication at Nature Immunology or published elsewhere.

Nature Immunology is committed to improving transparency in authorship. As part of our efforts in this direction, we are now requesting that all authors identified as 'corresponding author' on published papers create and link their Open Researcher and Contributor Identifier (ORCID) with their account on the Manuscript Tracking System (MTS), prior to acceptance. ORCID helps the scientific community achieve unambiguous attribution of all scholarly contributions. You can create and link your ORCID from the home page of the MTS by clicking on 'Modify my Springer Nature account'. For more information please visit www.springernature.com/orcid.

Sincerely,

Nick Bernard, PhD
Senior Editor
Nature Immunology

Reviewers' Comments:

Original Nature Reviewer #1 (Remarks to the Author):

I have previously reviewed this paper in Nature. The authors have addressed previous comments. The work is conceptually novel and immunologically interesting. The data support their major claim.

Reviewer #4 mediating for Nature Reviewer 2 (Remarks to the Author):

In this study, Sharma et al. present a novel mechanism demonstrating how arginine methylation of the calcium-dependent potassium channel KCa3.1 regulates the fate and function of CD8 T cells. The research is conceptually innovative, intriguing and timely. However, the previous revision highlighted certain methodological limitations.

This reviewer has been assigned to replace the previous referee #2 and evaluate the authors' responses to the prior comments. In this capacity, I will refrain from "moving the goalpost" and will focus solely on assessing whether the responses adequately address the original concerns. Although I have not reviewed the initial manuscript and only have access to the revised version of the paper and the point-by-point responses to the reviewers' concerns, I hope the following comments will be helpful in further refining the manuscript for publication.

The comments from the former reviewer #2 primarily focus on the quality and controls of the calcium influx measurements conducted via flow cytometry, as well as the microscopic analysis of NFAT nuclear translocation. These concerns also extend to the statistical analysis associated with these experiments.

I agree that the overall quality of the calcium influx data is suboptimal, particularly because some of the observed differences are fairly small, and certain controls do not perform as expected. For instance, ionomycin, used as a positive control to maximize calcium influx independently of KCa3.1 channel function. One would anticipate that the differences observed between the 0.1 and 0.03 mM Met conditions or the overexpression of KCa3.1 mutants would disappear under ionomycin treatment. However, this is not the case in Fig. 3A, Fig. 5A, and Extended Fig. 6E. Similarly, the use of the KCa3.1 channel inhibitor BTP2 (YM58483) in Extended Data Fig. 4A raises concerns. Higher concentrations of this inhibitor or more effective pharmacological SOCE blockers, such as CM4620, are commercially available and should be considered. I agree with the authors that these supplemental experiments may be not essential but may be still worth to improve.

Additionally, the previous reviewer requested calcium influx data from naïve cells, which has not been provided. The authors' argument that this reflects the baseline before cross-linking with anti-hamster IgG is unconvincing, as there is no extracellular calcium present in this scenario. The appropriate control would involve adding calcium to cells in the absence of anti-CD3 activation to exclude nonspecific calcium influx caused by factors such as shear stress, membrane depolarization, or defective plasma membrane integrity.

Given that calcium influx measurements using flow cytometry are inherently challenging, noisy and often lack resolution, I recommend repeating some key experiments (e.g., Fig. 3A, 4G, 5A) using alternative methods such as time-lapse microscopy or plate reader-based calcium influx recordings. These techniques enable the use of ratiometric dyes that do not interfere with GFP or other fluorescent markers, thereby providing higher-quality data. Since all these experiments involve *in vitro* activated T cells, improving the data quality should be feasible, as this represents a critical mechanistic aspect of the study.

Furthermore, the authors should conduct proper statistical analyses of the new data. It appears that most, if not all, calcium influx measurements were conducted in duplicates. Experimental duplicates are insufficient for robust statistical evaluation, and it remains unclear whether these were independent biological experiments or merely technical replicates. The extremely low p-values reported in the respective figures raise questions about whether the experiments were biologically independent or technical repeats. Additionally, the statistical analysis should include individual data points and assess not only the AUC (as a measure of overall intracellular calcium) but also parameters such as the peak and slope of the calcium influx.

A similar concern applies to the NFAT nuclear translocation analyses presented in Fig. 3B, Fig. 5B, and Extended Fig. 4B. It is unclear how many independent experiments were performed, and the quantification differences between the 0.1 and 0.03 mM Met conditions or KCa3.1 mutant overexpression are not visible in the representative micrographs.

Given that these experiments also use *in vitro* activated T cells, a more robust approach to quantify and display these differences would be through immunoblot analyses. By assessing the amount of NFAT in separated nuclear and cytosolic fractions or detecting the size shift of NFATc2 caused by calcineurin-mediated Ser/Thr dephosphorylation in Western blot analyses, these differences would be easier to illustrate. In addition, the authors only analyzed NFATc2 (NFAT1) in this manuscript. It would be valuable to investigate whether the 0.03 mM Met condition similarly affects NFATc1 (NFAT2) nuclear translocation.

I agree with the authors that they have provided several lines of evidence (calcium influx, NFAT nuclear translocation, CUT&RUN analyses) supporting the proposed mechanism. However, this does not justify presenting data of questionable quality or statistical power. Overall, I believe it should be feasible to enhance the quality of the data through a few additional *in vitro* experiments as suggested above. Employing alternative methodological strategies could further address any concerns regarding the validity of these results and strengthen the manuscript.

Original Nature Reviewer #3 (Remarks to the Author):

The manuscript has substantially improved since the last round of reviews. I acknowledge the substantial amount of work the authors have done. The addition of new data and the restructuring of the manuscript have helped, and the data is generally interesting and conceptually novel.

I continue, however, to have problems with the immunological aspects of the study.

Figure 6, panels H and J: the FACS plots show CD8 T cells with CD8 staining displayed in the Y axis. Evidently, there is no CD8 staining. How were these cells identified? Are they CD8 T cells, or is there a mistake in data display?

Extended data Fig. 2: In panel M, the authors show that the majority of tumor infiltrating cells express high levels of IL7R (CD127) and around 50% of the cells are designated as IL7R+/CD27+ memory. In panel O, the same cells are shown to be negative for IL7R. What is the correct result?

Expanded data Fig. 8, B16 tumor data: In panel G, TCF1 vs TIM3 is shown. However, something does not add up. In the author's data TIM3+ wildtype cells are TCF1 high. This cannot be correct, the opposite is the case. Is there a mistake in the data display? In panel H, KLRG1 vs IL7R is shown. Based on this display, almost all cells are KLRG1+. This would be very unusual for B16 tumors.

Expanded data Fig. 8, LCMV data: In panel K, PD1 vs TIM3 is shown. However, the plots cannot be correct. Just looking at the WT cells, there is no difference between PD-1 expression in Armstrong vs Clone 13. Yet, in Armstrong most cells should be negative for PD-1, while in clone 13 all cells should be positive for PD-1. Either the stain did not work or there is a mistake in the data display.

I suggest checking all of the FACS results carefully again.

A remark related to terminology: the authors refer to "precursors of exhaustion (TEXprog) and TEXprogenitor-like cells, both of which produce a replicative burst and effector function upon checkpoint blockade". I am not sure what TEXprogenitor-like cells are. The authors define them as Ly108+CX3CR1+ (line 113), which is an unusual phenotype. It would be useful if the

authors would point to a reference where this population was defined.

Related to this point, in figure 6L, they define TEXprog as CX3CR1+PD1+. This is incorrect. CX3CR1+PD1+ cells are effector-like cells, not progenitors (Zander et al Immunity, Hudson et al Immunity). Progenitors should be identified as TCF1+.

The authors refer to “memory precursors in chronic infection” (line 305). There are no memory precursors in chronic infection. These cells would be TEXprog

These points may appear to be semantics. But they make a big difference when it comes to interpreting the data. Thus, I feel strongly that the data need to reflect the reality and need to be interpreted correctly.

Additional point:

In their results, they refer to “differentially enriched (DE) genes in T cells” (line 124). Do they mean “differentially expressed” genes? What analysis was done here?

Version 1:

Decision Letter:

16th Apr 2025

Dear Dr Green,

As you are aware, your Article, "An early, novel arginine methylation of KCa3.1 attenuates subsequent T cell exhaustion." has now been seen again by referees 2 and 3 and whilst reviewer 2 is now happy with the calcium data, reviewer 3 maintains some issues with the FACS data.

We have also looked over your Author Response document (thanks for emailing that to me) and we are satisfied by the plan to remove some FACS data and clean up some other parts. If you do this final short revision we will seek a new reviewer to mediate and comment on the discussion between yourself and reviewer 3

* Include a “Response to referees” document detailing, point-by-point, how you addressed each referee comment. If no action was taken to address a point, you must provide a compelling argument. This response will be sent back to the referees along with the revised manuscript.

* If you have not done so already please begin to revise your manuscript so that it conforms to our Article format instructions at <http://www.nature.com/ni/authors/index.html>. Refer also to any guidelines provided in this letter.

* Please include a revised version of any required reporting checklist. It will be available to referees to aid in their evaluation of the manuscript goes back for peer review. They are available here:

Reporting summary:

When submitting the revised version of your manuscript, please pay close attention to our <https://www.nature.com/nature-portfolio/editorial-policies/image-integrity> Digital Image Integrity Guidelines. and to the following points below:

Please note, Extended Data figures and tables are online-only (appearing in the online PDF and full-text HTML version of the paper), peer-reviewed display items that provide essential background to the Article but are not included in the printed version of the paper due to space constraints or being of interest only to a few specialists. A maximum of ten Extended Data display items (figures and tables) is typically permitted. When re-submitting your manuscript, please ensure that any supplementary figures and tables that are more critical to the manuscript’s conclusions are converted to Extended data to increase these data’s visibility.

Link Redacted

We hope to receive your revised manuscript within two weeks. If you cannot send it within this time, please let us know. We will be happy to consider your revision so long as nothing similar has been accepted for publication at Nature Immunology or published elsewhere.

Nature Immunology is committed to improving transparency in authorship. As part of our efforts in this direction, we are now requesting that all authors identified as 'corresponding author' on published papers create and link their Open Researcher and Contributor Identifier (ORCID) with their account on the Manuscript Tracking System (MTS), prior to acceptance. ORCID helps the scientific community achieve unambiguous attribution of all scholarly contributions. You can create and link your ORCID from the home page of the MTS by clicking on 'Modify my Springer Nature account'. For more information please visit www.springernature.com/orcid.

Sincerely,

Nick Bernard, PhD
Senior Editor
Nature Immunology

Reviewers' Comments:

Reviewer #2 (Remarks to the Author):

The authors have done an excellent job and now provide satisfactory calcium influx data. The main finding that methionine limitation leads to increased calcium influx, NFATc2 hyperactivation, and ultimately, T cell exhaustion is well supported by the new data. Notably, the observation that defective KCa3.1 methylation enhances calcium influx, selectively affecting NFATc2 but not NFATc1, suggests distinct activation thresholds and/or functions among NFAT family members during T cell exhaustion.

While these mechanistic details are intriguing, they extend beyond the scope of this study. Overall, I recommend the revised manuscript for publication in Nature Immunology.

Minor comment: Please double-check line 115, as it appears the authors may have omitted a reference.

Reviewer #3 (Remarks to the Author):

Unfortunately, this reviewer continues to doubt the quality of the flow cytometry data and thereby the interpretation of the related results. The additional data provided by the authors have solidified these concerns.

In their response to my concerns related to original figure Ext Data Fig 8K, the authors argue that no difference in PD-1 expression is expected when comparing LCMV cl13 vs Armstrong at day 9. This is incorrect. Although PD-1 is upregulated in acute and chronic infection during the initial phase of the response, it is always much higher in chronic infection. Furthermore, by day 9 post infection, the peak of response, PD-1 is essentially off on acute infection cells, while it remains very high on chronic infection cells. The reason for these odd-looking results can be found in the highly problematic gating strategy shown in Ext Data Fig 10A. It is clear that neither the TCRb nor the CD8 stain worked. Accordingly, the downstream analysis (Ext Data Fig. 10C-G) is uninterpretable.

Extended Data Fig. 9I remains problematic (as pointed out in the last submission). In their response, the authors propose that high KLRG1 expression in all tumor infiltrating T cells is due to prior activation. That is not the case. KLRG1 is not induced in vitro.

Related: The new data in Ext Data Fig 2O,P look odd. Based on this analysis, all cells are KLRG1+ including the IL-7R+ cells. This is not the case. It would also imply that the majority of tumor-infiltrating T cells is IL-7R+. That is incorrect.

New data in Ext Data Fig 9G look odd. Based on this all cells are CD62L+. That is not the case in tumors.

Version 2:

Decision Letter:

Our ref: NI-A38888B

25th Apr 2025

Dear Dr. Green,

Thank you for submitting your revised manuscript "An early, novel arginine methylation of KCa3.1 attenuates subsequent T cell exhaustion." (NI-A38888B). It has now been seen by a new mediating reviewer to look at your response to reviewer 2.

The reviewer finds that the paper has improved in revision, and therefore we'll be happy in principle to publish it in Nature Immunology, pending minor revisions to satisfy the referees' final requests and to comply with our editorial and formatting guidelines.

We will now perform detailed checks on your paper and will send you a checklist detailing our editorial and formatting requirements in about a week. Please do not upload the final materials and make any revisions until you receive this additional information from us.

If you had not uploaded a Word file for the current version of the manuscript, we will need one before beginning the editing process; please email that to immunology@us.nature.com at your earliest convenience.

Thank you again for your interest in Nature Immunology Please do not hesitate to contact me if you have any questions.

Sincerely,

Nick Bernard, PhD
Senior Editor
Nature Immunology

Reviewer #4 (Remarks to the Author):

Most of the concerns have been effectively addressed in the revision. >50% CD62L+ OT1 is unexpected. However, the authors used Rag1^{-/-} mice instead of immunocompetent mice. It would be helpful to briefly discuss the potential limitations of using Rag1^{-/-} mice.

Reviewer #1

(Remarks to the Author)

I have previously reviewed this paper in Nature. The authors have addressed previous comments. The work is conceptually novel and immunologically interesting. The data support their major claim.

We thank the reviewer for these supportive comments.

Reviewer #2

(Remarks to the Author)

In this study, Sharma et al. present a novel mechanism demonstrating how arginine methylation of the calcium-dependent potassium channel KCa3.1 regulates the fate and function of CD8 T cells. The research is conceptually innovative, intriguing and timely. However, the previous revision highlighted certain methodological limitations.

We appreciate the reviewer's enthusiasm for our findings.

This reviewer has been assigned to replace the previous referee #2 and evaluate the authors' responses to the prior comments. In this capacity, I will refrain from "moving the goalpost" and will focus solely on assessing whether the responses adequately address the original concerns. Although I have not reviewed the initial manuscript and only have access to the revised version of the paper and the point-by-point responses to the reviewers' concerns, I hope the following comments will be helpful in further refining the manuscript for publication.

We thank the reviewer for these useful suggestions and hope that in addressing them we have significantly improved the paper.

The comments from the former reviewer #2 primarily focus on the quality and controls of the calcium influx measurements conducted via flow cytometry, as well as the microscopic analysis of NFAT nuclear translocation. These concerns also extend to the statistical analysis associated with these experiments.

I agree that the overall quality of the calcium influx data is suboptimal, particularly because some of the observed differences are fairly small, and certain controls do not perform as expected. For instance, ionomycin, used as a positive control to maximize calcium influx independently of KCa3.1 channel function. One would anticipate that the differences observed between the 0.1 and 0.03 mM Met conditions or the overexpression of KCa3.1 mutants would disappear under ionomycin treatment. However, this is not the case in Fig. 3A, Fig. 5A, and Extended Fig. 6E. Similarly, the use of the KCa3.1 channel inhibitor BTP2 (YM58483) in Extended Data Fig. 4A raises concerns. Higher concentrations of this inhibitor or more effective pharmacological SOCE blockers, such as CM4620, are

commercially available and should be considered. I agree with the authors that these supplemental experiments may be not essential but may be still worth to improve.

We now show all of the calcium flux experiments repeated with the ratiometric dye Indo-1 including a control with an ionomycin concentration of 5 μ M (Fig. 3A-C, 4G-I, 5A-C, Extended Data Fig. 4A-C). We show AUC and maximum peak data from repeated experiments. We now also show YM58483 at a higher concentration with significant effect on calcium flux (Extended Data Fig. 4F-G). We hope that the reviewer will agree that the results obtained with this method are more robust and convincing than what we had previously shown.

Additionally, the previous reviewer requested calcium influx data from naïve cells, which has not been provided. The authors' argument that this reflects the baseline before cross-linking with anti-hamster IgG is unconvincing, as there is no extracellular calcium present in this scenario. The appropriate control would involve adding calcium to cells in the absence of anti-CD3 activation to exclude nonspecific calcium influx caused by factors such as shear stress, membrane depolarization, or defective plasma membrane integrity.

We now provide the calcium flux data (Indo-1) of naïve samples (with and without CD3 crosslinking). We hope that the reviewer appreciates that the naïve samples show minimal flux, validating our activation signaling and accompanying calcium flux conclusions (Fig. 3A-C). We also performed calcium flux recording by activating cells in Ca²⁺-sufficient ringer solution and observed similar differences of increased calcium flux in the 0.03mM Met condition (Extended Data Fig. 4A-C).

Given that calcium influx measurements using flow cytometry are inherently challenging, noisy and often lack resolution, I recommend repeating some key experiments (e.g., Fig. 3A, 4G, 5A) using alternative methods such as time-lapse microscopy or plate reader-based calcium influx recordings. These techniques enable the use of ratiometric dyes that do not interfere with GFP or other fluorescent markers, thereby providing higher-quality data. Since all these experiments involve in vitro activated T cells, improving the data quality should be feasible, as this represents a critical mechanistic aspect of the study.

Unfortunately, we do not have access to UV-laser enabled microscopy and therefore we could not use ratiometric dyes (Indo-1 or Fura-2). Furthermore, in our hands our departmental plate-reader was not sensitive enough to provide satisfactory data with CD8 T cells. We acquired all of the data on a Cytex spectral flow cytometer, ensuring the quality and sensitivity of the acquired data. We hope that the reviewer appreciates the provided new data using Indo-1.

We would like to stress that there is no controversy surrounding the fact that calcium flux via CRAC channels occurs following TCR signaling in T cells, nor are we suggesting otherwise. Using higher concentrations of the inhibitor produced better results.

Furthermore, the authors should conduct proper statistical analyses of the new data. It appears that most, if not all, calcium influx measurements were conducted in duplicates. Experimental duplicates are insufficient for robust statistical evaluation, and it remains unclear whether these were independent biological experiments or merely technical replicates. The extremely low p-values reported in the respective figures raise questions about whether the experiments were biologically independent or technical repeats. Additionally, the statistical analysis should include individual data points and assess not only the AUC (as a measure of overall intracellular calcium) but also parameters such as the peak and slope of the calcium influx.

We now provide the AUC and max peak values from individual biological replicate with all of the experiments. Each experiment was repeated several times (and the major claim, that restricting Met during activation increases Ca^{2+} influx, was repeated extensively not only in the initial comparisons but also in the presence or absence of TRAM34). The AUC and max peak data shown is the combination of all the experiments in each case. We are not showing slope as we did not observe any difference in the slope upon addition of Ca^{2+} . The p-values in experiments using Fluo-8 are calculated via binning the samples, essentially taking into account the changes at the single cell level and we have removed these statistics (providing the data from Fluo-8 as support of our Indo-1 results). For the indo-1 experiments, statistics is now performed comparing separate experiments.

A similar concern applies to the NFAT nuclear translocation analyses presented in Fig. 3B, Fig. 5B, and Extended Fig. 4B. It is unclear how many independent experiments were performed, and the quantification differences between the 0.1 and 0.03 mM Met conditions or KCa3.1 mutant overexpression are not visible in the representative micrographs.

Given that these experiments also use in vitro activated T cells, a more robust approach to quantify and display these differences would be through immunoblot analyses. By assessing the amount of NFAT in separated nuclear and cytosolic fractions or detecting the size shift of NFATc2 caused by calcineurin-mediated Ser/Thr dephosphorylation in Western blot analyses, these differences would be easier to illustrate. In addition, the authors only analyzed NFATc2 (NFAT1) in this manuscript. It would be valuable to investigate whether the 0.03 mM Met condition similarly affects NFATc1 (NFAT2) nuclear translocation.

We have now marked the nucleus with dashed lines to highlight the NFAT1 staining. We hope that the reviewer appreciates that the naïve samples (Fig. 3D) shown little to no staining of nuclear NFAT1, showing the specificity of the antibody and staining procedure. Upon activation, NFAT1 staining is visible in the nucleus, which is inhibited upon cyclosporin treatment, as expected. The quantification is shown at a single cell level (~40 cells/samples). Each experiment was repeated at least two times, often with similar groups being compared across experiments. We did not observe any significant differences in NFAT2 nuclear translocation between 0.1 mM and 0.03 mM Met (Extended Data Fig. 5E).

Due to the small size of CD8 T cells (6-10 μ) we encountered technical challenges in nuclear fractionation. We did seem to observe increased nuclear NFAT1 in cells activated in 0.03mM Met, but the fractionation was suboptimal (based on control nuclear and cytosolic proteins), and therefore we are not comfortable in adding the data to the publication. Also, respectfully, we disagree that immunoblot analysis is “more robust” than microscopy, since that latter assesses nuclear localization at the single cell level, which is not possible by immunoblot.

I agree with the authors that they have provided several lines of evidence (calcium influx, NFAT nuclear translocation, CUT&RUN analyses) supporting the proposed mechanism. However, this does not justify presenting data of questionable quality or statistical power.

We appreciate that the reviewer recognizes that we have drawn our conclusions by multiple, independent approaches that all of these support our conclusions.

Overall, I believe it should be feasible to enhance the quality of the data through a few additional in vitro experiments as suggested above. Employing alternative methodological strategies could further address any concerns regarding the validity of these results and strengthen the manuscript.

We have done many additional experiments and hope that the reviewer agrees that the quality of the presented data is greatly improved.

Reviewer #3

(Remarks to the Author)

The manuscript has substantially improved since the last round of reviews. I acknowledge the substantial amount of work the authors have done. The addition of new data and the restructuring of the manuscript have helped, and the data is generally interesting and conceptually novel.

I continue, however, to have problems with the immunological aspects of the study.

Figure 6, panels H and J: the FACS plots show CD8 T cells with CD8 staining displayed in the Y axis. Evidently, there is no CD8 staining. How were these cells identified? Are they CD8 T cells, or is there a mistake in data display?

The data was acquired on a Cytex spectral flow cytometer, with appropriate single color controls. We now provide the gating strategies of the flow cytometry analysis showing the identification of transferred Thy1.1 CD8 T from the CD8 T cell population (Extended Data Fig. 10A). Because we have used an extensive antibody panel in our analyses, and since only CD8⁺ T cells were transferred, we used the fluor with the lowest emission to stain CD8, which can give the appearance that CD8 levels were low. Hopefully, showing the

gating strategy helps to rectify this concern.

Extended data Fig. 2: In panel M, the authors show that the majority of tumor infiltrating cells express high levels of IL7R (CD127) and around 50% of the cells are designated as IL7R+/CD27+ memory. In panel O, the same cells are shown to be negative for IL7R. What is the correct result?

We apologize for the mistake and the resulting confusion. The data shown originally in panel M and panel O were from different tumor models (B16 and MC38). We have revised the figure to show the data from the B16 model in Extended data Figure 2P and show data from the MC38 experiment in Extended Data Fig. 2Q as further support for our conclusions using a different tumor model.

Expanded data Fig. 8, B16 tumor data: In panel G, TCF1 vs TIM3 is shown. However, something does not add up. In the author's data TIM3+ wildtype cells are TCF1 high. This cannot be correct, the opposite is the case. Is there a mistake in the data display? In panel H, KLRG1 vs IL7R is shown. Based on this display, almost all cells are KLRG1+. This would be very unusual for B16 tumors.

We apologize as we should have been clear in the manuscript. These samples have an EGFP signal, which can interfere with the TCF1 antibody (AF-488). Despite the background correction, we were not able to get clear plots. We have now replaced this figure to show TCF1 using TOX+Tim-3+ (Fig. 6E, Extended Data Fig.9H), which avoids this artifact.

We observe similar KLRG1 expression in all of the B16 tumors with our flow cytometry panel (Extended Data Fig. 2P, 9I). We believe that as they are ex-vivo activated OT-I T cells, it is possible that they are prone to persistent priming and therefore display a global increase in KLRG1 expression in this system.

Expanded data Fig. 8, LCMV data: In panel K, PD1 vs TIM3 is shown. However, the plots cannot be correct. Just looking at the WT cells, there is no difference between PD-1 expression in Armstrong vs Clone 13. Yet, in Armstrong most cells should be negative for PD-1, while in clone 13 all cells should be positive for PD-1. Either the stain did not work or there is a mistake in the data display.

We understand the reviewers concern, however the experiment was done at D9 post infection, so the T cells are not fully exhausted. Multiple studies have shown that in both acute and chronic infection, T cells express PD1, Tim-3 along with other exhaustion specific markers as early as D5 post-infection (McManus et. al. Nature, 2025; Bally et. al. J Immunol, 2020; Utzschneider et. al, JEM, 2016; Jin et. al. PNAS, 2010). Due to this, we also suggest that in our infection experiments, the KCa3.1R350A mutant promotes T cells towards an exhaustion trajectory.

I suggest checking all of the FACS results carefully again.

A remark related to terminology: the authors refer to “precursors of exhaustion (TEXprog) and TEXprogenitor-like cells, both of which produce a replicative burst and effector function upon checkpoint blockade”. I am not sure what TEXprogenitor-like cells are. The authors define them as Ly108+CX3CR1+ (line 113), which is an unusual phenotype. It would be useful if the authors would point to a reference where this population was defined.

To avoid confusion, we have removed this figure and do not now use the term.

Related to this point, in figure 6L, they define TEXprog as CX3CR1+PD1+. This is incorrect. CX3CR1+PD1+ cells are effector-like cells, not progenitors (Zander et al Immunity, Hudson et al Immunity). Progenitors should be identified as TCF1+.

We thank the reviewer for pointing this out. We have now changed this to show TCF1+Tim-3- TEXprog (Fig. 6L).

The authors refer to “memory precursors in chronic infection” (line 305). There are no memory precursors in chronic infection. These cells would be TEXprog

We understand the reviewer’s point, however this data is also from day 9 post infection, so T cells are not exhausted. We cannot call them TEXprog as we gate via KLRG1 and CD127, which is used to differentiate terminal effector and memory precursor T cells. We could call these “memory precursor-like” cells (based on markers) if the reviewer feels that would be more acceptable.

These points may appear to be semantics. But they make a big difference when it comes to interpreting the data. Thus, I feel strongly that the data need to reflect the reality and need to be interpreted correctly.

Additional point:

In their results, they refer to “differentially enriched (DE) genes in T cells” (line 124). Do they mean “differentially expressed” genes? What analysis was done here?

We thank the reviewer for noticing the mistake, which is now corrected. The bioinformatic analysis used for differential expression is described in the method section.

We hope that in addressing these comments the reviewer will agree that we have significantly improved the paper.

Reviewers' Comments:

Reviewer #2 (Remarks to the Author):

The authors have done an excellent job and now provide satisfactory calcium influx data. The main finding that methionine limitation leads to increased calcium influx, NFATc2 hyperactivation, and ultimately, T cell exhaustion is well supported by the new data. Notably, the observation that defective KCa3.1 methylation enhances calcium influx, selectively affecting NFATc2 but not NFATc1, suggests distinct activation thresholds and/or functions among NFAT family members during T cell exhaustion.

While these mechanistic details are intriguing, they extend beyond the scope of this study. Overall, I recommend the revised manuscript for publication in Nature Immunology.

Minor comment: Please double-check line 115, as it appears the authors may have omitted a reference.

We thank the reviewer for this positive response and for their suggestions and help in improving the manuscript. We checked line 115 and have corrected our mistake and added the appropriate references.

Reviewer #3 (Remarks to the Author):

Unfortunately, this reviewer continues to doubt the quality of the flow cytometry data and thereby the interpretation of the related results. The additional data provided by the authors have solidified these concerns.

In their response to my concerns related to original figure Ext Data Fig 8K, the authors argue that no difference in PD-1 expression is expected when comparing LCMV cl13 vs Armstrong at day 9. This is incorrect. Although PD-1 is upregulated in acute and chronic infection during the initial phase of the response, it is always much higher in chronic infection. Furthermore, by day 9 post infection, the peak of response, PD-1 is essentially off on acute infection cells, while it remains very high on chronic infection cells. The reason for these odd-looking results can be found in the highly problematic gating strategy shown in Ext Data Fig 10A. It is clear that neither the TCRb nor the CD8 stain worked. Accordingly, the downstream analysis (Ext Data Fig. 10C-G) is uninterpretable.

We understand the reviewer's concern; however, we analyzed Thy1.1 cells which were transferred into the mice prior to the infection. As we highlighted in our last response that

these are P14 transgenic T cells and due to the dynamic state of TCRb and CD8 and the weak fluors used, the staining was low. To resolve the potential issue, we have now reanalyzed the data by gating directly on Thy1.1, instead of gating from CD8+ cells (Extended Data Fig. 10A). We hope now the reviewer appreciates the plots of TCRb and Thy1.1. All the downstream analysis is done on the Thy1.1+ population.

We apologize for the confusion, but we never proposed that no differences in PD1 expression were observed between acute and chronic infection but rather emphasized the observed difference in PD1+Tim-3+ cells (we never showed the quantification of only PD1 expression). We also respectfully disagree with the reviewer that PD1 is essentially turned off at the peak of response in acute infection. As these are transgenic P14 cells, the expression of PD1 along with other activation and effector specific markers is not turned off (Jin et. al, PNAS, 2010; Fagerberg et. al. Science, 2025; Bally et. al. JI, 2020). Nevertheless, as we show that KCa3.1 mutant drives T cells towards exhaustion, an event primarily associated with chronic infection, we have removed the entire acute infection data and chose to show data from chronic infection only. We hope that this removes any remaining doubts concerning the interpretation of the data.

Extended Data Fig. 9I remains problematic (as pointed out in the last submission). In their response, the authors propose that high KLRG1 expression in all tumor infiltrating T cells is due to prior activation. That is not the case. KLRG1 is not induced in vitro.

Related: The new data in Ext Data Fig 2O,P look odd. Based on this analysis, all cells are KLRG1+ including the IL-7R+ cells. This is not the case. It would also imply that the majority of tumor-infiltrating T cells is IL-7R+. That is incorrect.

We have removed all of the KLRG1 CD127 data to avoid any potential confusion for the readers.

New data in Ext Data Fig 9G look odd. Based on this all cells are CD62L+. That is not the case in tumors.

We have replaced the Ext Data Fig 9G with TEXprog and TEXterm identified via expression levels of TCF1/Tim-3 (Fig 6E-F). We further show the expression of CD62L as separate data which shows that not all the cells are CD62L+, but there is an effect of the mutant KCa3.1 on the levels of CD62L+ cells (Extended Data Fig. 9J-K).

We appreciate the reviewer's rigor in the critique and hope that with these changes and re-analyses, the reviewer agrees that our results support our conclusions.

Reviewer #4:

Remarks to the Author:

Most of the concerns have been effectively addressed in the revision. >50% CD62L+ OT1 is unexpected. However, the authors used Rag1^{-/-} mice instead of immunocompetent mice. It would be helpful to briefly discuss the potential limitations of using Rag1^{-/-} mice.

We thank the reviewer for assessing our response to the previous reviewer's concern. We have now added a sentence describing our justification for using Rag1^{-/-} mice to assess the responses of activated and transferred T cells to tumors in the manuscript (line 385-389) and note that our general findings are in line with our infection experiments in immunocompetent mice.

We thank the reviewer again for reviewing our manuscript.